# Single cell spatial analysis reveals the topology of immunomodulatory purinergic signaling in glioblastoma

Shannon Coy[1,2,3], Shu Wang[2,3,4], Sylwia A. Stopka[5], Jia-Ren Lin[2,3], Clarence Yapp [2,3], Cecily C. Ritch [1,2], Lisa Salhi [6], Gregory J. Baker [2,3], Rumana Rashid[1,2,7], Gerard Baquer [5], Michael Regan[5], Prasidda Khadka [8,9], Kristina A. Cole[10], Jaeho Hwang[1], Patrick Y. Wen[11,12], Pratiti Bandopadhayay [8,9], Mariarita Santi[13,14], Thomas De Raedt[10], Keith L. Ligon [1,8,15,16], Nathalie Y. R. Agar [5,17], Peter K. Sorger [2,3,17], Mehdi Touat [6,11,17] ✉ & Sandro Santagata [1,2,3,17] ✉

How the glioma immune microenvironment fosters tumorigenesis remains incompletely defined. Here, we use single-cell RNA-sequencing and multiplexed tissue-imaging to characterize the composition, spatial organization, and clinical significance of extracellular purinergic signaling in glioma. We show that microglia are the predominant source of CD39, while tumor cells principally express CD73. In glioblastoma, CD73 is associated with *EGFR* amplification, astrocyte-like differentiation, and increased adenosine, and is linked to hypoxia. Glioblastomas enriched for CD73 exhibit inflammatory microenvironments, suggesting that purinergic signaling regulates immune adaptation. Spatially-resolved single-cell analyses demonstrate a strong spatial correlation between tumor-CD73 and microglial-CD39, with proximity associated with poor outcomes. Similar spatial organization is present in pediatric high-grade gliomas including H3K27M-mutant diffuse midline glioma. These data reveal that purinergic signaling in gliomas is shaped by genotype, lineage, and functional state, and that core enzymes expressed by tumor and myeloid cells are organized to promote adenosine-rich microenvironments potentially amenable to therapeutic targeting.

[1]Department of Pathology, Brigham and Women's Hospital, Harvard Medical School, Boston, MA, USA. [2]Laboratory of Systems Pharmacology, Harvard Program in Therapeutic Science, Boston, MA, USA. [3]Ludwig Center at Harvard, Harvard Medical School, Boston, MA, USA. [4]Harvard Graduate Program in Biophysics, Harvard University, Boston, MA, USA. [5]Department of Neurosurgery, Brigham and Women's Hospital, Harvard Medical School, Boston, MA, USA. [6]Sorbonne Université, Inserm, CNRS, UMR S 1127, Institut du Cerveau et de la Moelle Epinière, and AP-HP Hôpitaux Universitaires La Pitié Salpêtrière - Charles Foix, Service de Neurologie 2-Mazarin, Paris, France. [7]Pitt-CMU Medical Scientist Training Program, University of Pittsburgh-Carnegie Mellon, Pittsburgh, PA, USA. [8]Department of Pediatric Oncology, Dana-Farber Boston Children's Cancer and Blood Disorders Center, Boston, MA, USA. [9]Broad Institute of MIT and Harvard, Cambridge, MA, USA. [10]Children's Hospital of Philadelphia, University of Pennsylvania, Pennsylvania, PA, USA. [11]Department of Neurology, Brigham and Women's Hospital, Harvard Medical School, Boston, MA, USA. [12]Center for Neuro-Oncology, Dana-Farber Cancer Institute, Boston, MA, USA. [13]Department of Pathology and Laboratory Medicine, Children's Hospital of Philadelphia, Philadelphia, PA, USA. [14]Department of Pathology and Laboratory Medicine, University of Pennsylvania Perelman School of Medicine, Philadelphia, PA, USA. [15]Department of Pathology, Dana-Farber Cancer Institute, Boston, MA, USA. [16]Department of Pathology, Boston Children's Hospital, Boston, MA, USA. [17]These authors jointly supervised this work: Nathalie Y. R. Agar, Peter K. Sorger, Mehdi Touat, Sandro Santagata. ✉e-mail: mehdi.touat@aphp.fr; ssantagata@bics.bwh.harvard.edu

Cancers develop numerous molecular mechanisms to evade immune surveillance[1]. These include expression of 'checkpoint' proteins that suppress cytotoxic cells[2], production of local and systemic inflammatory mediators[3], recruitment of regulatory immune cells[4], and alteration of the tumor metabolic 'niche'[5]. However, many unresolved questions remain regarding the specific roles of tumor and immune cells in generating a microenvironment with the permissive immunologic conditions necessary for tumor growth and progression. Classical histopathological analysis suggests that specific patterns of cell–cell interaction (cell neighborhoods) are critical for tumorigenesis, but the molecular mechanisms that give rise to and maintain these neighborhoods remain poorly understood.

A detailed characterization of tissue structure and function can now be achieved by integrating information from multiple analytical modalities. Single cell RNA-sequencing (scRNA-seq) and other transcriptomic methods provide a cellular inventory of cancer and immune cell types and states, while recently developed multiplexed tissue imaging technologies[6–11] permit investigation of the spatial organization of these cells and interactions among them[12]. Cyclic immunofluorescence (CyCIF)[9] is one such method, enabling quantification of up to 60 antigens at single-cell resolution and generation of maps of functional neighborhoods. Integrating data from scRNA-seq and multiplexed tissue imaging has begun to reveal substantial spatial and molecular variation within human tumors[13–15], and population-level regulation of tumor-immune interactions[16]. Such studies promise to reveal novel vulnerabilities in difficult to treat cancers.

One such cancer is glioblastoma, the most common adult primary brain tumor and a highly aggressive disease with a dismal prognosis[17]. Median survival is approximately one-year and long-term survival is extremely rare. Pediatric high-grade gliomas (HGG), including H3K27M-mutant diffuse midline glioma—a highly aggressive tumor that involves midline structures such as the brainstem and thalamus—exhibit similarly poor outcomes with a 2-year survival of <20%[18]. Unfortunately, numerous investigational therapies including genotype-targeted approaches have thus far failed to provide substantial clinical benefits[19]. PD-1 checkpoint pathway inhibitors initially showed little clinical efficacy in recurrent/residual glioblastoma, including those with high tumor mutation burden[20–23]. However, neoadjuvant administration of PD-1 inhibitor was found to improve survival, particularly in steroid-naïve patients, though the effects remain modest and long-term survival has not yet been achieved[24,25]. These findings suggest that modulation of multiple immunoregulatory pathways may be necessary to achieve substantial therapeutic benefit in HGG, and that this will require a better understanding of the tumor microenvironment.

The extracellular purine pathway is increasingly recognized as an important regulator of the tumor microenvironment and of tumorigenesis[26]. Purine metabolites such as ATP and adenosine are phylogenetically ancient biochemical compounds found in the microenvironment that link genetics, metabolism, and cell behavior in diverse organisms from bacteria to humans[27]. Extracellular purine metabolites are mediators of inflammatory signaling in normal and pathologic tissues. Adenosine triphosphate (ATP) may be released into the extracellular space in damaged or neoplastic tissues as a result of apoptosis, necrosis, exocytosis, or channel-mediated transport, and drives pro-inflammatory signaling[28]. Conversely, adenosine generally functions to suppress inflammation via binding to adenosine receptors on multiple cell types. Release of extracellular ATP followed by catabolism to adenosine is one mechanism by which inflammatory responses are first amplified and then controlled and self-limited in tissues. In the central nervous system (CNS), extracellular purine metabolite levels have additional significance, as they may also function as neurotransmitters and neuroregulatory ligands that signal to neuronal and glial populations at both synapses and extra-synaptic sites[29]. It is increasingly appreciated that neural signaling

mechanisms may be co-opted to support the growth of primary brain tumors[30].

The extracellular purine metabolites ATP and ADP are hydrolyzed to AMP via the enzymatic activity of CD39 (ENTPD1), an integral plasma membrane ectonucleotidase predominantly expressed by immune cells and vascular endothelium. AMP is then catabolized to adenosine via ecto-5′-nucleotidase CD73 (NT5E), a glycosyl-phosphatidylinositol (GPI)-anchored extracellular protein which is typically expressed by regulatory T-cells (Tregs) and other immune cells. Conversion of AMP to adenosine is the rate-limiting step in catabolism of extracellular purines, and CD73 is, therefore, a critical regulator of local purinergic signaling and inflammatory responses[27].

Aberrant CD73 expression occurs in numerous cancers, and is hypothesized to result in elevated adenosine levels, which may promote tumor cell proliferation, survival, and immune evasion via effects on multiple cell types[31,32]. CD73 knockout in mouse models augments anti-tumor immunity, increasing infiltrating T cells and reducing metastases, suggesting that CD73 inhibition may be effective as an immunomodulatory therapeutic approach[33]. Prior studies have suggested that CD73 contributes to the pathogenesis of glioblastoma in adults through dysregulation of purinergic signaling[34–38], and expression of CD73 has been correlated with increased tumor cell proliferation[31], cell-adhesion[39], invasiveness, and NK cell infiltration[40]. However, most studies to date have examined non-physiologic glioma cell lines, transgenic animal models, or small sets of tumor resections. These studies have not characterized spatial and population-based features of the immune microenvironment in human glioblastoma tissue. Moreover, the purinergic signaling pathway has not been characterized to date in pediatric gliomas and other CNS tumors.

In this study, we use multiplexed tissue imaging, single-cell RNA-sequencing, and tissue mass spectrometry imaging to explore the activity and significance of immunomodulatory purinergic signaling in adult and pediatric high-grade gliomas (HGGs). We show that this pathway is regulated by heterogeneous cellular components, including CD39 expressing microglia positioned in close proximity to CD73-expressing tumor cells. CD73 levels correlate with tumor genotype and differentiation-state, levels of tissue adenosine, and patient outcome in adult glioblastoma. While pediatric tumors harbor genetic drivers distinct from those in adult gliomas, they also exhibit enrichment for interactions between CD73-expressing tumor cells and CD39-expressing microglia, suggesting that the spatial organization of core components of extracellular purinergic metabolism contributes to the development of brain tumors in both age groups. Our results suggest that inhibition of purine signaling represents a potentially attractive immunotherapeutic strategy in both adult and pediatric HGG and confirms that characterization of multiple cell types and states through the integration of genetic and spatially-resolved tissue-based imaging is necessary to advance our understanding of the tumor microenvironment.

## Results

### Cell type and state-specific expression of the core purinergic effectors CD39 and CD73 in glioblastoma

Given that high-grade gliomas (HGG) are complex and exhibit marked genomic and cellular heterogeneity within and between tumors[41,42], we sought to leverage multiple single-cell methods and datasets to provide a detailed characterization of the immunoregulatory landscape of extracellular purine signaling. We first analyzed public scRNA-seq data from 21 adult IDH-wildtype glioblastoma (IDH-WT GBM) and 7 pediatric HGG resections encompassing 24,131 total cells[41]. As previously described, most cells in these 28 specimens clustered into four broad lineage categories according to multi-gene expression signatures. These categories were previously labeled as tumor cells, oligodendrocytes, 'T cells', and 'macrophages' (Fig. 1A). We independently

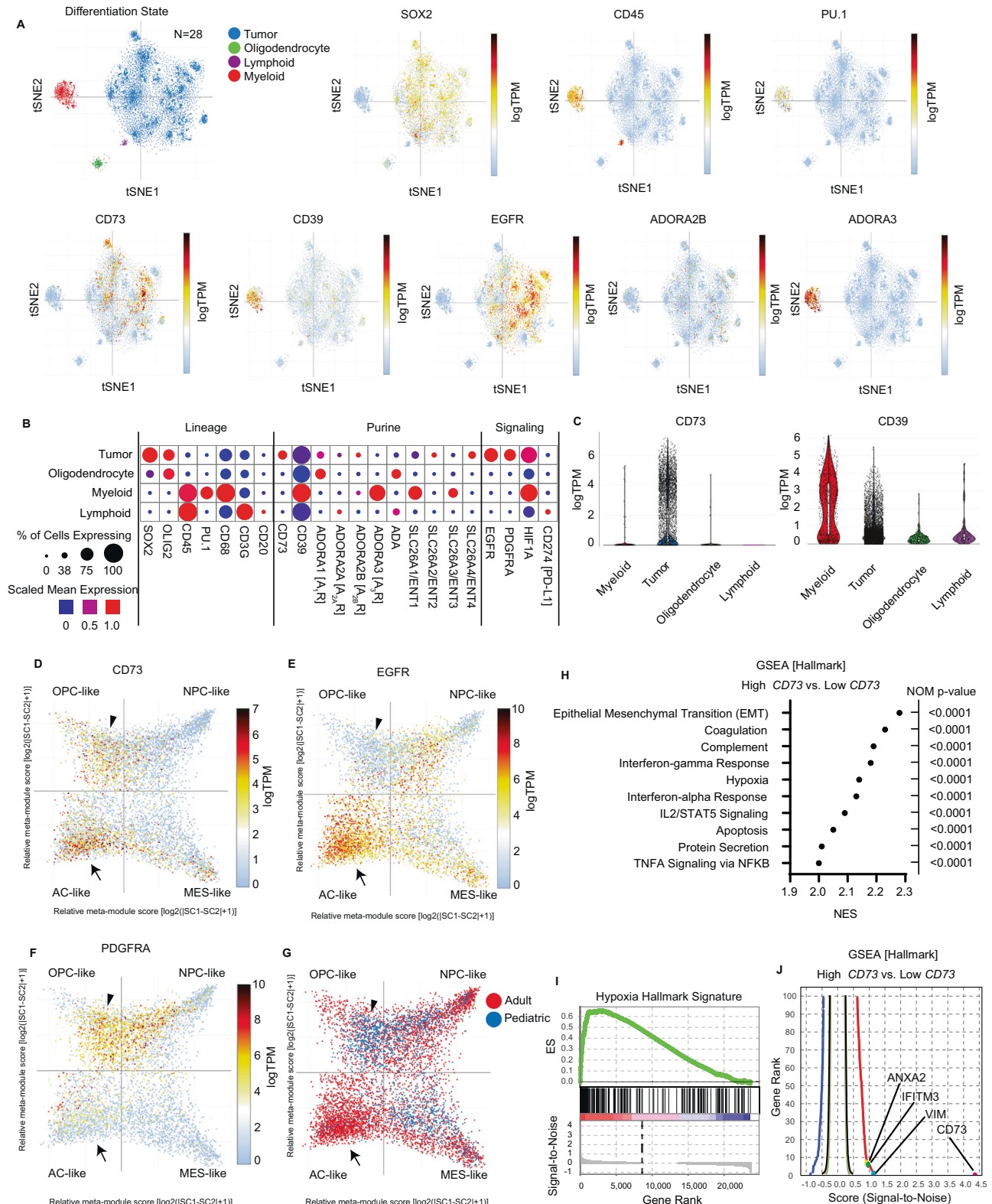

validated the cell-type classification by using distributed stochastic neighbor embedding (t-SNE) to quantify and map the expression of well-established lineage markers. We confirmed the restriction of *SOX2* expression to tumor cells, *PTPRC/CD45* to immune cells, *SPI1/PU.1*, *CD68*, and *CD163* to 'macrophages', *CD3G* to 'T cells', and *OLIG1/2* and *MBP* to oligodendrocytes (Fig. 1A, B; Fig. S1a). In the brain, markers such as *PU.1*, *CD68*, and *CD163* are expressed by multiple myeloid subtypes, including microglia, macrophages, monocytes, and

dendritic cells; we, therefore, re-classified the 'macrophage' category as 'myeloid'. There was minimal expression of the granulocyte-specific chemokine receptors *CXCR1* and *CXCR2* in these cells, suggesting that most myeloid cells are monocytic (Fig. S1a). We found that the 'T cell' category was predominantly composed of *CD3G* positive T cells (85%), but it also included sub-populations of NK cells (*NCAM1/CD56*; 1.1%), and B cells (*MS4A1/CD20*; 3.2%); we therefore relabeled the category as 'lymphoid' (Fig. 1A, B and Fig. S1a).

**Fig. 1 | State-specific expression of core purinergic effectors in glioblastoma.**
**A** Single-cell RNA-sequencing data from 21 adult IDH-wildtype human glioblastomas and 7 pediatric high-grade gliomas encompassing 24,131 cells were analyzed for core purine regulators, lineage-specific markers, and related biomarkers. Cells were clustered as previously described (Neftel et al.) into lineage categories by multi-gene expression signatures using t-distributed stochastic neighbor embedding (t-SNE). **A, B** Lineage differentiation was validated by canonical cell type-specific markers, including *SOX2* (tumor), *PTPRC*/CD45 (pan-immune), *SPI1*/PU.1, *CD68* (myeloid), *OLIG2* (oligodendrocyte) and *CD3G*/CD3, *MS4A1*/CD20 (lymphoid). **A–C** *CD73* was predominantly expressed by tumor cells (35.9% of tumor cells; 1.19 scaled-mean expression), with only rare expression in myeloid (2.5% of cells; 0.06 mean-expression) or other cell types. *CD39* was expressed by all lineages, with higher expression in myeloid cells (98.3% of cells; 2.16 mean-expression). Regarding adenosine receptors, *ADORA1*/A$_1$R was predominantly expressed by oligodendroglia, *ADORA2A*/A$_{2A}$R by lymphoid and tumor cells, *ADORA2B*/A$_{2B}$R by tumor cells, and *ADORA3*/A$_3$R by myeloid cells. Adenosine deaminase (*ADA*) was more strongly expressed by oligodendroglia and lymphoid

cells. Myeloid cells predominantly expressed *ENT1/3*, while tumor cells expressed *ENT2/4*. **D–F** Tumor cells were clustered according to differentiation state (Neftel et al.) into astrocyte-like (AC-like), mesenchymal-like (MES-like), oligodendroglial progenitor cell-like (OPC-like), and neural progenitor cell-like (NPC-like) signatures (meta-modules) using relative expression of each multi-gene signature (relative meta-module score). *CD73* expression was predominantly focused in two clusters: an AC-like cluster associated with *EGFR* expression (arrow), and an OPC-like cluster associated with *PDGFRA* expression (arrowhead), with only scattered NPC-like or MES-like cells expressing *CD73*. **G** *CD73* expressing cells in the AC-like cluster were predominantly derived from adult glioblastomas, while OPC-like cells were predominantly derived from pediatric high-grade gliomas. **H–J** Gene-set enrichment analysis (GSEA) of tumor cells with the highest (top 1%) vs. lowest (1%) *CD73* expression showed multiple significantly enriched pathways (Nominal $p < 0.0001$, GSEA (Kolmogorov-Smirnov test, unpaired, one-sided)) including epithelial-to-mesenchymal transition (EMT), interferon (α/γ) response, and hypoxia most strongly associated with high *CD73* expression, with core genes from each pathway represented among the most enriched genes.

Prior studies of immune populations isolated from glioblastoma by fluorescence-activated cell sorting (FACS) suggested a role for abundant CD73-expressing macrophages in the pathogenesis of glioma[34]. Analysis of scRNA-seq data confirmed that subsets of myeloid cells express *CD73*; however, in this dataset, we found that *CD73* was much more prominently expressed by tumor cells, with transcripts detected in 35.9% of tumor cells, 1.19 mean-expression (log2(TPM/10 + 1)) than by myeloid cells (2.52%; 0.06 mean-expression), or other cell populations (3.2% of oligodendrocytes, 0% of lymphoid cells) (Fig. 1A–C). Conversely, *CD39*, which encodes the upstream ectoenzyme that hydrolyzes extracellular ATP and ADP to AMP, while weakly expressed by multiple populations, including tumor cells (94.8% of tumor cells; 0.73 mean-expression), was much more strongly expressed by myeloid cells (98.3%; 2.16 mean-expression) (Fig. 1A–C).

CD73 is thought to exert its effects principally through the production of adenosine, which can bind to one of four transmembrane adenosine receptors (A$_1$R, A$_{2A}$R, A$_{2B}$R, and A$_3$R) found on multiple cell types. Analysis of adenosine receptor expression showed that tumor cells express each subtype at different levels. *ADORA2B*/A$_{2B}$R, the receptor subtype previously associated with chemoresistance in glioblastoma was detected in 9.0% of tumor cells (0.22 mean-expression)[43]. *ADORA1*/A$_1$R, which may inhibit tumor proliferation[44] was expressed in a distinct subset of predominantly A$_{2B}$R-negative tumor cells (24.4%, 0.45 mean-expression), while *ADORA2A*/A$_{2A}$R was expressed by 5.3% of tumor cells (0.12 mean-expression), and *ADORA3*/A$_3$R by only 0.64% of tumor cells (0.01 mean-expression). Myeloid cells broadly expressed high levels of *ADORA3*/A$_3$R (81.8% of myeloid cells; 4.30 mean-expression), while only small numbers of cells expressed other adenosine receptors. In the CNS, A$_3$R is typically associated with microglia and macrophages with immunomodulatory differentiation[45]. *ADORA2A*/A$_{2A}$R was most strongly expressed by lymphoid cells (7.4%, 0.35 mean-expression), while oligodendrocytes predominantly expressed *ADORA1*/A$_1$R (45.6%, 0.86 mean-expression) (Fig. 1B). The downstream catabolic enzyme *ADA* was predominantly expressed by oligodendrocytes (39.7% of oligodendrocytes, 1.59 mean-expression) and lymphoid cells (22.3% of lymphoid cells, 0.97 mean-expression). The nucleoside transporters *ENT1-4*, which may transport adenosine between the intracellular and extracellular spaces exhibited lineage-dependent expression, with *ENT2* and *ENT4* expressed by subsets of tumor cells (10.7% and 25.8% of tumor cells), while *ENT1* and *ENT3* were expressed by myeloid cells (66.9% and 39.7% of myeloid cells). There was little expression of these transporters in oligodendroglial and lymphoid populations (Fig. 1B).

Single-cell studies have shown that glioblastoma cells differentiate into four principal states mirroring developmental stages or injury-response programs. These states include astrocyte-like (AC-

like), oligodendrocyte progenitor-like (OPC-like), neural progenitor cell-like (NPC-like), and mesenchymal-like (MES-like)[41]. These states are associated with specific underlying genomic alterations, including *EGFR* amplification (AC-like), *PDGFRA* amplification (OPC-like), *CDK4* amplification (NPC-like), and *NF1* loss (MES-like). Analysis of scRNA-seq data from all 28 tumors in the dataset showed that elevated *CD73* expression was principally focused in two discrete clusters of AC-like and OPC-like tumor cells (Fig. 1D), which exhibited high expression of *EGFR* (Fig. 1E) and *PDGFRA* (Fig. 1F), respectively. Fewer tumor cells with strong *CD73* expression were detected in the NPC-like and MES-like clusters. While *CD39* is expressed at low levels by some tumor cells, in this data, expression was not correlated with a specific differentiation state (Fig. S1b). In addition, there was no discernable association between *CD73* expression and the *CDK4* or *NF1* expression characteristic of the NPC-like and MES-like clusters, respectively (Fig. S1c, d). In contrast to the strong expression of *CD73* in specific lineages, few tumor cells had detectable *PD-L1*, with slight enrichment in the AC-like and MES-like states, suggesting that *CD73* may play a more prominent role in immunomodulation in these tumors (Fig. S1e).

Congruent with scRNA-seq data from these 28 adult and pediatric HGG specimens, analysis of non-neoplastic developing human brains (UCSC Cell Browser)[46,47] showed that *CD73* expression is enriched in astrocytes and oligodendrocyte-precursor cells (OPC) (Fig. S2a). *CD39* was expressed by numerous cell types, but most strongly by microglia and vascular endothelial cells (Fig. S2b). Developing astrocytes were enriched for *EGFR* (Fig. S2c), while developing *OPC* were enriched for *PDGFRA* (Fig. S2d). These data suggest that the programs driving *CD73* expression in tumor cells may reflect the retention of lineage-specific transcriptional programs present in non-neoplastic cells.

*EGFR* amplification is common in adult IDH-WT glioblastoma, while *PDGFRA* activation is more common in pediatric tumors, including H3K27M-mutant diffuse midline gliomas. Stratification of tumor cells by differentiation and demographics showed that *EGFR*-hi*CD73*-hi AC-like cells were predominantly derived from adult glioblastoma, while *PDGFRA*-hi*CD73*-hi OPC-like cells were typically associated with pediatric tumors (Fig. 1G). While *H3F3A/B* gene sequencing was not available for all cases, some pediatric tumors strongly associated with OPC-like differentiation and *PDGFRA* expression (e.g., BT749, BT786) had midline and/or caudal anatomic locations (bifrontal, posterior fossa)[41], suggesting that these samples may represent diffuse midline gliomas, the pediatric HGG subtype with the worst overall prognosis[18].

To identify gene programs associated with *CD73* and *CD39* in adult and pediatric HGG, we performed gene-set enrichment analysis (GSEA)[48] on scRNA-seq data[41]. CD73 expression in tumor cells was most strongly associated with inflammatory signaling pathways, including chemokine secretion (CXCL2), major histocompatibility complex class

I (MHC-I) expression, and interferon signaling (Fig. S1f). Comparing tumor cells with the highest (top 1%) and lowest (bottom 1%) *CD73* expression showed significant enrichment of pathways corresponding to epithelial-mesenchymal transition (EMT), coagulation, complement activation, hypoxia, and inflammation, including interferon-α and γ response and TNFA signaling (Fig. 1H, I). As expected, *CD73* was the most enriched gene in this analysis (signal-to-noise = 4.3), with core effectors of each pathway including *VIM* (1.1) (EMT), *IFITM3* (0.9) (IFN-γ), and *ANXA2* (0.9) (hypoxia) among the most enriched genes (Fig. 1J). In myeloid cells, *CD39* expression was strongly associated with purine receptor signaling along with MHC-II expression and interferon signaling (Fig. S1g).

Under conditions of limited oxygen availability, multiple cell types may release purine metabolites[49]. To further explore the significance of hypoxia in *CD73* regulation, we analyzed the expression of hypoxia-responsive genes, including *HILPDA*, which is highly expressed almost exclusively in MES-like cells having a hypoxic transcriptional signature (Fig. S1h)[42]. Interestingly, while *CD73* expression was far less common in the MES-like state than the AC-like or OPC-like states, among MES-like cells with the highest levels of *HILPDA* (log2(TPM/10 +1 ) > 8), a substantial subset (15.8%) were in the top quartile of *CD73* expression (*HILPDA*hi*CD73*hi) with many demonstrating low expression of *EGFR* and *PDGFRA* (Fig. S1i), implicating hypoxia as an additional driver of *CD73* expression. Thus, hypoxia may complement or act independently of cell lineage (e.g., *EGFR* in AC-like and *PDGFRA* in OPC-like populations) in promoting *CD73* expression. Collectively, these data suggest that core purine regulatory genes (*CD39*, *CD73*) are distributed between tumor and myeloid cells, and tumor cell *CD73* expression is associated with lineage-differentiation and functional state.

## CD39 is associated with tumor-associated microglia and myeloid cells in glioblastoma

Myeloid cells typically constitute the largest fraction of the immune cells in gliomas[50]. The myeloid lineage includes numerous distinct functional categories, and the precise cell types and states associated with *CD39* expression are not well-defined. In scRNA-seq data from 28 adult and pediatric HGG samples[41], analysis of lineage-specific markers for microglia and peripheral macrophages showed that most myeloid cells in glioblastoma (70–80%) exhibit a microglial expression signature (e.g., *P2RY12*, *TMEM119*, *SLC2A5*, and *CX3CR1* genes) rather than a peripheral macrophage signature (e.g., *CCR2*, *CLEC12A*, *F10*; 20–30% of cells) (Fig. 2A, B). *CD39* expression was enriched in microglia as a whole and nearly all cells with the highest *CD39* expression levels were microglia (Fig. 2A–C; Fig. 2C shows gene expression of the top-100 *CD39* expressing cells).

To study immune states associated with purine signaling, we evaluated an additional scRNA-seq dataset from a recently published database of 7 newly-diagnosed and 4 recurrent/residual glioblastomas comprising 21,303 and 42,870 FACS-sorted CD45-positive immune cells, respectively[51]. Similar to our prior analyses, cells were clustered by multi-gene lineage signatures and represented by t-SNE (Fig. 2D). In newly-diagnosed tumors, *CD73* expression was detectable in very few immune cells (0.5% of cells) and these were predominantly myeloid cells (Fig. 2E, F). *CD39* expression was more widespread, being present in a subset of cells of all types (39% in total), although the strongest and most frequent *CD39* expressors were tumor-associated microglia, dendritic cells (DC) and other myeloid cells (in addition to some *CD39*hi lymphoid cells; (Fig. 2G, H). Recurrent tumors had slightly more immune cells with detectable *CD73* transcripts (0.7% vs. 0.5%) with increased numbers of *CD73* expressing B and T cells. As in newly-diagnosed tumors, *CD39* was predominantly expressed by myeloid populations, albeit with a greater proportion of peripheral macrophages to microglia, with the latter still comprising most of the highest expressing cells. In recurrent tumors, the overall proportion of *CD39*-

expressing immune cells was lower (27% vs. 39%), due to an increase in the population of *CD39*-negative T, B, and plasma cells; however, there was an increase in the number of *CD39*-expressing regulatory T cells (Treg) (Fig. S3a–e).

Tumor-associated myeloid (TAM) populations include numerous sub-groups of monocytic and microglial populations identifiable by multi-gene signatures (Fig. 2I)[51]. *CD73* expression was rare in myeloid cells (0.4% of cells), with the few weakly *CD73* expressing cells mainly corresponding to hypoxic and transitory monocytes (Fig. 2J, K, Clusters-9 and 10, inset). *CD39* was expressed by all myeloid sub-groups (45% of myeloid cells), with enrichment of *CD39*hi cells in the microglial cluster (Fig. 2L, M, Cluster-1). *CD73* expression was slightly increased in recurrent tumors (0.7% vs. 0.4% of myeloid cells), with expression predominantly in hypoxic monocytic populations and a slightly larger subgroup of infiltrating monocytes. *CD39* showed a similar pattern of myeloid expression in recurrent tumors (44.5% of myeloid cells), with similar enrichment in microglia (Fig. S3f–j). *CD73* expression was not detected in dendritic cells, whereas *CD39* was broadly expressed by multiple dendritic sub-groups (41.9% of dendritic cells). Expression was more common in classical dendritic cells, and less common in plasmacytoid dendritic cells (pDC) and migratory DC (migDC) (Fig. S3k–m). Further analysis of scRNA-seq data derived from a third independent cohort of microglia derived from 15 human brain specimens[50] confirmed these findings, with minimal *CD73* expression in microglial populations, and *CD39* expression across subgroups (Fig. S3n–p). Collectively, these data confirm that *CD73* expression is rare in glioma-associated immune cells, which likely provide only a minor contribution to overall CD73 activity compared to tumor cells. However, *CD39* is expressed by a wide-variety of tumor-associated immune cells, with microglia exhibiting the highest levels of expression, though recurrent tumors demonstrate shifts in the immune microenvironment and a greater influx of peripheral macrophages and lymphoid cells including CD39-expressing regulatory T cells.

## CD73 is associated with CD39 and inflammatory signatures in glioblastoma

In non-neoplastic tissues, CD73 often serves a homeostatic function by suppressing ongoing inflammatory reactions in damaged tissue[52]. However, the precise inflammatory state of glioblastoma and its relationship to heterogeneous cell populations and purinergic pathway enzyme expression is not well-defined. To further analyze the overall landscape of immune cells and inflammatory signaling between *CD73*hi and *CD73*lo adult glioblastoma, we analyzed mRNA-sequencing data from 168 tumors in The Cancer Genome Atlas (TCGA) database[53]. We used this data to estimate the population abundance of tumor and tumor-infiltrating immune populations using gene expression signatures derived from human glioblastoma scRNA-seq datasets[41,50,51] (Fig. 3A; Supplementary Data 1, 2, see "Methods"). We found that TAM, monocyte, pre-dendritic cell (DC), migratory DC, pDC, and T-cell populations were enriched in *CD73*hi versus *CD73*lo glioblastoma (Fig. 3B), as were inflammatory signatures for IFNγ and MHC-I signaling and signatures of Immunoscore, T cell inflammation, and cytolytic activity (CYT) (Fig. 3C). Glioblastoma with combined elevation of *CD73* and *CD39* (i.e., *CD73*hi/*CD39*hi tumors) were enriched for immune populations, while those with low levels of both enzymes (*CD73*loC-D39lo) were depleted for nearly all immune cell populations. Notably, 'double high' tumors had even more significant elevation of immune cell types and inflammatory signatures than tumors elevated for either *CD73* or *CD39* alone (Fig. 3D). Given the immunosuppressive functions of CD73 and adenosine, these data suggest that *CD73* and *CD39* expression may be upregulated in tumors in response to ongoing inflammation similar to normal tissues, or that inflammation drives selection for *CD73*hi tumor cells and *CD39*hi immune cells that may support ongoing tumorigenesis.

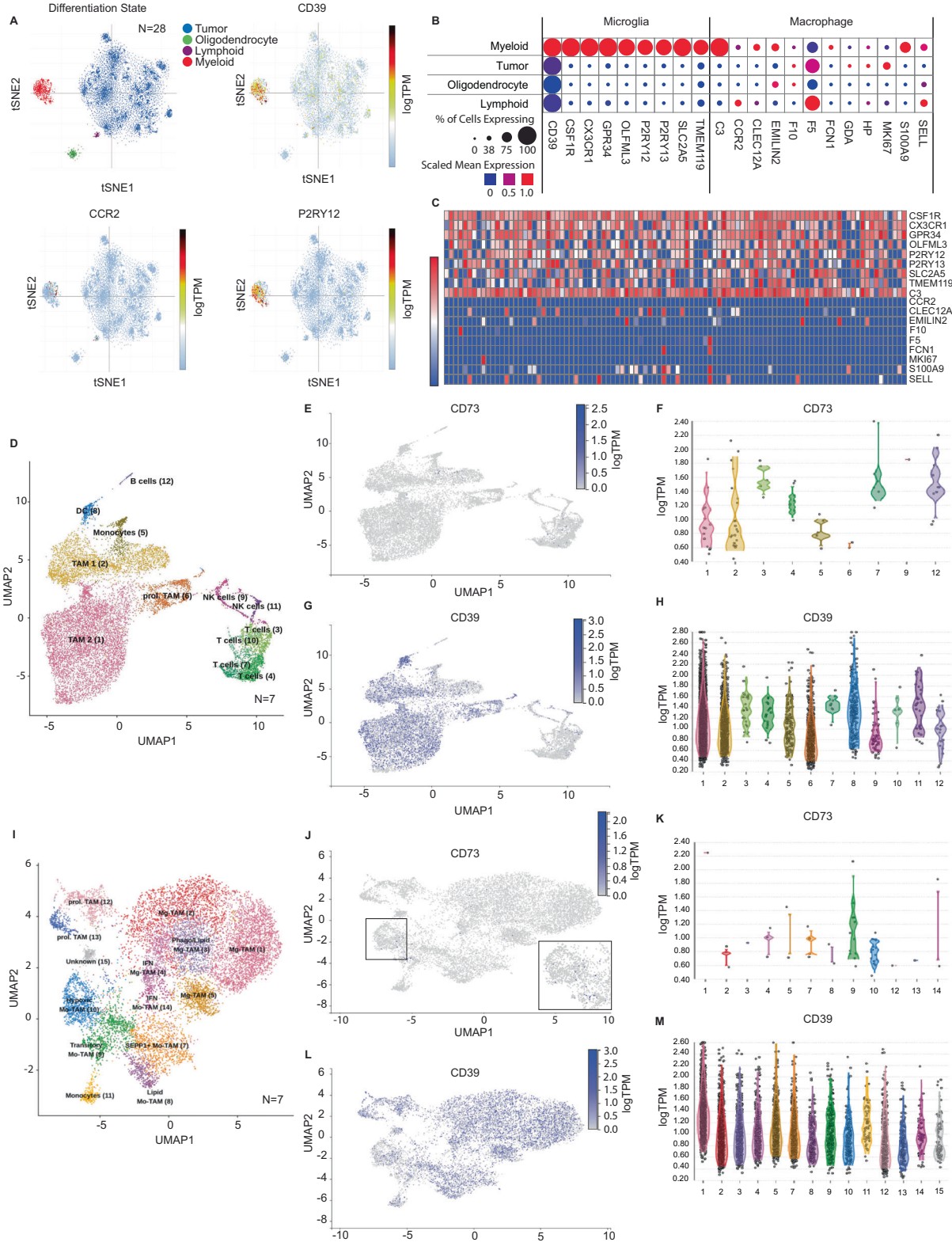

## CD73 protein exhibits distinctive expression patterns in glioblastoma and other central nervous system tumors

To assess the relative significance of CD73 expression and purine activity in glioblastoma as compared to other CNS and non-CNS tumors, we first analyzed bulk mRNA sequencing data from TCGA (32 tumor types in the Pan-Cancer Atlas, n = 10,967 samples). Compared to most other tumor subtypes, glioblastomas exhibited significantly higher levels of *CD73* (Fig. 4A and Supplementary Data 3). Median *CD73*

expression in glioblastoma was similar to that of non-small-cell lung cancer (NSCLC), pancreatic ductal carcinoma, and colorectal carcinoma, tumor types in which anti-CD73 antibody therapy has shown promise in early-stage clinical trials[54,55]. Of note, elevated *CD73* expression does not appear to be due to genomic alterations in the *CD73* gene; mutations, chromosomal copy-number changes, and rearrangements in *CD73* are uncommon in glioblastoma (<0.5%) and other tumor types (Fig. S4a). Moreover, *CD73* mRNA levels are not

**Fig. 2 | CD39 is associated with tumor-associated microglia and myeloid cells in glioblastoma. A, B** Analysis of single-cell RNA-sequencing data from 21 adult IDH-wildtype human glioblastomas and 7 pediatric high-grade gliomas (24,131 cells) showed that most myeloid cells expressed genes associated with microglia (e.g., *P2RY12/13, CX3CR1, TMEM119,* and *SLC2A5)*, rather than peripherally-derived macrophages (e.g., *CCR2, F10, and CLEC12A). **C** Myeloid cells with the highest *CD39* expression (top-100 cells shown) demonstrated microglial gene expression signatures. **D** Single-cell RNA-sequencing data from FACS-sorted CD45-positive immune cells from a second cohort of 7 newly-diagnosed and 4 recurrent/residual adult glioblastomas (21,303 and 42,870 cells, respectively) (Pombo Antunes et al.), were clustered using uniform manifold approximation and projection (UMAP) by cell lineage (TAM = tumor-associated macrophage; TAM1 = peripheral macrophage, TAM2 = microglia, prol; TAM = proliferative TAM; DC = dendritic cell). **E, F** These data showed only rare *CD73*-expressing immune cells (0.5%), with the greatest number of myeloid cells (groups 1,2). **G, H** *CD39* exhibited broad expression in immune populations, with the greatest enrichment in TAM2 (microglia) and TAM1 (peripheral macrophages), and dendritic cells (groups 1,2,8). **I** Tumor-associated myeloid cells were then clustered by functional sub-group according to gene expression (Mo-TAM = monocytic TAM; Mg-TAM = microglial TAM; prol. TAM = proliferative TAM). **J, K** *CD73* expression was rare in tumor-associated myeloid cells (0.4% of cells) but showed the greatest relative abundance in hypoxic and transitory TAM (groups 9,10). **L, M** *CD39* expression was present in all TAM sub-groups, but most strongly enriched in microglial TAM (group 1). (lineage clusters without any detected cells are not represented in **F, H, K, M**).

significantly elevated in the small subset of cancers having *CD73* gain or amplification as compared to those diploid for *CD73* (Fig. S4b). Glioblastomas also exhibited higher levels of *CD39* expression than many other tumor types (Fig. S5a). Analysis of adenosine receptors showed that expression of ADORA2A/A$_{2A}$R and ADORA2B/A$_{2B}$R in glioblastoma and diffuse glioma were moderate compared to that of other cancers, while ADORA1/A$_1$R and ADORA3/A$_3$R were expressed at significantly higher levels in glioblastoma compared to all other tumor types (Fig. S5b–d and Supplementary Data 4).

To confirm the presence of elevated CD73 protein expression in CNS tumors, we used immunohistochemistry (IHC) to systematically characterize CD73 levels in tissue sections from 605 human brain tumors of most major histologic subtypes obtained from the BWH archives and the Children's Brain Tumor Network (CBTN) (Fig. 4B–D; Supplementary Data 5, antibodies used in this study are listed in Supplementary Table 1). This cohort includes 194 adult glioblastoma specimens from 179 distinct patients, including 128 primary and 66 recurrent tumors, and 39 pediatric high-grade gliomas. IHC revealed moderate-to-strong CD73 expression in IDH-wildtype and IDH-mutant adult glioblastomas, H3K27M-mutant diffuse midline gliomas, and pediatric HGG, as well as oligodendrogliomas, pilocytic astrocytomas, meningiomas, and papillary craniopharyngiomas (Fig. 4B–D; Fig. S6; Supplementary Data 6–10). In adamantinomatous craniopharyngioma (ACP), tumor cells were CD73 negative except along cyst lining epithelium, a pattern reminiscent of PD-L1, which is upregulated by inflammatory mediators present in the cyst fluid[56,57]. Peri-tumoral stromal cells also showed widespread CD73 expression in ACP, suggesting that some tumors may exhibit regional or predominantly stromal CD73 expression (Fig. S6c; Supplementary Data 7). Thus, gliomas exhibit high CD73 expression relative to both CNS tumors and other cancers more broadly, indicating that purinergic signaling may play an important role in glioma pathogenesis.

## CD73 expression correlates with oncogenic signaling and clinical outcome in glioblastoma

Focusing on 58 molecularly-profiled, IDH-wildtype adult glioblastomas, we searched for clinical or molecular correlations with CD73 expression. Each case had data from a 447-gene targeting exome sequencing assay, chromosomal copy-number analysis, and O6-methylguanine-DNA methyltransferase (*MGMT*) promoter analysis; *MGMT* silencing is predictive of glioblastoma sensitivity to alkylating chemotherapy[58]. Polysomy of chromosome 7 and *EGFR* amplification on 7p are highly recurrent events in IDH-wildtype glioblastoma and were present in 66 and 48% of tumors, respectively (Fig. 5A). Analysis of all common sequence (447 genes) and chromosomal (genome-wide) variants was notable for a significant positive correlation between CD73 protein expression (by IHC) and *EGFR* amplification ($p = 0.029$) (Fig. 5B), consistent with the link we identified between *CD73* expression and the *EGFR*-associated AC-like state in scRNA-seq data from adult glioblastoma as well as reports of EGFR-mutated NSCLC expressing higher levels of CD73 compared with EGFR wild-type

tumors[59,60]. In a large independent cohort of glioblastoma from the TCGA, we also found a statistically significant positive correlation between *EGFR* and *CD73* expression ($p = 0.039$). The correlation was relatively weak ($r = 0.17$), likely due to bulk measurement of mRNA levels derived from mixed tumor, immune, and other cells (Fig. 5C), but this was the only positive correlation observed amongst relevant lineage-encoded drivers of glioblastoma (Fig. S7a–f). We found no significant association between bulk *CD73* and *CD39* mRNA levels in the TCGA dataset ($n = 145$) (Fig. S7g), indicating that while coordinate expression of *CD73* and *CD39* correlates with greater changes to the inflammatory microenvironment, these enzymes, which are expressed by multiple cell populations, may not necessarily be co-regulated.

In the 194 tumor cohort, CD73 expression was slightly lower in recurrent/residual tumors as compared to primary lesions (Fig. 5D; Supplementary Data 9) and in IDH-WT glioblastoma with an unmethylated (unsilenced) *MGMT* promoter compared to ones with *MGMT* promoter methylation (Fig. S7h and Supplementary Data 10). There was no notable association between *CD73* expression and hypermutation (>10 mutations/MB) or mismatch repair (MMR)-deficiency (Fig. S7i). Survival analysis based on semi-quantitative scoring of CD73 expression in tumor cells by IHC showed significantly reduced progression-free survival (Mantel–Cox; $p = 0.048$) in IDH-WT glioblastoma patients ($n = 58$) that also had high CD73 (score ≥2) as compared to low CD73 protein expression (score ≤1) (Fig. 5E). This result is consistent with the association between reduced progression-free survival and elevated *CD73* mRNA expression in glioblastoma as evaluated by bulk mRNA expression profiling[40,61]. However, IHC provides a more specific measure of CD73 expression in tumor cells and is better suited for use in clinical practice for patient stratification and trial enrollment.

In addition to the link identified between *CD73* and *EGFR*-amplified AC-like adult glioblastoma, scRNA-seq showed that *CD73* expression may be independently associated with OPC-like high-grade gliomas in pediatric patients, some of whose tumors also exhibit elevated *PDGFRA* expression. To confirm these findings and better understand this pediatric subgroup, we analyzed CD73 mRNA and protein expression in a cohort of 102 pediatric brain tumor specimens (from 69 patients) obtained from the Children's Brain Tumor Tissue Consortium (CBTTC)[62]. In keeping with our prior survey, we found that pediatric medulloblastomas and ependymomas exhibited minimal *CD73* expression, while numerous high- and low-grade gliomas, gangliogliomas, and a subset of craniopharyngiomas exhibited high *CD73* levels (Fig. 5F). Across pediatric tumors we found a significant positive correlation between CD73 and PDGFRA at both mRNA and protein levels (Fig. 5G, H). There was also a significant positive correlation between *CD73* and *CD39* levels in pediatric tumors (Fig. 5I, J). In contrast to adult HGG, there was a negative correlation between *CD73* and *EGFR* expression in this population (Fig. 5K).

We further analyzed 34 pediatric HGG for which both clinical and molecular data were available. This showed that mutations in *H3F3A* (36%) and *TP53* (45%) were the most common, while *EGFR*

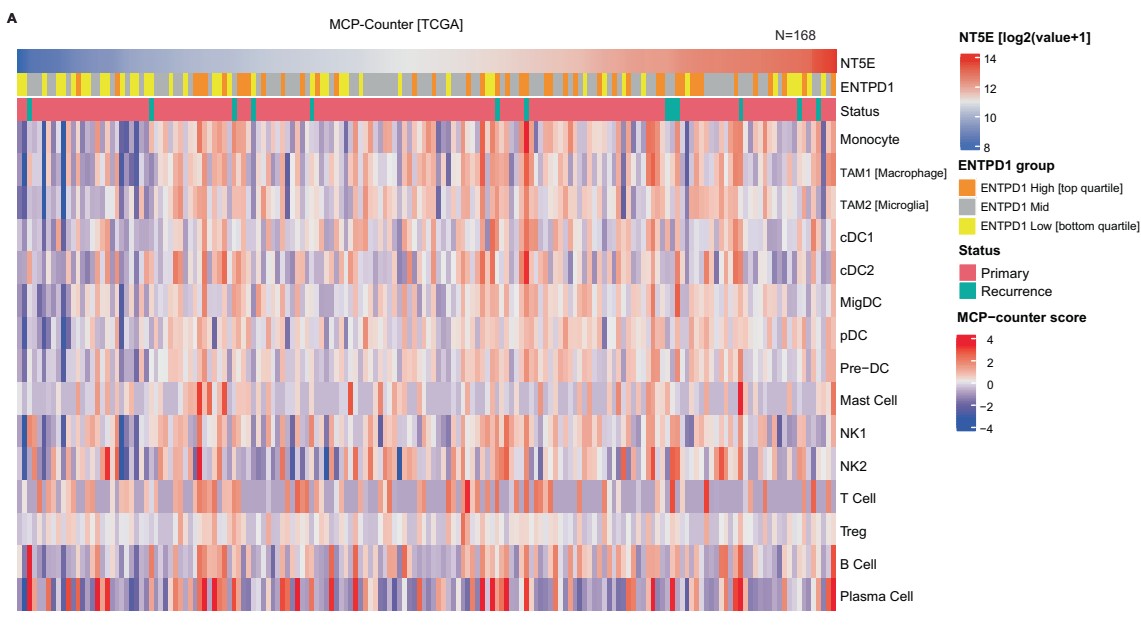

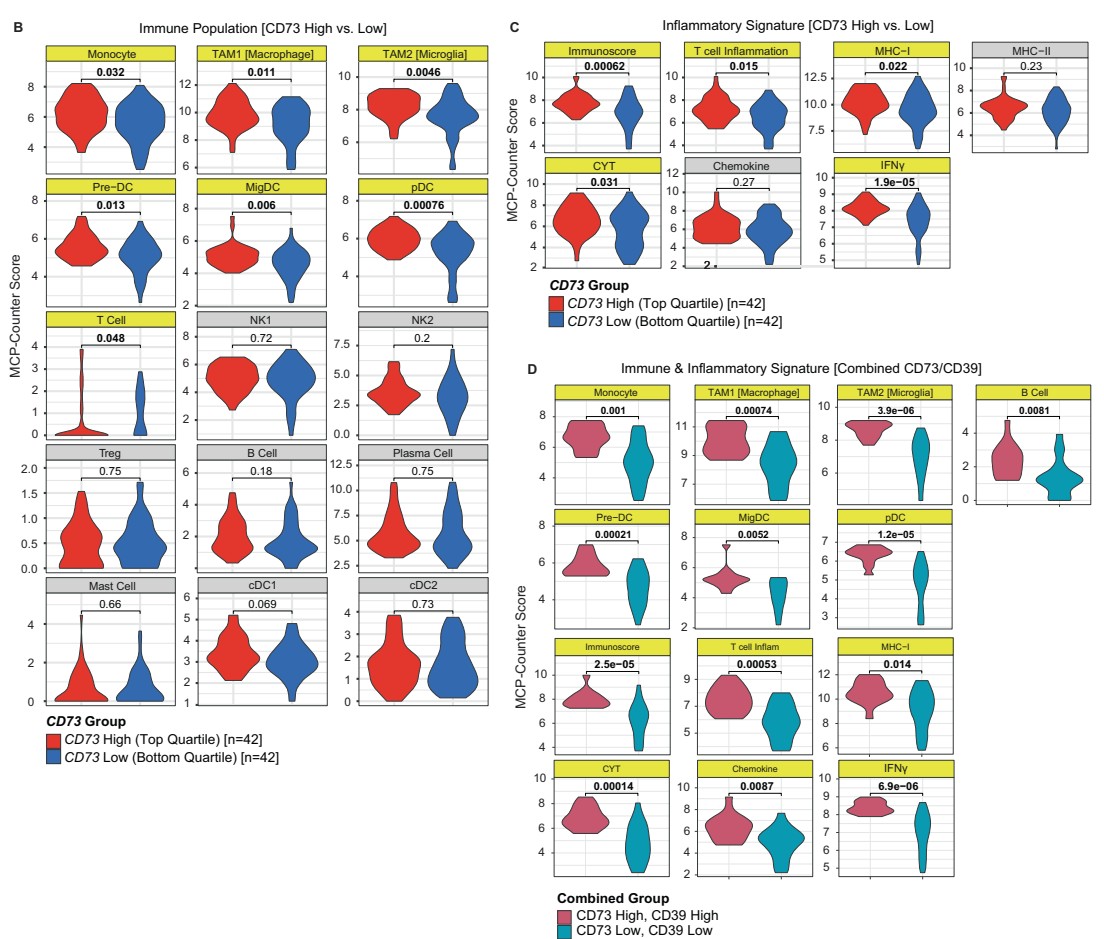

amplification was present in only one tumor (Fig. S8a). IHC performed on the same set of samples showed high level (IHC score ≥2) CD73 expression in 59% (20/34) of pediatric HGG cases. IHC score showed a positive trend with *CD73* mRNA expression in all HGG cases with bulk sequencing from the larger CBTTC cohort (*n* = 39), though this did not reach statistical significance (*p* = 0.052; *n* = 39) (Fig. S8b). CD73 IHC score was strongly correlated with CD73 protein levels measured by mass spectrometry (*p* = 0.008; *n* = 22) (Fig. S8c), and in tumors with

both measurements CD73 protein and bulk mRNA levels were strongly correlated (*p* = 2.62E−7; *n* = 19) (Fig. S8d). There was a negative correlation between *CD73* and *EGFR* bulk mRNA levels in the full 102 tumor CBTTC cohort (Fig. S8e), in keeping with the relative absence of gene amplification events. These data support the conclusion that *CD73* regulation in pediatric HGG is biologically distinct from adult HGG. Most H3K27M-mutant diffuse midline gliomas exhibited high CD73 expression (IHC ≥ 2), though CD73 levels (as measured by IHC or bulk

**Fig. 3 | *CD73* and *CD39* are associated with inflammatory signatures in glioblastoma. A** To better understand the inflammatory landscape of glioblastoma, bulk mRNA-sequencing data from 168 adult glioblastomas (TCGA) were deconvolved by Microenvironment Cell Population-Counter (MCP-Counter) to estimate the relative abundance of tumor and immune populations and correlate with purine regulatory enzymes and inflammatory signaling signatures. Analysis of all glioblastoma cases showed a positive correlation between *CD73* and *CD39* expression, and trend of greater numbers of most immune populations in *CD73*-high samples. **B** Comparison of cases with high *CD73* expression (top-quartile, $n = 42$), and those with low expression (bottom-quartile, $n = 42$), showed that high levels were significantly ($p < 0.05$, Wilcoxon test, unpaired, two-sided) associated with a greater relative density of myeloid (naïve monocyte, TAM1 (macrophage), TAM2 (microglial)), dendritic (plasmacytoid dendritic cells (pDC) and pre-dendritic cells (pre-DC)), and T-cell populations, with no significant difference in NK, B cell, T

regulatory cells, plasma cells, classical dendritic cell (cDC1/2), or mast cell populations. **C** Bulk analysis of gene signatures associated with inflammatory signaling showed that tumors with high *CD73* expression exhibited significantly higher inflammation, including elevation of signatures associated with interferon-γ, MHC-I, Immunoscore, cytolytic (CYT), and T cell inflammatory pathways ($p < 0.05$, Wilcoxon test, unpaired, two-sided). **D** We noted that cases with coordinate expression of *CD73* and *CD39* showed particularly pronounced changes in immune populations and signaling. Direct comparison of *CD73*[hi]*CD39*[hi] tumors with *CD73*[lo]*CD39*[lo] tumors showed an even more strongly significant relative enrichment of myeloid (monocyte, TAM1, TAM2), dendritic (pre-DC, migDC, pDC), and B cell populations in *CD73*[hi]*CD39*[hi] tumors ($p < 0.05$, Wilcoxon test, unpaired, two-sided), as well as greater elevation of inflammatory signaling (T cell, MHC-I, IFN-γ, and chemokine) ($p < 0.05$, Wilcoxon test, unpaired, two-sided).

mRNA sequencing) were not significantly different than non-H3F3A mutated pediatric HGG, which also showed strong expression (Fig. S8f, g). Interestingly, tumors expressing *H3F3A*-G34R mutations exhibited significantly lower *CD73* expression as compared to *H3F3A*-K27M mutants ($p = 0.04$) (Fig. S8g). Unlike adult HGG, survival analysis of non-*H3F3A* mutated HGG revealed no significant difference in overall survival (OS) between cases with high and low expression by CD73 IHC ($p = 0.85$) (Fig. 5L) or mRNA expression ($p = 0.79$) (Fig. S8h) in this small cohort. Given the high levels of *CD73* in most H3K27M-mutant tumors, there were insufficient cases with low expression in this cohort for a survival comparison between high and low expressing cases.

### CD73 is associated with increased adenosine levels in glioblastoma

Although tumor CD73 is thought to exert its effects through the elevation of extracellular adenosine levels in the tumor microenvironment, adenosine levels have not yet been measured directly in tumor tissues or correlated to CD73 levels. We therefore used spatially resolved mass spectrometry imaging (MSI) of tissue sections to quantify metabolite levels from multiple metabolic pathways. Analysis of ion-maps from tissue sections of 9 fresh-frozen glioblastomas and non-neoplastic brain (see "Methods") showed that purine metabolism was the second most significantly enriched metabolic pathway ($p < 0.001$) in tumor tissue relative to non-neoplastic brain tissue[63,64]. This included metabolites in the immunoregulatory adenine purine pathway (ATP, ADP, AMP, and adenosine). Other enriched pathways included amino acids (aspartate, glutamine, arginine, proline), glutathione and phospholipid metabolism, amino sugar, and nucleotide sugar metabolism (Fig. 6A). IHC on these tumor specimens showed that four had high and five had low CD73 expression levels (Fig. 6B). Comparison of whole-tissue purine metabolite levels from these samples demonstrated that adenosine was the most highly enriched purine metabolite in tumors with high CD73 expression relative to those with low CD73 expression (3.5-fold difference, $p = 0.04$) (Fig. 6C and Supplementary Data 11). These data show that expression of CD73, which is observed primarily in tumor cells, is associated with increased adenosine levels in glioblastoma tissue sections. A caveat in this finding is that MSI cannot distinguish intracellular and extracellular adenosine. However, we hypothesize that differences in the levels of CD73, an extracellular enzyme, are at least partly responsible for the observed differences in adenosine levels.

To further explore the relationship between CD73 protein and adenosine levels in CNS tissue in the absence of cancer, we performed CD73 immunofluorescence (IF) and MSI on serial coronal sections from normal mouse brain. The brain had heterogeneous CD73 expression with the highest levels found in the basal ganglia (Fig. 6D, arrows), and weaker expression in cortical gray-matter regions and minimal expression in white matter tracts. Visual inspection (Fig. 6D, E and Fig. S9a, b) and computational analysis (Fig. 6F–H and Fig. S9c) of

registered regions showed a strong correlation ($p = 0.001$) between CD73 and tissue adenosine levels independent of cell density (Fig. 6I) The observed spatial correlation between adenosine and CD73 levels supports the existence of a causal link between the two.

### Multiplexed single cell analysis of glioblastoma tissue

To better understand the single cell states associated with purinergic signaling in glioblastoma, we next used multiplexed immunofluorescence to define and map diverse cell populations in human tumor tissues. We first used pixel-by-pixel analysis[65] to establish the specificity of anti-CD73 antibodies in the context of multiplexed tissue imaging and observed a strong correlation in staining for four different anti-CD73 antibodies (Fig. S10a) as well as strong concordance with IHC results (Fig. S10b). We selected the best-performing CD73 antibody clone (see Methods) and performed 36-marker tissue-based cyclic immunofluorescence (CyCIF)[9] on 194 glioblastoma cases, including 139 adult glioblastoma specimens for which clinical and/or genetic data were available (Fig. 7A and Fig. S11) as well as an additional set of 14 oligodendroglioma and 15 anaplastic oligodendrogliomas. From this cohort, we focused primarily on IDH-wildtype glioblastoma, which accounts for approximately 90% of adult glioblastoma diagnoses, for further single-cell analysis ($n = 172$ tumors). We segmented individual cells as previously described and quantified fluorescence intensities on a per-cell basis[66] generating marker expression data for 940,891 cells followed by assignment of cell-types via Gaussian Mixture Modeling (GMM) (Fig. 7b)[63]. Examination of lineage-specific marker expression revealed four principal clusters corresponding to tumor and glia (using the SOX2, OLIG2, GFAP markers), lymphoid cells (CD45, CD3, CD8, and CD4), myeloid cells (CD45, PU.1, CD163, CD68, and CD11b), and endothelial cells (CD31) (Fig. 7B). In most samples myeloid cells were the most abundant immune cells (Fig. 7C) and were up to >20 fold more common than lymphoid cells (Fig. S10c), consistent with other results showing that myeloid cells represent the predominant immune population in glioblastoma.

Consistent with findings from scRNA-seq data, CyCIF showed that CD73 was expressed at the highest levels on tumor cells in glioblastoma specimens (Fig. 7A, B). Expression was also detected on a subset of myeloid cells and when these cells were present, CD73 levels were similar to those on tumor cells (Fig. 7B and Fig. S10d)[34] Strong staining for CD39 was most commonly observed on myeloid and endothelial cells though as with scRNA-seq data tumor cells also expressed CD39 at lower levels (Fig. 7A, B; Fig. S10e). EGFR and GFAP were predominantly expressed by tumor cells, often at varying levels, with expression of both proteins highest in *EGFR*-amplified tumors (Fig. S10f, g), consistent with the AC-like differentiation of these tumors. Analysis of CyCIF data revealed a strong positive correlation between EGFR and CD73 expression at a single cell level in tumor cells (Fig. 7D). This correlation in protein levels was also observed in tumor cells from cases lacking activating *EGFR* amplifications. Thus, tumors without evidence of alterations at the *EGFR* locus may still harbor sub-

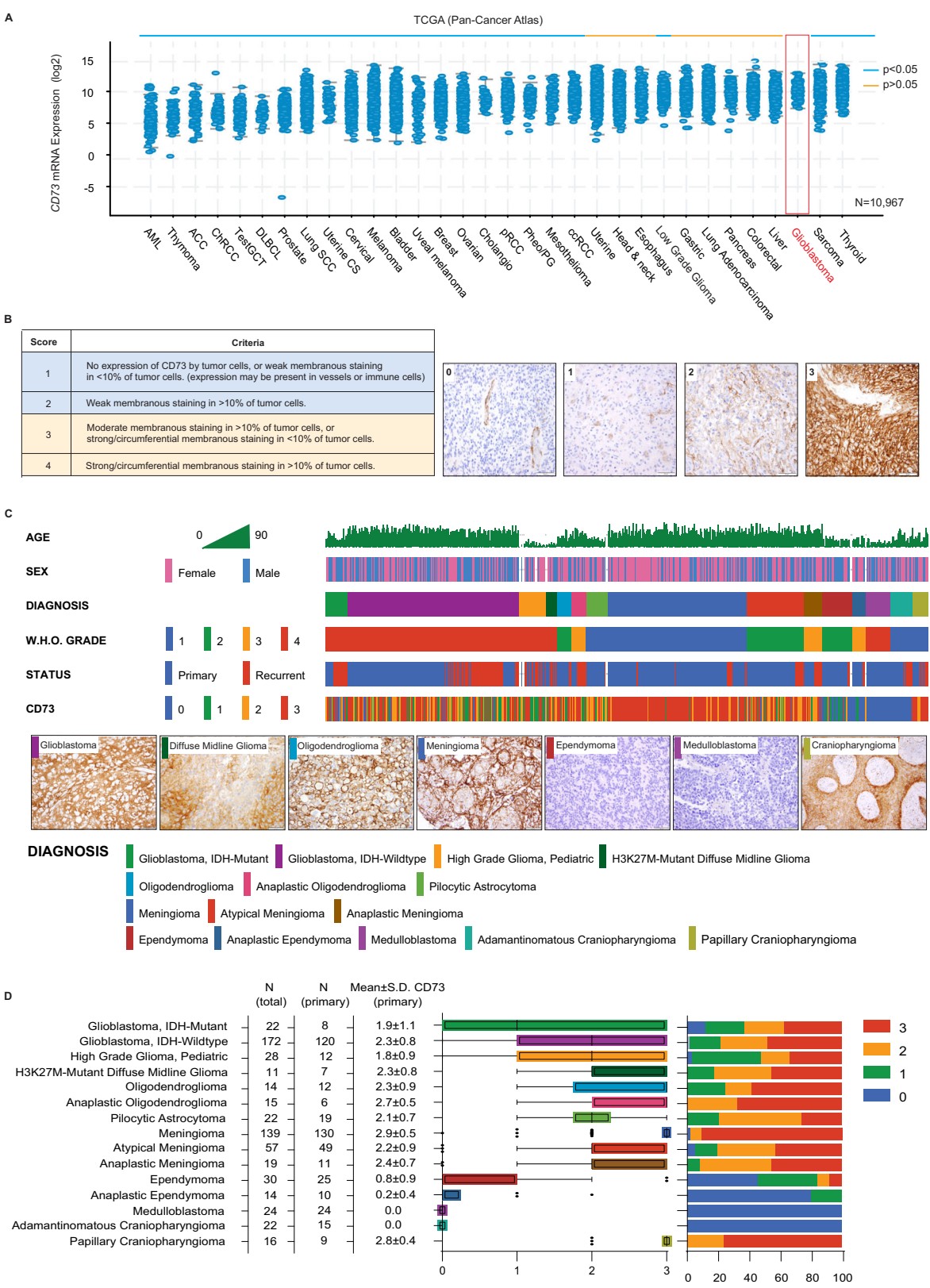

populations of tumor cells expressing both EGFR and CD73. CD73 levels were also correlated with increased cell division, as judged by levels of the proliferation marker Ki-67 (Fig. S10h) which is a potential explanation for the observation that cases with high mean tumor cell CD73 expression exhibited significantly shorter progression-free survival ($p = 0.016$) (Fig. 7E).

We also noted a positive correlation between levels of CD73 and HIF1α (Fig. S10i) and thus with hypoxia. Using microregional spatial transcriptomic data from the IVY Glioblastoma Atlas Project (IVY-GAP) resource[67], we compared transcriptomic data from 7 anatomically distinct tumor regions from 10 adult glioblastomas, including regions of palisading necrosis and peri-necrotic tumor. *CD73* levels were higher

**Fig. 4 | CD73 exhibits distinctive expression patterns in glioblastoma and other central nervous system tumors. A** Bulk *CD73* mRNA expression was assessed in tumor tissue ($n = 10,967$) from 32 cancer subtypes (TCGA). Glioblastoma (red box) exhibited significantly higher *CD73* expression ($p < 0.05$, blue bar, t-test, unpaired, two-sided, with Welch's correction; see Supplementary Data 3 for precise values) relative to most tumor sub-types. **B** CD73 immunohistochemistry was performed on a cohort of 605 independent human CNS tumors from the BWH archives and Children's Brain Tumor Tissue Consortium (CBTTC), including each major subtype and multiple tumors which have not been systematically evaluated for CD73 expression previously (oligodendroglioma, meningioma, ependymoma, craniopharyngioma, medulloblastoma, pediatric glioma). Each tumor in the larger cohort was stained once. Following qualitative evaluation of the specimens by a pathologist, semi-quantitative (0–3) immunohistochemistry (IHC) index scoring criteria were defined to characterize and compare membranous CD73 protein expression in these tumors (representative staining for each score in glioblastoma specimens are depicted in the right-panels); scale bars 50 μm. **C** This scoring system was then applied to the full cohort of 604 CNS tumors, demonstrating distinct expression patterns in each histologic subtype (representative staining from cases of major subtypes are demonstrated); scale bars 50 μm (except for craniopharyngioma 100μm). **D** IDH-mutant [$n = 22$] and IDH-wildtype [$n = 172$] adult glioblastomas, H3K27M-mutant diffuse midline glioma [$n = 11$] and pediatric high-grade glioma [$n = 28$], pilocytic astrocytomas [$n = 22$], and oligodendrogliomas [$n = 29$] frequently exhibited strong (IHC ≥ 2) CD73 expression. Medulloblastomas of all histologic and molecular subtypes [$n = 24$] and ependymomas [$n = 44$], exhibited minimal CD73 expression. Adamantinomatous craniopharyngiomas exhibited no CD73 expression in the tumor epithelium but consistently strong expression in peri-tumoral fibrovascular stroma [$n = 22$], while papillary craniopharyngiomas exhibited strong epithelial/tumor CD73 expression in all cases [$n = 16$]; middle panel uses a Tukey box-and-whisker plot with midline = median, box limits = Q1 (25th percentile)/Q3 (75th percentile), whiskers = 1.5 inter-quartile range (IQR), dots = outliers (>1.5IQR).

in peri-necrotic and palisading tumor regions as compared to cellular tumor regions. Peri-necrotic and palisading tumor regions also exhibited elevated expression of hypoxia-response genes such as *HILPDA, LDHA, ENO2*, and *DDIT3* relative to other regions (Fig. 7F). In situ hybridization (ISH) data from IVY-GAP confirmed these findings, showing elevated *CD73* expression in peri-necrotic cells (Fig. 7G), as well as elevated expression of hypoxia-response elements such as *LDHA* (Fig. 7H). Pairwise comparison of anatomic regions showed that compared to cellular tumor regions, *CD73* is significantly elevated in peri-necrotic ($p = 0.0027$) and palisading ($p = 1.2e−5$) tumor regions. In contrast, *CD39* expression was detected in all tumor regions, with enrichment in regions of microvascular proliferation and hyperplastic vessels (Fig. 7F). CyCIF data also supported that *CD39* expression is broadly distributed across the tumor with elevated expression in regions rich in vascular endothelium (Fig. 7A, B). Thus, hypoxia may promote local upregulation of *CD73* in restricted neighborhoods.

## Spatial patterning of core purine regulators in adult glioblastoma and pediatric high-grade glioma

The compartmentalization of CD39 and CD73 into myeloid and tumor cells (as revealed by scRNA-seq and CyCIF data) in both adult glioblastoma and pediatric HGG suggested that purinergic metabolism and signaling might be modulated by the relative locations of these two cell types with consequent effects on immune activity and tumorigenesis. To test these ideas, we first determined the spatial distribution of different cell types in IDH-wildtype glioblastoma from the CyCIF data ($n = 172$) and observed a non-random distribution of cell types, including collections of tumor and myeloid cells (Fig. 8A). To investigate this relationship, we performed high-resolution 3D CyCIF of an adult glioblastoma sample followed by mapping of marker expression and co-localization analysis (Fig. 8B, C). This involved 18-plex imaging, optical sectioning with high-numerical-aperture objectives followed by image deconvolution[68]. These data revealed a close spatial association between CD39 protein expressed by myeloid cells (cells that scored as PU.1+), and CD73-expressing tumor cells (cells that scored as SOX2+), but not other populations of cells such as endothelial or lymphoid cells. These findings are consistent with co-localization of the CD39 and CD73 ectoenzymes in the 3D environment of GBMs and suggestive of functional crosstalk between tumor and myeloid cells.

To quantify the relative positioning of CD73 and CD39 ectoenzymes in a large set of specimens, we used spatial statistics to study the physical relationships between tumor, immune, and endothelial cell. The analysis was performed on 748,345 segmented cells identified using 36-plex conventional 2D CyCIF[9] of 172 IDH-WT glioblastomas. Spatial proximity was evaluated using Latent Dirichlet Allocation (LDA), a probabilistic modeling method that reduces complex collections of different entities (cell types in this case) into distinct recurrent cellular neighborhoods ('topics' in the nomenclature of LDA). LDA has been widely used in biodiversity studies and has recently been applied by our group and others to study the spatial organization of tissues[68,69]. LDA analysis revealed multiple distinct recurrent cellular neighborhoods in adult glioblastoma, including a neighborhood that combined CD39, CD73, SOX2, and CD163 expressing cells (Topic #11). Thus, glioblastoma tissues frequently contain neighborhoods with CD73 expressing tumor cells co-localized into communities that also contained CD39 myeloid cells (Fig. 8D). Mapping of the topics showed that glioblastomas are often heterogeneous for different functional neighborhoods between samples (Fig. 8E, F).

To investigate the statistical significance of cell-cell interactions identified by LDA analysis, we used spatial correlation. This simple approach has the advantage that it can accommodate the observation that CD73 and CD39 staining intensities in single cells exhibit a continuous distribution of signal with no objectively definable 'positive' and 'negative' populations. We found that CD73 expression in tumor cells was strongly correlated with CD39 expression in nearby myeloid cells ($p = 4.78e−19$) as well as expression of the immune checkpoint ligand PD-L1 ($p = 1.84e−64$) in myeloid cells (Fig. 8G and Fig. S12a). An interaction between CD73 and PD-L1 was also identified in recurrent neighborhood #11 (Topic #11) from the LDA analysis (Fig. 8D), though as previously noted the absolute intensity and number of PD-L1 expressing cells in glioblastoma cells is typically low. Notably, we did not identify a strong spatial correlation between tumor CD73 and all myeloid cells (defined by PU.1 expression) when CD39 is not considered ($p = −0.04$) (Fig. 8G). This suggests that the spatial relationship with tumor CD73 protein is specific to myeloid cells which also express CD39 protein, and not from generalized co-localization of tumor and myeloid cells regardless of functional state with incidental association of CD73 and CD39 (Fig. 8G). There was no strong correlation between tumor CD73 expression and endothelial or lymphoid cells (Fig. 8G). In keeping with the single-cell association between HIF1a and CD73 (Fig. S10i), CD73 and hypoxic myeloid populations (Fig. 2J, K), and the regional association between CD73 and hypoxia (Fig. 7F–I), spatial statistics showed a strongly significant correlation between tumor CD73 and HIF1a protein expression (in any cell population) ($p = 5.56e−56$) (Fig. 8H). Thus, tumor cells with CD73 expression exhibit specific co-localization with myeloid cells expressing CD39, and tumor cells with CD73 expression often exist in neighborhoods of hypoxic cells.

Because microglia are the most common myeloid cell type in glioblastoma[70], and exhibit the highest expression of CD39, we examined whether tumor CD73 was spatially co-localized with specific functional microglial classes that have been recently defined by scRNA-seq analysis[50]. Using CyCIF data, we found that CD73 expression in tumor cells was spatially correlated with CD163 and CD11b expression in PU.1-positive myeloid cells (Fig. 8i). While CD39 was broadly expressed across myeloid and microglial clusters in the scRNA-seq

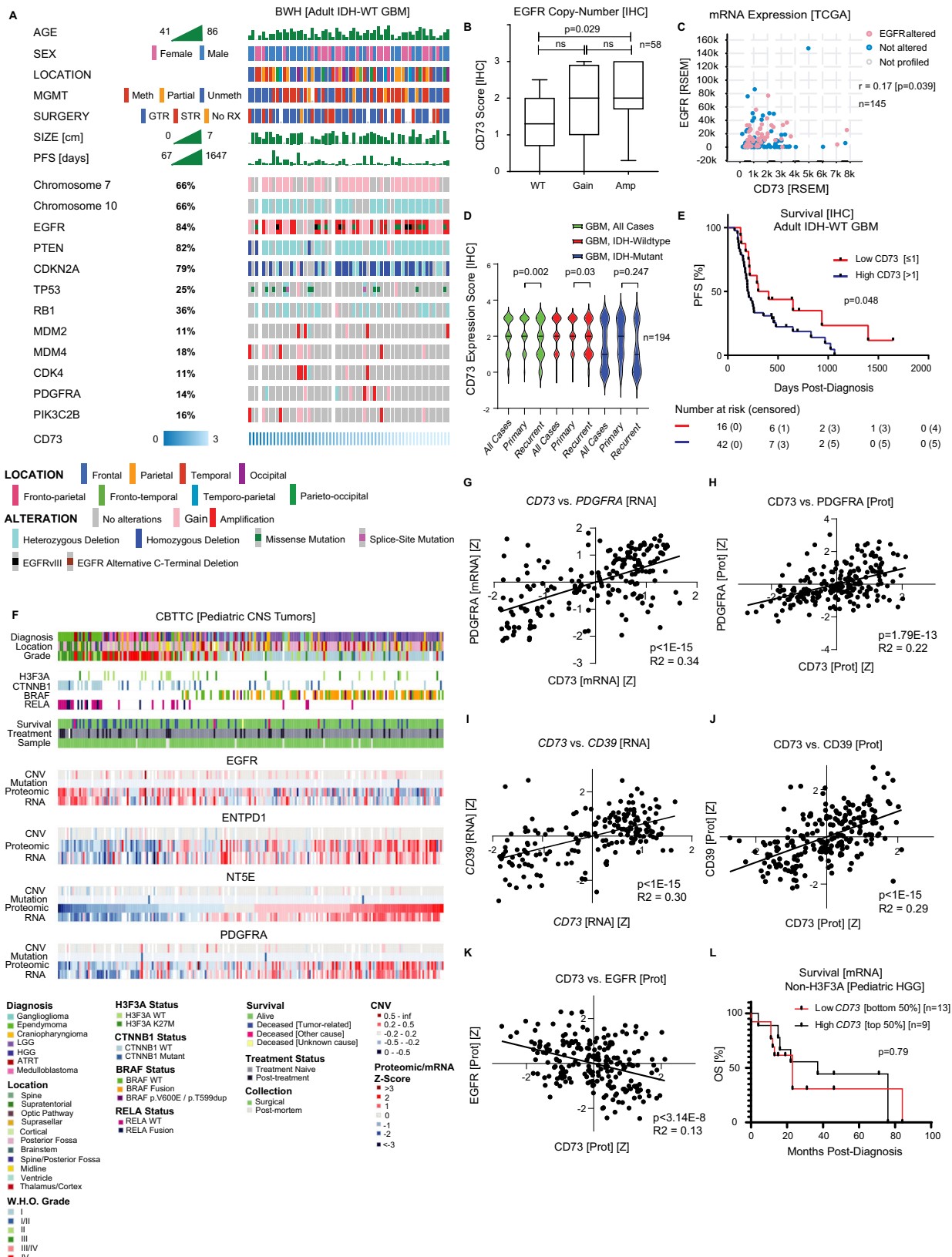

data, mapping a signature characterized by combined expression of CD39, CD163, and CD11b in this dataset revealed strong enrichment for a specific tumor-associated microglial subtype (Cluster 14 from San-kowski et al.[50]) that is characterized by inflammatory, interferon, and hypoxia-response genes (Fig. S12b)[50], similar to findings in scRNA-seq datasets, again supporting that expression of the core purine

regulators CD39 and CD73 is associated with a microenvironment characterized by inflammatory signaling and hypoxia.

If, as we hypothesize, spatial co-localization of tumor CD73 and microglial CD39 increases local production of adenosine by coordinating activity of rate-limiting catabolic enzymes, then tumors with increased spatial correlation might be expected to exhibit more

**Fig. 5 | CD73 expression correlates with genotype and clinical outcome in glioblastoma. A** To explore the genetic and clinical features associated with CD73 expression in glioblastoma, we evaluated a sub-cohort of 58 primary IDH-wildtype human glioblastoma from the BWH cohort with associated genome-wide chromosomal copy-number (array comparative genomic hybridization (ACGH)), 447-gene targeted exome sequencing (Oncopanel), *MGMT* promoter methylation analysis, and clinicopathologic data. These data showed a typical genomic profile for adult IDH-WT glioblastoma, with frequent polysomy 7, monosomy 10, *EGFR* gain/amplification, *PTEN*, *CDKN2A*, *TP53* loss, and other recurrent alterations. **B** Correlation of CD73 and a wide variety of recurrent genomic alterations was principally notable for a significant association between *EGFR* gene amplification and CD73 protein expression ($n = 58$, $p = 0.0029$, t-test, unpaired, two-sided); Tukey box-and-whisker plot with midline = median, box limits = Q1 (25th percentile)/Q3 (75th percentile), whiskers = 1.5 inter-quartile range (IQR), dots = outliers (>1.5IQR)). **C** Analysis of bulk mRNA expression in a broader cohort of IDH-WT glioblastoma (TCGA, $n = 145$) validated these findings, showing a weakly significant ($p = 0.03$, $r = 0.17$, Pearson's correlation test, unpaired, two-sided) direct correlation between *CD73* and *EGFR* levels. **D** In the full GBM cohort ($n = 194$), mean CD73 protein expression level (by IHC score) was higher in primary ($n = 128$, mean = 2.2) vs. recurrent tumors ($n = 66$, mean = 1.8) ($p = 0.002$, t-test, unpaired, two-sided, line indicates median) and this relationship was also significant in primary IDH-WT tumors ($n = 120$, mean = 2.3) vs. recurrent/residual IDH-WT tumors ($n = 52$, mean = 1.9) ($p = 0.03$, t-test, unpaired, two-sided) but not primary ($n = 8$, mean = 1.9) vs. recurrent ($n = 14$, mean = 1.2) IDH-mutant GBM ($p = 0.247$, t-test, unpaired, two-sided). **E** Mantel−Cox Log-Rank survival analysis showed that elevated CD73 protein expression (by IHC) was associated with significantly ($p = 0.048$) shorter progression-free survival (PFS) in adult IDH-WT glioblastoma. **F** We next assessed a cohort of 102 pediatric tumor specimens from the Children's Brain Tumor Network (CBTN), including most major tumor subtypes. Strong *CD73* expression was present in gliomas and glioneuronal tumors, including H3K27M-mutant diffuse midline glioma, with minimal expression in medulloblastomas, ATRT, and ependymoma. **G–J** Across pediatric CNS tumors, *CD73* expression was positively correlated with *PDGFRA* at the mRNA ($p < 1e−15$, t-test, unpaired, two-sided) and protein levels (1.79e−13; t-test, unpaired, two-sided), as well as *CD39* mRNA ($p < 1e−15$, t-test, unpaired, two-sided) and protein ($p < 1e−15$; t-test, unpaired, two-sided). **K** Unlike adult glioblastoma, there a negative correlation with *EGFR* ($p = 3.14e−8$, t-test, unpaired, two-sided). **L** Mantel−Cox Log-Rank survival analysis showed no significant difference in survival between non-H3F3A-mutated pediatric HGG with higher and lower expression of *CD73* (top 50% vs. bottom 50%) ($p = 0.79$).

aggressive behavior. This was indeed the case in our data. Survival analysis of the previously characterized cohort of 58 molecularly and clinical characterized primary IDH-WT glioblastoma (Fig. 5A) revealed shorter progression-free survival in tumors with strong spatial interaction of tumor CD73 and myeloid CD39 relative to tumors with weaker interaction ($p = 0.04$, Mantel−Cox Long-Rank, Fig. 8J). While most cases with strong interaction scores (87.5%, $n = 56$) also exhibited strong and widely-distributed expression of CD73 (IHC score ≥2) a subset (12.5%, $n = 8$) had low (IHC score <2) overall expression of CD73 with clusters of interacting tumor and myeloid cells. There was particularly poor progression-free survival (<200 days) in 3/8 (37.5%) of these cases. These data suggest that aggressive tumor behavior and poor clinical outcomes are associated not only with elevated CD73 expression, but more specifically, with spatial co-localization between tumor CD73 and myeloid CD39. Such co-localization can be observed even in tumors without diffuse CD73 expression.

To determine whether pediatric HGGs exhibit a similar spatial organization as adult glioblastoma, despite having different genomic drivers, we performed 24-plex CyCIF on 95 pediatric CNS tumors, including 39 HGGs previously characterized by IHC and/or genotypic analysis (Fig. 8K)[63]. As with the adult glioblastoma, CD73 staining by CyCIF was strongly correlated with previously defined CD73 IHC index scores, cross-validating the two approaches (Fig. S12c). Similar to adult glioblastoma, visual inspection showed that CD73-expressing tumor cells commonly clustered with CD39-expressing myeloid cells, implying spatial association of purine signaling components between tumor and myeloid cells (Fig. 8K). Spatial statistics confirmed this interpretation by demonstrating a correlation between CD73 expressed on tumor cells and CD39 expressed on physically proximate myeloid cells ($p = 1.56e−8$, Fig. 8L). This analysis also revealed a correlation between tumor CD73 and myeloid CD11b and CD163, suggesting that similar populations of microglia associate with tumor cells in pediatric and adult gliomas (Fig. 8L). While spatial statistics showed no significant correlation between tumor CD73 and HIF1a in pediatric tumors, there was spatial correlation between tumor CD73 and hypoxia-responsive genes including GLUT1, iNOS, and ARG1 (Fig. S12d). Inspection of these markers in regions of elevated CD73 showed that HIF1A, GLUT1, and ARG1 were enriched in tumor cells adjacent to regions of palisading necrosis, again suggesting that hypoxia may promote the expression of CD73 in neighborhoods that drive immunomodulatory signaling (Fig. S12e).

## Discussion

In this study, we used data from scRNA-seq, spatially resolved mass spectrometry, and highly-multiplexed tissue imaging of human clinical samples to generate a detailed inventory of the tumor and immune cell types and states involved in purinergic signaling in high-grade gliomas and to determine the spatial organization of these different cell types and states. Our findings suggest that purinergic signaling involves a division of labor between glioma tumor cells and microglia, each of which provides one of the two essential ectoenzymes needed to catabolize extracellular ATP into adenosine, a potent immunosuppressive metabolite. Myeloid cells, most prominently microglia, supply CD39, whereas tumor cells predominantly provide CD73, which catalyzes the rate-limiting catabolic step. We show that increased CD73 expression is quantitatively and spatially associated with elevated adenosine levels in glioma tumor tissue, validating the hypothesis that CD73 promotes tumorigenesis through enzymatic production of adenosine (Fig. 9).

Integrating information from multiple measurement modalities allowed us to identify several cell-autonomous and non-autonomous factors that likely influence CD73 expression in glioma tumor cells. In adult glioblastomas, the highest levels of CD73 are associated with *EGFR* activation and astrocyte-like differentiation, whereas in pediatric gliomas CD73 expression is linked to *PDGFRA* activation and OPC-like differentiation. In addition to the apparent role of cell lineage and oncogene activation in CD73 levels, CD73 is also strongly induced in peri-necrotic hypoxic regions in high-grade gliomas, suggesting that dynamic microenvironmental features and intrinsic genetic characteristics may interact to tune CD73 expression. Notably, while most tumor cells with mesenchymal-like differentiation have low levels of CD73, hypoxia is associated with CD73 induction in a subset of these glioblastoma cells, likely corresponding to those from focal peri-necrotic regions.

Single-cell transcriptomics also allowed us to interrogate the functional states associated with CD39 expression in myeloid cells in multiple datasets[41,50,51]. This analysis showed that the most prominent CD39 expressing cells are microglia which express core lineage signature genes such as *P2RY12*, *TMEM119*, *CX3CR1*, and *CSF1R*, with fewer cells exhibiting a peripheral macrophage signature, though this equilibrium shifts towards a greater influx of peripheral cells in recurrent tumors. *CD39*-expressing myeloid cells exhibit a pro-inflammatory phenotype characterized by purine pathway, MHC-I/II expression, hypoxia-response signaling, and interferon pathway activity[50], suggesting that purine signaling may be upregulated in an inflammatory microenvironment.

While we did not identify a spatial correlation between tumor CD73 and lymphocytes, endothelial cells, or myeloid cells in general, we did find a strong spatial correlation between tumor cells expressing CD73 and myeloid cells expressing CD39. The non-random distribution of these cells and the essential components of extracellular purine

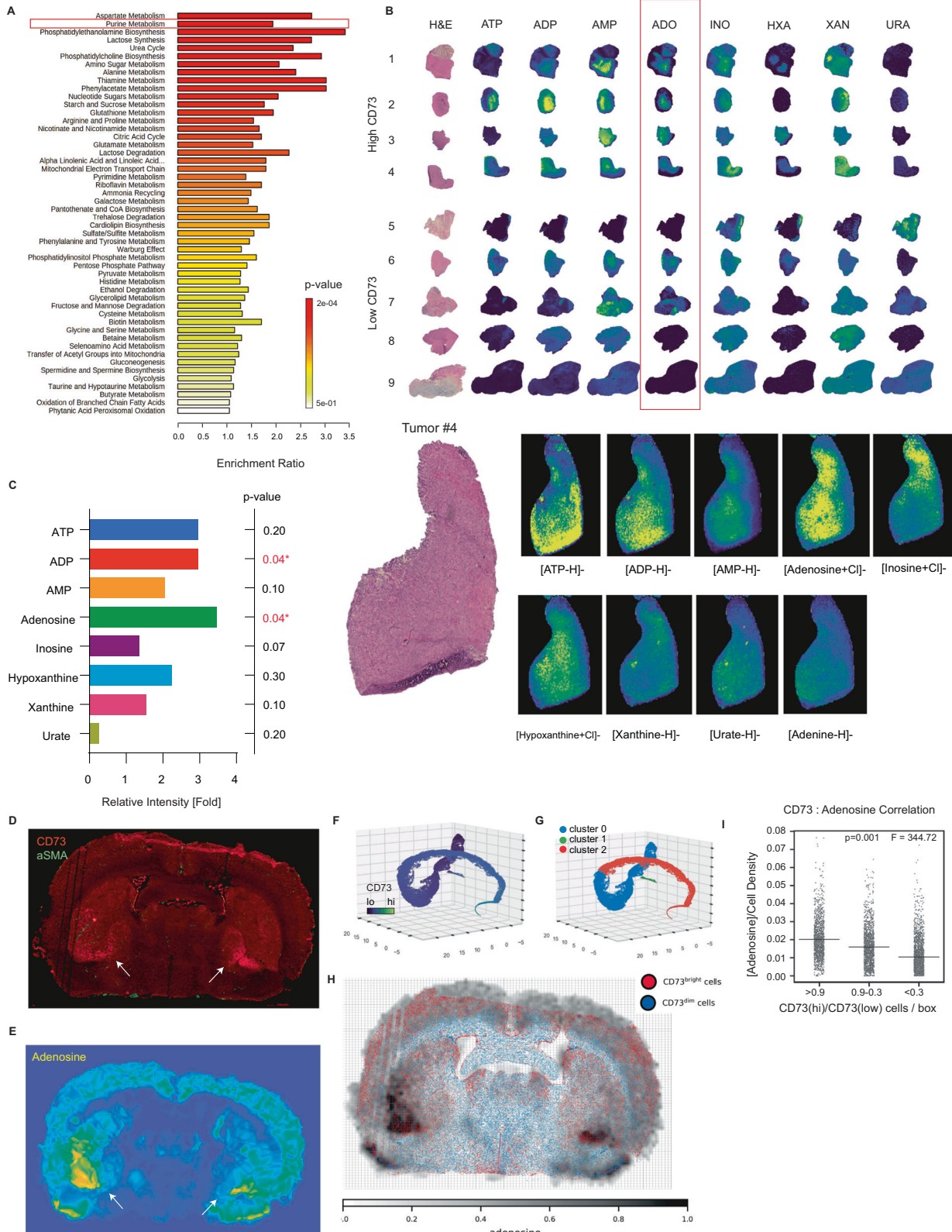

signaling that they elaborate suggest that positioning CD73 and CD39 near one another provides a selective advantage for glioma cells. Whether the spatial interaction of these ectoenzymes is a feature of other tumors besides gliomas is unknown; characterizing their spatial organization is of potential relevance to guiding the precise use of newly available therapies that target this pathway and which are showing early signs of efficacy in several tumor types[54,55]. Notably, pan-cancer analysis shows that glioblastomas exhibit high levels of *CD73* similar to tumors that have shown a promising response to anti-CD73 antibody therapy in early trials (e.g., NSCLC). Glioblastomas also demonstrate higher levels of multiple adenosine receptors (*ADORA1*/A$_1$R, *ADORA3*/A$_3$R) compared to all other cancer subtypes, suggesting that gliomas may have particularly active purinergic signaling, and therefore represent an attractive target for anti-CD73 therapy. While

**Fig. 6 | CD73 is associated with elevated adenosine levels in glioblastoma.**
**A** Mass spectrometry metabolomic profiling (MALDI-MSI) of frozen tissue from 9 IDH-wildtype glioblastoma resections showed that purine metabolism was among the most significantly enriched pathways ($p < 0.001$, t-test, unpaired, two-sided). **B**, **C** CD73 IHC performed on these samples showed that 4 cases exhibited high CD73 expression, while 5 cases exhibited low CD73 expression. Spatially-resolved analysis of all purine metabolites in these cases showed that adenosine was the most significantly elevated metabolite in tumor tissue in cases with high CD73 compared to those with low expression (3.5-fold, $p = 0.04$, t-test, unpaired, two-sided). ADP levels were also significantly elevated in these tumors ($p = 0.04$, t-test, unpaired, two-sided). **D**, **E** To further assess the association of CD73 and adenosine in CNS tissue, we performed immunofluorescence and spatially-resolved mass spectrometry on serial whole-mount murine coronal brain sections. CD73 and adenosine showed a similar distribution, with the greatest levels in the basal ganglia (arrows), with weaker expression in the cortical gray matter and minimal white matter expression. **F** Matched sections were spatially-registered and cells were segmented and clustered by 3D-embedded uniform manifold approximation and projection (UMAP). **G** Single-cell analysis revealed clusters associated with high (cluster 2) and low (cluster 0) CD73 expression, as well as αSMA-positive vascular cells (cluster 1). **H** Registered IF/MS maps were evaluated for cell-density corrected correlation between adenosine and CD73 in all brain regions. **I** This analysis demonstrated a strongly significant spatial correlation between CD73 and adenosine levels ($p = 0.001$, $F = 344.72$, Tukey's HSD test, unpaired, one-sided).

spatial biomarkers have not yet been implemented for patient management, new multiplexed imaging technologies[71,72] make it feasible to implement such tests to the standard of Laboratory Developed Tests (LDTs) conducted in CLIA-certified laboratories. Moreover, methods for rapid measurement of metabolite levels directly in human tissues using mass spectrometry imaging may provide additional information to guide the development and use of therapies[73,74].

Tumors with high CD73 and CD39 expression exhibit elevated numbers of immune cells and heightened inflammation compared to those with lower expression. In other pathophysiological settings, release and catabolism of ATP to adenosine functions as a mechanism to first amplify and then limit inflammation[52]. Given the well-established strongly immunosuppressive functions of CD73 and adenosine, and evidence of elevated adenosine levels in glioblastoma tissues with high CD73 expression, upregulation of CD73 may occur as an adaptive response by tumor cells to ongoing inflammation and cell death. On the other hand, CD73 expression may be aberrantly upregulated by underlying genetic features such as *EGFR* amplification and provide a selective advantage to strongly expressing cells in the face of immune activity, particularly those in close spatial proximity to CD39-expressing microglia, which enable more efficient enzymatic generation of adenosine.

In gliomas, the mechanisms underlying the spatial association of tumor CD73 and microglial CD39 are not certain and will require further study, though GSEA analysis of *CD73*[hi] tumor cells reveals elevated expression of chemokines such as CXCL2, which may recruit myeloid cells in cancer[75]. Prior studies have also shown that glioblastoma tumor cells may upregulate the expression of CD39 in macrophages via secretion of molecules such as kynurenine[76]. Thus, spatial associations may be promoted by local microenvironmental conditions (e.g., hypoxia), recruitment of myeloid cells by chemokine secretion, and induction of CD39 expression.

Although we find that gliomas generally exhibit few cells with substantial expression of PD-L1, the spatial correlation that we observe between PD-L1 and CD39 in myeloid cells and CD73 in tumor cells suggests that the cellular neighborhoods that form in glioblastoma involve multiple, partially redundant, immunomodulatory mechanisms. One recent study specifically analyzing tumor-associated immune cells through dissociative methodologies identified a distinct population of tumor-infiltrating macrophages that express CD73 in glioblastoma[6], and showed that these cells may persist through recurrence and promote resistance to immunotherapy[3]. By scRNA-seq and tissue-based imaging we also find a subset of tumor-associated myeloid cells that express CD73 and find that these cells express hypoxic gene signatures. However, we find that such cells may be significantly less numerous in glioma than previously implied, indicating they make a minor contribution to overall CD73 activity as compared to diffuse tumor cell CD73 expression. Nevertheless, glioblastoma tumor growth was reduced in *CD73*-null mice with combined inhibition of CTLA-4 and PD-1 checkpoints[34] suggesting a potential benefit of combined inhibition of extracellular purinergic pathway and checkpoint pathways. A similar benefit of combined anti-CD73/anti-

PD1 therapy has been observed in models of EGFR-mutated NSCLC[59], and early-stage clinical trials[57,58], suggesting that CD73 inhibition may provide additional complementary or synergistic therapeutic benefits with other immunotherapies.

In conclusion, using multiple dissociative and non-dissociative analytical methods, we comprehensively identify and map cell types and states in high-grade gliomas, demonstrating that tumor cells are, in fact, the predominant source of CD73, while microglia are the predominant source of CD39. We show that these cell populations interact within discrete neighborhoods in the tumor microenvironment, leading to increased tissue adenosine-levels and poor clinical outcomes. CD73 expression is associated with genotype, lineage-differentiation, and dynamic functional states such as hypoxia in high-grade gliomas, suggesting that purinergic signaling and metabolic activity is a product of both intrinsic genetic features of the tumor as well as emergent population-level tissue interactions. Significant population-level features such as the interaction of CD73-expressing tumor cells and CD39-expressing microglia and their contribution to tumor biology and clinical outcomes are not captured by focused analysis of specific cell populations (e.g., FACS-sorted immune cells), nor dissociative sequencing methods, reinforcing the importance of understanding and mapping cell states within native tissue architecture. The high levels of CD73 expression and purinergic pathway activity identified in both adult and pediatric glioblastoma indicate that these tumors may represent attractive potential targets for therapeutic intervention in clinical trials.

# Methods

## Ethics

Our research complies with all relevant ethical regulations and was reviewed and approved by the Institutional Review Boards (IRB) at Brigham and Women's Hospital (BWH), Harvard Medical School (HMS), Dana Farber Cancer Institute (DFCI), and Children's Hospital of Philadelphia (CHOP).

Discarded human formalin-fixed paraffin-embedded (FFPE) tissue samples were used after diagnosis under excess tissue discarded tissue protocol 2018P001627 (reviewed and managed by the Mass General Brigham Institutional Review Board), which waives the requirement for patient consent. Adult glioblastoma samples and pediatric high-grade glioma samples subjected to genomic analysis were used after written informed patient consent had been obtained under Dana-Farber Cancer Institute IRB protocol 10–417 and Children's Hospital of Philadelphia IRB protocol 19-016112. For pediatric patients, written informed consent for genomic analysis was obtained from the legally authorized representatives/guardians. The study is compliant with all relevant ethical regulations regarding research involving human tissue specimens.

## Tissue characterization

FFPE tissue specimens were retrieved from the archives of Brigham and Women's Hospital with institutional review board (IRB) approval under DFHCC Protocol #10-417 or waiver of consent protocols. The

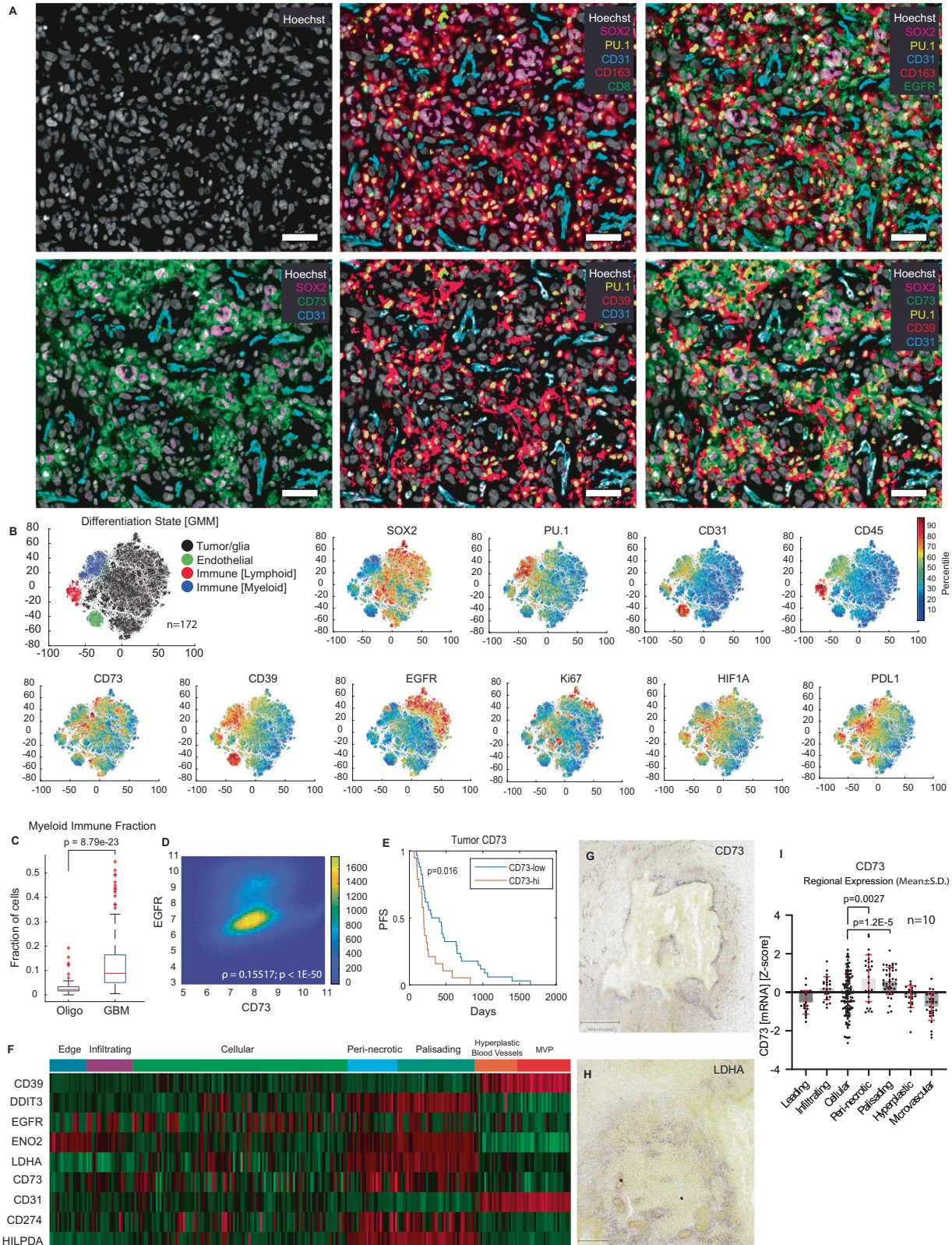

study is compliant with all relevant ethical regulations regarding research involving human tissue specimens. The Principal Investigator is responsible for ensuring that this project was conducted in compliance with all applicable federal, state, and local laws and regulations, institutional policies, and requirements of the IRB. All cases were reviewed and classified according to the revised WHO 2016 Classification of Tumors of the Central Nervous System (S.C., S.S.). The

characteristics of cases, including demographics, genotyping, immunohistochemical, and clinical data are provided (Supplementary Data 5–10) in accordance, when possible, with emerging metadata standards for tissue imaging[77] and IRB guidelines.

Specimens included 194 adult glioblastoma specimens (GBM, W.H.O. grade 4) from 179 distinct patients, including 128 primary and 66 recurrent tumors; 14 oligodendroglioma, IDH-mutant, 1p/19q co-

**Fig. 7 | Multiplexed single cell analysis of glioblastoma tissue. A** 36-plex tissue-based cyclic immunofluorescence (CyCIF) encompassing core lineage, signaling, immune, and purine pathway components was performed on a tissue microarray including 172 IDH-WT glioblastoma tissue specimens (representative CyCIF images from one glioblastoma case are shown; each tumor specimen in the microarray cohort underwent one CyCIF staining experiment); scalebars 50 μm. **B** Cells were segmented and fluorescence intensity was quantified for each marker at a single-cell level. Cells were clustered by marker expression in an unbiased fashion using Gaussian mixture modeling (GMM) showing four dominant signatures, which were validated by evaluation of lineage-specific markers. As with transcriptomic analyses, CD73 was predominantly expressed by tumor cells (SOX2+), while CD39 was predominantly expressed by myeloid (PU.1+) and endothelial (CD31+) cells. Cells were colored by the percentile of signal intensity for each marker. **C** Immune profiling showed that a substantial proportion of all cells in IDH-WT glioblastoma ($n = 172$ tumors) were myeloid cells, which comprise the dominant immune population. There were significantly greater numbers of myeloid cells in glioblastoma compared to low grade oligodendrogliomas ($n = 14$ tumors) ($p = 8.79e$ $-23$, Wilcoxon rank sum test, unpaired, two-sided), suggesting a greater degree of immune activation in these tumors; Tukey box-and-whisker plot with midline = median, box limits = Q1 (25th percentile)/Q3 (75th percentile), whiskers = 1.5 inter-quartile range (IQR), dots = outliers (>1.5IQR)). **D** Single-cell analysis of functional

markers showed that EGFR protein expression was strongly correlated with CD73 expression ($p < 1e-50$, t-test on Z-transformed Pearson correlation, unpaired, two-sided). **E** Mantel−Cox Log-Rank survival analysis showed that higher mean levels of CD73 by single-cell CyCIF analysis correlated with significantly shorter PFS ($p = 0.016$). **F** To assess the relationship between CD73 and hypoxia in glioblastoma tissue, we first evaluated mRNA expression of *CD73*, *CD39*, and hypoxia-response genes (*DDIT3, ENO2, LDHA, HILPDA*) in curated histologic regions in the Ivy-GAP glioblastoma RNA-seq dataset ($n = 10$ tumors). This analysis showed that *CD73* was enriched in regions of palisading tumor necrosis and peri-necrotic tumor associated with high levels of hypoxia-response genes. *CD39* expression was present in all regions but showed stronger expression in regions of pathognomonic microvascular proliferation (MVP). **G, H** These results were validated by review of images of in-situ hybridization of tumor tissue, showing accentuation of *CD73* and *LDHA* signal in peri-necrotic hypoxic regions (for example, ISH from representative glioblastoma specimens are shown, one round of ISH was performed on cohorts of 37 independent glioblastomas (*NTSE*) and 29 glioblastomas (*LDHA*), with similar results across independent specimens (see https://glioblastoma.alleninstitute.org/ish)); scalebars: **G** 994 μm, **H** 576 μm. **I** Peri-necrotic ($p = 0.0027$, t-test, unpaired, two-sided, with Welch's correction) and palisading ($p = 1.2e-5$, t-test, unpaired, two-sided, with Welch's correction) regions exhibited significantly higher *CD73* than cellular tumor regions (data derived from 10 adult glioblastoma specimens)).

---

deleted specimens (W.H.O. grade 2) from 13 distinct patients, including 12 primary and 2 recurrent tumors; 15 anaplastic oligodendroglioma, IDH-mutant, 1p/19q co-deleted specimens (W.H.O. grade 3), including 6 primary and 9 recurrent tumors; 22 pilocytic astrocytomas (W.H.O. grade 1) from 22 distinct patients; 139 meningiomas (W.H.O. grade 1) from 139 distinct patients, including 130 primary and 9 recurrent tumors; 57 atypical meningioma specimens (W.H.O. grade 2) from 57 distinct patients, including 49 primary resections and 8 recurrent tumors; 19 anaplastic meningiomas (W.H.O. grade 3) from 19 distinct patients, including 11 primary and 8 recurrent tumors; 30 ependymomas (W.H.O. grade 2) from 30 distinct patients, including 25 primary and 5 recurrent tumors; 14 anaplastic ependymomas (W.H.O. grade 3) from 14 distinct patients, including 10 primary and 4 recurrent tumors; 24 medulloblastomas (W.H.O. grade 4), including 24 primary tumors, 22 adamantinomatous craniopharyngiomas (W.H.O. grade 1), including 15 primary tumors; 16 papillary craniopharyngiomas (W.H.O. grade I), including 9 primary tumors.

All medulloblastomas were classified by histologic subtype per 2016 W.H.O. criteria (S.C., S.S.) and included 15 classic, 3 desmoplastic/nodular, 1 extensive nodularity, and 5 large/cell anaplastic variants. Molecular data, including genotype and/or chromosomal copy-number analysis were available for 11/24 (45.8%) medulloblastomas, and all cases were classified according to molecular subtype per 2016 W.H.O. criteria (S.C., S.S.) (cases without molecular data classified as 'NOS'), including 13 NOS, 6 SHH, 3 Group 3, and 2 Group 4 tumors. Targeted exome sequencing (Oncopanel, 447 gene panel, including the *IDH1/2* loci) and chromosomal copy-number analysis (array comparative genomic hybridization, ACGH) was performed on adult GBM, oligodendroglioma, and anaplastic oligodendroglioma specimens during routine clinical practice. The presence or absence of IDH1 R132 or IDH2 R172 mutations were confirmed by IDH1 R132H mutation-specific IHC and/or sequencing. Of the 194 adult glioblastoma specimens, 178 underwent targeted exome sequencing. In the remaining 18 cases in which sequencing was not performed, IDH-status was assessed by routine clinical IDH1 R132H mutation specific immunohistochemistry. 172 glioblastomas were IDH-wildtype (156 sequence confirmed, 16 IHC-negative), while 22 were IDH-mutant (20 sequence confirmed, 2 IHC-positive). 108 primary and 48 recurrent IDH-WT glioblastoma had associated molecular pathology data. MGMT promoter methylation data was available for 103 primary IDH-WT glioblastoma. Clinical outcome data was available for 58 primary IDH-WT glioblastomas with associated molecular pathology data.

For oligodendrogliomas, the presence of whole-arm co-deletion of chromosomes 1p and 19q was confirmed by chromosomal microarray (array comparative genomic hybridization (ACGH)) and/or sequencing. All ependymomas exhibited predominantly classic histologic features. All PCP had *BRAF* V600E mutation by genotype and IHC and lacked *CTNNB1* mutations. 21/22 ACP had *CTNNB1* activation by genotype and/or IHC and lacked *BRAF* V600E mutations by genotype and/or IHC.

An additional independent tissue microarray of 88 FFPE pediatric tumors, including high-grade gliomas, primitive neuroectodermal tumors (PNET), atypical teratoid rhabdoid tumors (ATRT), and medulloblastomas was obtained from the Children's Brain Tumor Tissue Consortium (CBTTC) in accordance with IRB guidelines. From this cohort, we performed further immunohistochemistry and CyCIF analysis on 28 pediatric high-grade gliomas and 11 H3K27M-mutant diffuse midline gliomas with clinical and molecular characterization.

## Immunohistochemistry

FFPE sections were de-paraffinized, dehydrated, and endogenous peroxidase activity was blocked. Antigen retrieval was performed in Dako citrate buffer 123 °C+/−2, 45 s, at 15+/−2 PSI. Slides were incubated with anti-CD73 antibody, (Abcam, EPR6115, rabbit monoclonal clonal) at 1:5000 for 45 min or anti-CD39 antibody, (Abcam, EPR20627, rabbit monoclonal clonal) at 1:1000 for 45 min, washed, then incubated with Labeled Polymer-HRP anti-rabbit secondary antibody (DakoCytomation, K4011) for 30 min. Slides were then incubated with DAKO DAB+ solution for 3–5 min and counterstained with hematoxylin. We considered membranous expression of CD73 to represent a physiologically relevant (active) expression of the protein. Images were acquired using an Olympus BX41 microscope and an Olympus DP26 digital camera.

## CD73 expression profiling

Immunohistochemical expression of CD73 was evaluated using a semi-quantitative scoring system. In brief, cases were scored according to the following system: Absent (0): No expression of CD73 by tumor cells, or weak membranous staining in <10% of tumor cells, Weak (1): Weak membranous staining in >10% of tumor cells, Moderate (2): Moderate membranous staining in >10% of tumor cells, or strong/circumferential membranous staining in <10% of tumor cells, Strong (3): Strong/circumferential membranous staining in >10% of tumor cells.

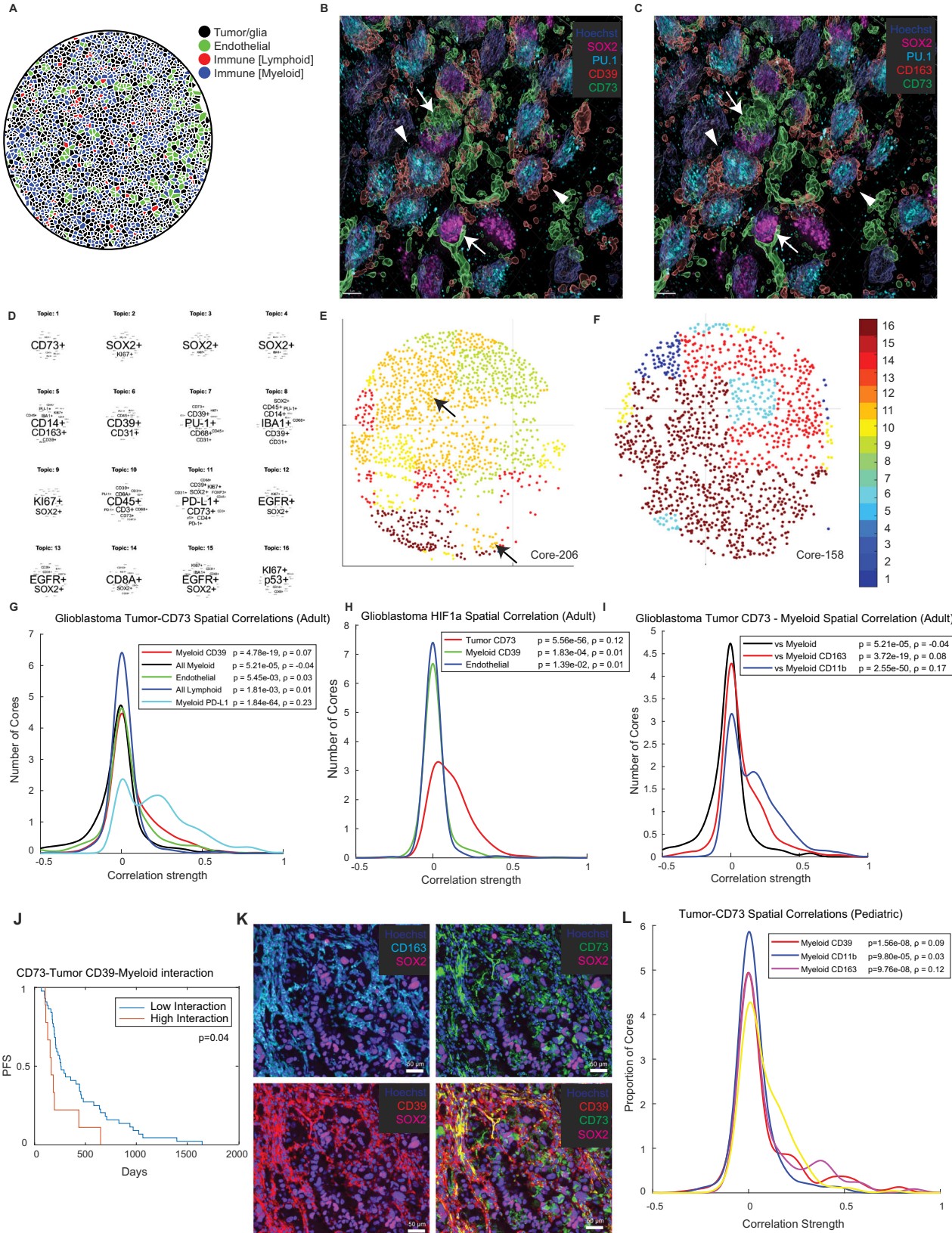

## Single-cell RNA-Seq analysis

Single-cell analyses of a cohort of 28 adult and pediatric high-grade gliomas in Figs. 1 and 2A–C, and Fig. S1 were performed using the data from a previously published study (Neftel et al. 2019)[41].

T-stochastic neighborhood embedding plots (t-SNE), lineage plots, and dot-plots were generated using the Broad Institute Single Cell Portal interface (https://singlecell.broadinstitute.org/single_cell/study/SCP393/single-cell-rna-seq-of-adult-and-pediatric-glioblastoma#study-visualize). The processed log TPM gene expression matrix, metadata file, and the hierarchy data were all downloaded from the Broad Single Cell portal (https://singlecell.broadinstitute.org/single_cell).

**Fig. 8 | Spatial patterning of core purine regulators in adult glioblastoma and pediatric high-grade glioma. A** Given the apparent spatial coordination of CD73, adenosine, and functional states such as hypoxia, we sought to better understand the precise spatial landscape of purinergic signaling in glioblastoma. Segmentation and Voronoi visualization of each cell type revealed an apparent non-random distribution of cells closely intermingled tumor and myeloid cells. **B, C** 18-plex cyclic multiplexed immunofluorescence (CyCIF) of an adult glioblastoma specimen with high-resolution deconvolution imaging and 3D reconstruction using a Deltavision Elite microscope confirmed close spatial co-localization of SOX2-positive tumor-associated CD73 (arrows) and PU.1/CD163-positive myeloid cell-associated CD39 proteins (arrowheads, representative region shown, CyCIF was performed once); scale bars 5 μm. **D** Latent Dirichlet Association (LDA) probabilistic modeling was used to globally analyze CyCIF data from the larger dataset of 172 IDH-WT glioblastoma to reduce cell populations into neighborhoods ("topics") defined by single-cell marker-expression patterns. This confirmed the presence of prominent neighborhoods with a spatial association between tumor (SOX2), myeloid (CD163/PU.1), and core purine regulators (CD39, CD73) (Topic 11), as well as other neighborhoods corresponding to various tissue niches. **E, F** Glioblastomas are heterogeneous collections of intermixed cellular neighborhoods, including large niches with prominent tumor-CD73:myeloid-CD39 interaction (**E**, arrows). **G** To assess the

strength of this spatial correlation, we next performed spatial statistics analysis, showing that tumor CD73 is strongly associated with myeloid CD39 ($p = 4.78e{-}19$, Pearson's correlation test, unpaired, two-sided) and PD-L1 protein ($p = 1.84e{-}64$), but less so with endothelial or lymphoid cells and myeloid cells themselves. **H, I** Tumor CD73 was also correlated ($p = 5.56e{-}56$, Pearson's correlation test, unpaired, two-sided) with regions of elevated nuclear HIF1A expression, and myeloid cells (PU.1-positive) with coordinate CD163 ($p = 3.72e{-}19$, Pearson's correlation test, unpaired, two-sided) and CD11b ($p = 2.55e{-}50$) expression. **J** Mantel–Cox survival analysis showed that cases with higher levels of spatial interaction between tumor CD73 and myeloid CD39 showed particularly poor progression-free survival (PFS). **K** 24-plex CyCIF analysis of a second cohort of 95 pediatric CNS tumors, including 28 pediatric HGG and 11 H3K27M-mutant diffuse midline gliomas, showing similar co-localization of tumor CD73 and myeloid CD39 (representative staining from a high-grade glioma specimen shown, CyCIF was performed once for each specimen); scale bars, 50 μm. **L** Spatial statistics again showed a strongly significant spatial correlation between tumor CD73 and myeloid CD39 ($p = 1.56e{-}8$, Pearson's correlation test, unpaired, two-side), as well as myeloid CD163 ($p = 9.76e{-}8$) and CD11b ($p = 9.80e{-}5$) in pediatric high-grade gliomas and diffuse midline gliomas.

Further analysis in Fig. 1H–J and Fig. S1f, g were performed in R (v4.2.0) using the Seurat (v4.1.1) package[78] and the figures were plotted using the ggplot2 package (v3.3.6). Pathway enrichment analyses were performed using GSEA[48] and GO Enrichment Analysis[79–82].

Single-cell analyses of an additional cohort of 7 newly diagnosed adult glioblastomas, and 4 recurrent adult glioblastomas in Fig. 2D–M and Fig. S3a–m were performed using the data from a previously published study (Pombo Antunes et al. 2021[52]). UMAP and dataplots were generated using the Brain Immune Atlas interface (https://www.brainimmuneatlas.org/umap-humanGbm-diag-full.php). The processed log TPM gene-cell count matrix and cell annotation matrices were downloaded for analysis from the Brain Immune Atlas (https://www.brainimmuneatlas.org/download.php).

Single-cell analyses of microglia from a cohort of 15 brain specimens from individuals undergoing surgery for epilepsy, glioma, or metastasis presented in Figs. S3n–p and S12b were performed using data from a previously published study (Sankowski et al. 2019). T-stochastic neighborhood embedding plots (t-SNE) were generated using the associated Single cell data viewer (https://single-cell.shinyapps.io/human-microglia/).

Single-cell analysis of a cohort of 13 developing human brains presented in Fig. S2 were performed using data from a previously published study (Nowakowski et al. Science 2017). T-stochastic neighborhood embedding plots (t-SNE) were generated and analyzed in the UCSC Cell Browser (https://cells.ucsc.edu/?ds=cortex-dev).

### Animal management
For mouse brain mass spectrometry experiments, brains were received after removal from the Mayo Clinic. Mice were maintained with a 12 h light, 12 h dark schedule with no entrance of researchers during the dark cycle. Temperature was 68–72 °F with 30–70% humidity. Chow was available ad libitum (fat content 21.6%, PicoLab mouse Diet, Cat # 5058). There were 5 mice per ventilated cage on corn cob bedding. Water was accessible at all times. Cervical dislocation without anesthesia was used to sac the mice. The research protocol was approved by the Mayo Clinic Institutional Review Board (IRB) and Brigham and Women's Hospital Institutional Review Board (IRB) and adhered to all ethical guidelines.

### Immune-related signatures assessment in TCGA GBM RNA-seq data
TCGA GBM RNA-seq data were processed using Kallisto[83] and the Gencode GTF[84] by the Piccolo team[85]. This counts matrix, along with the metadata table were downloaded from https://osf.io/gqrz9/files. Immune cell population signatures from Pombo Antunes et al.[51], were

assessed by computing the log2 mean of transcriptomic markers expression for each population (Supplementary Data 1). Inflammation-related signatures were taken from Gromeier et al.[86] (Supplementary Data 2). Heatmaps were drawn using the ComplexHeatmap (v3.15) R package[87], scaled per gene list for visualization. Violin plots of computed expression means were done using the geom_violin function from the ggplot2 (v3.3.6) R package[88]. Statistics (Wilcoxon test, unpaired, two-sided) between *NT5E*, *ENTPD1* single or double high and low sample groups were computed using the geom_signif function from the ggsignif R package. Tumor purity of TCGA GBM bulk RNA-seq data estimated with the ABSOLUTE R package[89] or the combination of several tools (CPE: consensus measurement of purity estimations) were extracted from Aran et al.[90]. 168 samples had an acceptable estimation of tumor purity in Aran et al. Of the 168 samples, 155 were primary tumors, and 13 recurrent tumors. Clinical data regarding recurrence status was extracted from TCGA.

### MALDI tissue preparation
Frozen resection tissue from 9 glioblastomas were selected for MALDI MSI. Immunohistochemistry showed that 4 cases had 'high' CD73 by IHC (IHC score = 2 or 3), while 5 cases had 'low' CD73 by IHC (IHC score = 0 or 1). Tissue was sectioned to 10 μm, thaw mounted onto indium-tin-oxide (ITO) slides, and serial sections were obtained for H&E staining. A high-resolution image of the whole H&E tissues was obtained through image stitching (Zeiss Observer Z.1, Oberkochen, Germany) using a plan-apochromat lens (20×) using an AxioCam MR3 camera. Matrix preparation of 1,5-diaminonaphthalene hydrochloride was prepared to a concentration of 4.3 mg/mL in 4.5/5/0.5 HPLC grade water/ethanol/1 M HCl (v/v/v). The matrix was sprayed using a TM-sprayer (HTX imaging, Carrboro, NC) with parameters of a flow rate (0.09 mL/min), spray nozzle velocity (1200 mm/min), spray nozzle temperature (75 °C), nitrogen gas pressure (10 psi), track spacing (2 mm) and a four-pass spray.

### MALDI FT-ICR mass spectrometry imaging
The mass spectrometry imaging experiments were conducted using a 9.4 Tesla SolariX XR FT-ICR MS (Bruker Daltonics, Billerica, MA) in negative ion mode. Using a tune mix solution (Agilent Technologies, Santa Clara, CA) the method was calibrated through the electrospray source. The instrumental parameters included a defined pixel step size of 100 μm that covered the m/z range of 58–1200. Each pixel consisted of 200 shots at a laser power of 12% (arbitrary scale) and frequency of 1000 Hz. To bias the acquisition to improve the capture of the purine metabolites mass range, continuous accumulation of selected ions (CASI) mode was used, in which the Q1 was set to m/z

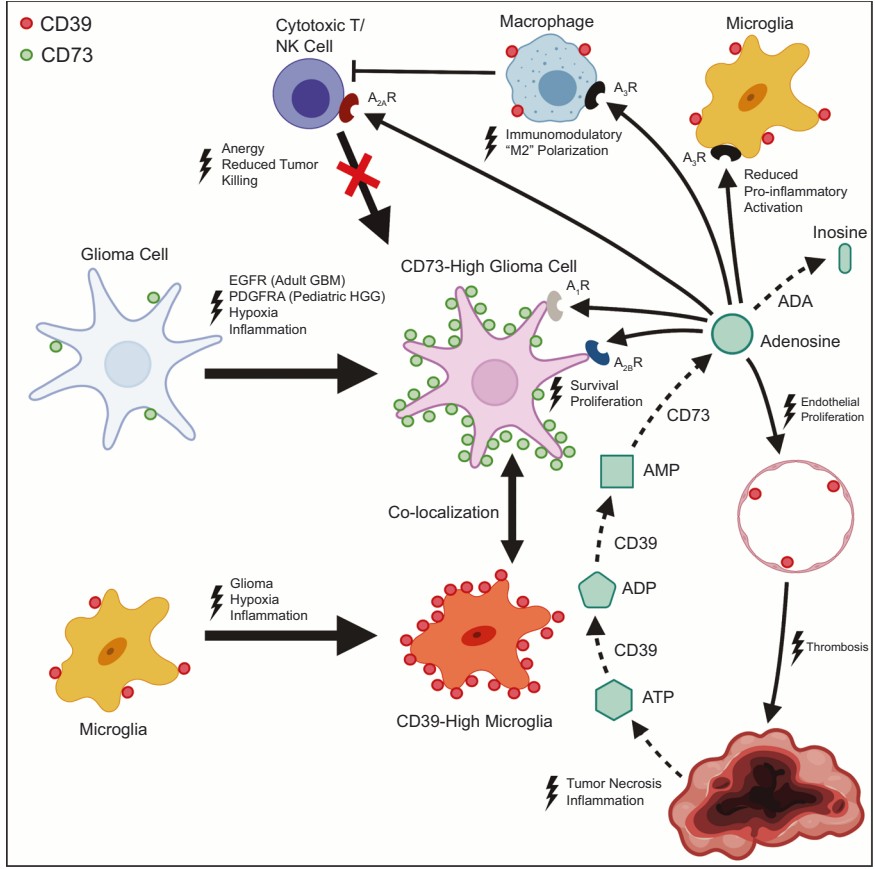

**Fig. 9 | Spatial topography of purine signaling in glioblastoma and pediatric high-grade glioma.** Collectively, data suggests a model in which *EGFR* activation (adult IDH-WT glioblastoma), *PDGFRA* activation (pediatric high-grade glioma) and/or hypoxia promote CD73 expression by tumor cells in an inflammatory microenvironment. Spatial co-localization of tumor cell-specific CD73 with microglial CD39 augments adenosine production in the tumor microenvironment, promoting immune tolerance and tumorigenesis through action on multiple cell populations. Created with BioRender.com.

350 with an isolation window of 400. A standard of adenosine was spotted on the same slide containing the tissues and imaged at the same time of the MSI acquisition. Further validation on metabolite annotations was performed by tandem MSMS from tissues using a timsTOF FleX (Bruker Daltonics, Billerica, MA) and compared to MS/MS measurements performed from standards (Supplementary Data 11). Ion images and mass spectra were viewed and processed using SCiLS Lab software (version 2019c premium, Bruker Daltonics, Billerica, MA), in which the dataset was normalized to the total ion current (TIC). Metabolites were putatively annotated using an accurate mass of Δppm <2.5 and MSMS measurements, and cross-matched using the Metlin metabolite database (https://metlin.scripps.edu/landing_page.php?pgcontent=mainPage)[91].

## t-CyCIF protocol
The detailed protocol for tissue-based cyclic immunofluorescence (t-CyCIF)[9] is available at protocols.io (dx.doi.org/10.17504/protocols.io.bjiukkew). The BOND RX Automated IHC/ISH Stainer was used to bake FFPE slides at 60 °C for 30 min, dewax using Bond Dewax solution at 72 °C, and perform antigen retrieval using Epitope Retrieval 1 (LeicaTM) solution at 100 °C for 20 min. Slides underwent multiple cycles of antibody incubation, imaging, and fluorophore inactivation. All antibodies were incubated overnight at 4 °C in the dark. Slides were stained with Hoechst 33342 for 10 min at room temperature in the dark following antibody incubation in every cycle. Coverslips were wet-mounted using 200 μL of 10% Glycerol in PBS prior to imaging.

For conventional CyCIF, images were taken using a 20× objective (0.75 NA) on a CyteFinderII slide scanning fluorescence microscope (RareCyte Inc. Seattle WA) using CyteFinder software (v3.11.024) or on an IN Cell 6000 Analyzer using Acquisition Software v6.1 and v7.3.

For 3D reconstruction, images were taken using a 60× objective (1.42 NA) on a Deltavision Elite deconvolution microscope using SoftWoRx software (v.7.0.0).

Fluorophores were inactivated by incubating slides in a 4.5% $H_2O_2$, 24 mM NaOH in PBS solution and placing under an LED light source for 1 h.

For CyCIF experiments, the EPR6115 anti-CD73 clone was used following cross-validation of multiple clones.

## Antibodies
The antibodies used in this study are provided in the Antibody List (see Supplementary Table 1).

## t-CyCIF image processing and data quantification
Image analysis was performed with the Docker-based NextFlow pipeline MCMICRO[66] and with customized scripts in Python (v3.6), ImageJ (v1.53r), and MATLAB (v2017, 2018a, 2019b). All code is available on GitHub (https://github.com/labsyspharm/cd73_coy_spatialcorrelation) and Zenodo (https://zenodo.org/record/6628875#.YqJVfqjMJD8). DOI: 10.5281/zenodo.6628875. Briefly, after raw images were acquired, the stitching and registration of different tiles and cycles were done with MCMICRO[66] using the ASHLAR (v1.10.2)[92] module. The assembled OME.TIFF files from each slide were then passed through

quantification modules. For background subtraction, a rolling ball algorithm with 50-pixel radius was applied using ImageJ/Fiji. For segmentation and quantification, UNMICST2[93] was used supplemented by customized ImageJ scripts[9] to generate single-cell data. More details and source code can be found at www.cycif.org.

### Single-cell data QC for t-CyCIF

Single-cell data was passed through several QC steps during the generation of the cell feature table. Initial QC was done simultaneously with segmentation and quantification, so that cells lost from the specimen in the later cycles would not be included in the output. Next, single-cell data was filtered based on mean Hoechst staining intensity across cycles; cells with coefficient of variation (CV) greater than three standard deviations from the mean were discarded as were any objected identified by segmentation as "cells" but having no DNA intensity. These steps are designed to eliminate cells in which the nuclei are not included as a result of sectioning. Highly autofluorescent (AF) cells (measured in cycle 1 or 2) were also removed from the analysis, using a customized MATLAB script that applied a Gaussian Mixture Model (GMM) to identify high-AF population. More details and scripts can be found online (https://github.com/sorgerlab/cycif).

### Blinding

When visual tissue analyses were performed (e.g., histologic tissue QC, IF/IHC evaluation and QC, CD73 IHC scoring by a pathologist), relevant investigators (S.C., S.S.) were blind to the precise diagnosis (e.g., IDH-mutant vs. IDH-wildtype, H3K27M-mutant diffuse midline glioma vs. pediatric HGG, NOS) and clinical (e.g., patient demographics, PFS, OS), molecular (e.g., IDH1/2-status, EGFR-status, MGMT-status, MMR-status, etc.), and other relevant data to avoid bias in assessment of tissue adequacy and scoring.

Some investigators (S.W., J.R.L.) performing the initial single-cell, spatial statistics, and outcome analyses using CyCIF imaging data were not involved in the initial tumor cohort selection or visual tissue analysis, and were generally blind to source tissue of cells except where necessary during the analysis of single-cell imaging data to further avoid potential bias in the processing of single-cell data and application of analytic methods to this data.

### Data exclusion and quality control

For tissue microarray data (CyCIF, immunohistochemistry), tissue quality control (QC) was performed by a pathologist, and a small minority of cases in which the tissue core was insufficient for analysis (folded, minimal tissue, absent tissue) were excluded, blinded to experimental parameters such as marker expression, diagnosis, and clinical data. Whenever data was filtered to select a sub-population for analysis or a figure (e.g., primary IDH-wildtype glioblastoma) the procedure is detailed in the text, figure legend, or methods section, and additional sub-cohort data is described (see Supplementary Data). For single-cell immunofluorescence data, QC was performed as described in the Methods section subtitled "Single-cell data QC for t-CyCIF."

### Spatial analysis

Spatial correlation functions were evaluated as the Pearson correlation between pairs of cells as a function of their relative distance. Specifically, the Pearson correlation between two variables U and V was evaluated between a cell of type A and its $k$'th nearest neighbor of type B, and the average distance to $k$'th nearest neighbors was evaluated in micrometers. In each correlation function A and B were some combination of the cell-type classes assigned to each cell (e.g., Tumor and Myeloid, Tumor and All, Tumor and non-Tumor). U and V were either marker log-intensity values or a binary variable indicating whether a cell was or was not of a given type (e.g., CD73 and CD39 values, HIF1a

values, and Endothelial binary). In Supplementary Fig. S12a, A = Tumor, B = Myeloid, U = CD73, V = CD39.

Spatial correlation functions were evaluated for each core up to $k = 200$, and then fit to a single exponential with two parameters:

$$C(d) = c * \exp(b * d)$$

for the correlation $C(d)$ as a function of distance $d$. The fitted parameter $c$, correlation strength, was used as a summary of the spatial correlation in each core (Fig. 8G–I, L; Fig. S12a, d). If the 95% confidence interval of $c$ contained 0, $c$ was considered statistically insignificant and set to 0 for downstream analysis. 'High interaction' in Fig. 8J was defined as the condition that all TMA cores from a given patient having significant values for $c$.

### Reporting summary

Further information on research design is available in the Nature Research Reporting Summary linked to this article.

## Data availability

Data (publicly available and newly generated) used in the preparation of this manuscript are detailed in the Source Data files provided with the manuscript (see Table of Contents in Supplementary Data 12). The processed cyclic immunofluorescence (CyCIF) single-cell data feature tables and mass spectrometry data for the adult glioma, pediatric glioma, and mouse brain datasets are deposited and available at the Synapse repository: https://www.synapse.org/#!Synapse:syn30803357[63].

The processed cyclic immunofluorescence (CyCIF) single-cell data feature tables and mass spectrometry data for the adult glioma, pediatric glioma, and mouse brain datasets are deposited and available at the Synapse repository: https://www.synapse.org/#!Synapse:syn30803357[63].

The publicly available adult glioblastoma single-cell RNA sequencing data (Neftel et al.) are available in the Broad Single Cell Portal (https://singlecell.broadinstitute.org/single_cell/study/SCP393/single-cell-rna-seq-of-adult-and-pediatric-glioblastoma#study-summary)[41].

The publicly available adult glioma single-cell immune RNA-sequencing data (Pombo-Antunes et al.) are available in the Brain Immune Atlas (https://www.brainimmuneatlas.org/)[51].

The publicly available developing brain single-cell RNA-sequencing data (Nowakowski et al.) are available in the UCSC Cell Browser (https://cells.ucsc.edu/?ds=cortex-dev)[47].

The publicly available human microglia single-cell RNA-sequencing data (Sankowski et al.) are available in the human microglia browser (https://single-cell.shinyapps.io/human-microglia/)[50].

The publicly available processed TCGA data are available at https://www.cbioportal.org/[53].

The publicly available processed CPTAC/CBTTC pediatric glioma data are available at: http://pbt.cptac-data-view.org/[62].

The publicly available TCGA GBM bulk full-length RNA-seq processed by the Piccolo's team are available at: https://osf.io/gqrz9/files[85].

Mass spectrometry metabolite identities were cross-matched using the Metlin metabolite database (https://metlin.scripps.edu/landing_page.php?pgcontent=mainPage)[91].

Mass spectrometry data is available at the NIH Common Fund's National Metabolomics Data Repository (NMDR) website, the Metabolomics Workbench, https://www.metabolomicsworkbench.org where it has been assigned Project ID (PR001406). The data can be accessed directly via it's Project DOI: (https://doi.org/10.21228/M80421)[64].

Remaining data are available within the Article, Supplementary Information, Source Data, or Supplementary Data.

There are no restrictions on data access. Raw imaging data from cyclic immunofluorescence experiments are available from the

corresponding authors on request. These files are large and may require significant computational pre-processing for final data analysis. The processes used by the authors are delineated in the Methods section and/or code repository (https://github.com/labsyspharm/cd73_coy_spatialcorrelation), but interested parties should first consider using processed counts tables available in the Synapse repository (https://www.synapse.org/#!Synapse:syn30803357)[63]. Source data are provided with this paper.

## Code availability

Custom code used in this manuscript are available at the following repositories: https://github.com/labsyspharm/cd73_coy_spatialcorrelation, https://zenodo.org/record/6628875#.YqJVfqjMJD8. Digital Object Identifier (DOI) 10.5281/zenodo.6628875.

There are no restrictions on access. Further information and requests should be directed to the corresponding authors and will be fulfilled.

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

## Acknowledgements

We thank Dana-Farber/Harvard Cancer Center in Boston, MA, for the use of the Specialized Histopathology Core, which provided tissue processing services. The results shown here are in part based upon data generated by the TCGA Research Network: https://www.cancer.gov/tcga. The schematic in Fig. 9 was created with BioRender.com. This work was supported by NIH grants U54-CA225088 (PKS, SS), R01-CA194005 (S.S.), R01-CA188288 (K.L.L.), R01-CA219943 (K.L.L.), P01-CA163205 (K.L.L.), U54-CA210180 (N.Y.R.A.), P41-EB028741 (N.Y.R.A.), T32-GM008313 (SW), T32-GM007748 (S.C.), T32-EB025823 (S.A.S.) and T32-GM008208 (R.R.), by the Ludwig Center at Harvard (P.K.S., S.S.), INCa, Fondation ARC and Ligue grant INCa-ARC-LIGUE_ID_16280 (M.T.), a Capital Award from the Massachusetts Life Sciences Center (N.Y.R.A.), by the Specialized Histopathology Core Facility of the Dana-Farber/Harvard Cancer Center (P30-CA06516), and by National Metabolomics Data Repository - Nextgen Metabolomics Workbench (U2C-DK119886). K.L.L. received support from the National Brain Tumor Society (NBTS) and Accelerate Brain Cancer Cure (ABC2).

## Author contributions

S.C., M.T., and S.S. developed the concept for the study. J.R.L., R.R., C.Y., and C.C.R. collected image data and performed analysis in collaboration with S.C. and S.W. S.A.S., M.R., G.B. and N.Y.R.A. designed, performed, and analyzed mass spectrometry experiments in collaboration with S.C., G.J.B., and S.S. S.C. and P.K. collected and analyzed single cell RNA-sequencing data. L.S. performed and analyzed T.C.G.A. in collaboration with M.T., S.C., and S.S. T.D.R., M.S., and K.A.C. collected and analyzed C.B.T.T.C. pediatric tumors and data in collaboration with S.C. and S.S. J.H. collected and analyzed clinical and demographic data in collaboration with S.C. All authors wrote and/or edited the manuscript. S.S., P.K.S., N.Y.R.A., P.Y.W., P.B., and K.L.L. provided supervision and funding.

## Competing interests

M.T. has a consulting or advisory role with Agios Pharmaceutical, Integragen, and Taiho Oncology, and research funding from Sanofi. P.K.S. is a member of the SAB or BOD member of Applied Biomath, RareCyte Inc., and Glencoe Software; P.K.S. is also a member of the NanoString SAB and a consultant for Montai Health and Merck. In the last five years the Sorger lab has received research funding from Novartis and Merck. KLL is supported by Eli Lilly and BMS, consults for BMS, Integragen, Travera LLC, is on the SAB of Integragen and Rarecyte, and holds equity in Travera LLC. P.B. receives grant funding from Novartis Institute of Biomedical Research and Deerfield, and consults for QED Therapeutics, for unrelated projects. P.Y.W. has research support from Agios, Astra Zeneca, Medimmune, Celgene, Eli Lilly, Genentech, Roche, Kazia, MediciNova, Merck, Novartis, Nuvation Bio, Chimerix, Vascular Biogenics, and VBI Vaccines. P.W. is on the advisory boards of Agios, Astra Zeneca, Black Diamond, Boston Pharmaceuticals, Chimerix, CNS Pharmaceuticals, Elevate Bio, Imvax, Karyopharm, Merck, Mundipharma, Novocure, Novartis, Nuvation Bio, Prelude Therapeutics, Vascular Biogenics, VBI Vaccines, Voyager, and QED. RareCyte manufactured instruments used for tissue imaging and Glencoe developed the OMERO Database used for image informatics. P.K.S. and S.S. declare that none of these relationships have influenced the content of this manuscript. The remaining authors declare no competing interests.
