## [Peer Review File · Nature Communications]

Single Cell Spatial Analysis Reveals the Topology of Immunomodulatory Purinergic Signaling in GlioblastomaReviewers' Comments:

Reviewer #1:

Remarks to the Author:

This is an interesting study that beautifully integrates various single-cell multi-omic modalities to assess the significance of CD73 and CD39 expression in glioblastoma (GBM) progression. The results that are presented are novel and extensive and combined form a strong argument for the importance of purinergic signaling in GBM. One of the highlights is their multiplexed spatial analysis. The article is well-written and has a logical flow.

Specific remarks:

1) The authors rely on published scRNA-seq datasets to assess the expression of CD73 and CD39 in GBM tumors. However, I feel that especially for CD39 expression in macrophages/immune cells, the analysis can be more fine-grained and informative. The authors primarily mention and consider microglia-derived tumor macrophages. It is true that this is the largest fraction in newly-diagnosed GBM tumors. However monocyte-derived macrophages also form a functionally important component, which furthermore strongly increase upon recurrence. Is CD39 differentially expressed between these two ontogenetically distinct populations? Furthermore, is CD39 differential in tumor macrophage populations that exhibit specific activation states? And what about CD39 in other tumor infiltrating immune cells, for example dendritic cell subsets or immunomodulatory Tregs? The authors could rely on the recently published scRNA-seq and CITE-seq datasets of human newly-diagnosed and recurrent GBM (Antunes et al. 2021 10.1038/s41593-020-00789-y) to investigate this.

2) On page 12 the authors mention:

"Further sub-classification of CD39- expressing microglia (which display a strong spatial association with CD73-high tumor cells) indicates that they exhibit a pro-inflammatory state highly correlated with hypoxia in glioma tissue."

I am not convinced that the data the authors present justifies this claim. The authors present no direct evidence that CD39 expression on microglia correlates with hypoxia.

3) The authors suggest that hypoxia may be one of the drivers of CD73 expression. However, work by Neftel et al. has indicated that the mesenchymal state (and not the astrocyte-like state which the authors show to correlate with CD73) is linked to hypoxia. The author should comment on this.

Reviewer #2:

Remarks to the Author:

Immunosuppression of glioblastoma has been one of the major challenges to treat this deadly cancer, and the immunomodulatory role of CD73 and CD39 has been extensively studied in multiple cancers, including glioblastoma. In this study, Coy et al. set out to define the expression of CD73 in glioblastoma using a wide variety of databases and patient specimens. Also, the authors have taken cutting-edge technologies, such as scRNA-seq, multiplexed tissue imaging, and t-CyCIF. While the changes in CD73 expression in cancers have been well known, the authors found the spatial colocalization of CD73-expressing glioblastoma and CD39-expressing microglia by the t-CyCIF imaging of 172 glioblastoma specimens, concluding that these newly found populations are likely corresponding to the immunosuppressive feature of glioblastoma microenvironment. This is a dense, well-articulated work. On the other hand, a mechanistic insight and functional outcome (i.e., immunosuppression) are lacking.

Major points:

1. There is a number of enzymes, such as adenosine deaminase/CD26, ATP transporter, and ENTs, and receptors involved in extracellular purinergic signaling. While the finding is potentially exciting, functional relevance in GBM immunosuppression is unclear.

2. Related to the above, the metabolomics analysis (Figure 4), in which levels of adenosine and the other adenylates are higher in the high-CD73 tumor, maybe the most relevant one. However, I believe that the detected signals are mostly from intracellular metabolites, not from extracellular space metabolites, based on the nature of these methods. Also, to me, the data that show 3-5 folds increases in ATP and ADP is a bit surprising since these are highly abundant metabolites.

3. While I appreciate the immense work and dense data, Fig 1-3 are somewhat distractive and repetitive of the previous work. For example, the authors use cell line analysis (Figure 1) regardless of mentioning its uncertainty in the introduction. Figure 2 lacks the correlation analysis of CD73 and CD39. Fig 2E partly shows it, but it is not correlated within the same sample. While CD73 protein expression in multiple brain tumors was assessed in Figure 3, there is no analysis on CD39. Another minor concern in Figure 3 is about the validity of the monoclonal anti-CD73 antibody (Abcam: EPR6115) used for the IHC analysis. Has this antibody ever been validated about its specificity in the setting of IHC using proper controls, such as CD73 knockout/knockdown cells? This antibody has been discontinued by the company. For the spatial analysis, the authors seem not to take advantage of using the IvyGAP database, which would be another key resource that could further verify their findings.

Minor points:

- Each figure lacks a figure number. It was hard to read.
- The results of RNAi analysis should be further verified to eliminate the possible off-target effect.
- Page 8, CD73 is also described as NT5E. It was confusing.
- Page 8, I am not convinced that EGFR and CD73 mRNA is significantly correlated with $r=0.17$ and $p=0.03$.
- Fig. 5C is difficult to interpret based on the data comparison to tonsil. Probably, the authors should use more proper control.
- Page 11: Expression of HIF1 does not mean that the cells were in hypoxic condition, and there could be a number of reasons.

Reviewer #3:

Remarks to the Author:

This manuscript combines multiplexed tissue imaging, single-cell RNA sequencing, and mass spectrometry imaging to characterize the composition and spatial organization of human glioblastoma components. The authors have performed a thorough and systematic exploration of extracellular purinergic signalling in glioblastoma and its clinical importance, and present some interesting and compelling findings. I have a few minor comments for amendment or clarification.

1. In Figure 1A, there doesn't seem to be much visual distinction between glioblastoma and many of the other subtypes. This figure might benefit from some statistics to demonstrate that glioblastoma did in fact exhibit high expression of CD73 in comparison to other tumor sub-types.

2. Please add additional figure legends where necessary. For instance, in Figure 1b, what do the blue and red markers denote? In Figure 1C, what do the difference box plots relate to? Such details are not clear in some of the figures.

3. Please clarify the mass spectrometry instrumentation used for MS imaging. The results specify timsTOF MS was used, however the methods only specify details for FT-ICR MS. If these techniques

were coupled, please clarify and provide more detail.

4. In the methods section, the authors state that metabolite identification following MS analysis was only performed putatively. If no analyte identification was performed beyond database matching (i.e. using analytical standards or MS/MS), the authors should be cautious about claiming identifications. It should perhaps be made clearer in the main body of the manuscript that metabolite identifications have not been confirmed. If further steps were taken to identify metabolites, details should be provided.

5. When discussing Figure 6, the authors state that CD73-expressing SOX2+ tumor cells clustered in close-proximity with CD39-expressing PU.1+/CD163+ myeloid cells was observed in "multiple samples". If pertinent, what proportion of samples was this observed in – it is unclear whether this was observed in just few samples or the majority?

Reviewer #4:

Remarks to the Author:

Coy et al. presented an immunosuppressive role of CD39 and CD73 in Glioblastoma. Specifically, tumor CD73 and microglial CD39 were close to each other and their proximity was associated with poor outcome.

1) CD73/CD39/Adenosine have been shown to be linked to immunosuppressive microenvironments in gliomas/glioblastomas many times:

<https://www.ncbi.nlm.nih.gov/pmc/articles/PMC7526457/>

<https://www.sciencedirect.com/science/article/pii/S1471489220300394>

<https://www.deepdyve.com/lp/oxford-university-press/synergy-between-the-ectoenzymes-cd39-and-cd73-contributes-to-vyckmtAd9f>

and many others

2) Because this is a well known mechanism in immunosuppression, the findings were not surprising. The immunohistochemistry and conventional immunofluorescence should be able to provide the sufficient validation of the CD39/CD73 markers in glioblastomas, along with the spatial analysis.

3) Multiplexing proteins by CyCIF was presented in two figures, but they are already established techniques. The analysis of spatial correlations and maps are not novel and only limited to short four paragraph descriptions. Thus, the power of multiplexing is underutilized here and spatial bioinformatics methods do not yield novel insights that can solve the mechanism of CD73/CD39.

Overall, this paper is a conventional pathology study of two biomarkers and the findings remain within the limits of conventional pathology.

Thus, this paper is not suitable for broad interest of Nature Communications, instead a topical pathology, oncology, or neuroscience journal would be better fit to the scope of the presented work.

DETAILED RESPONSE TO REVIEW: NCOMMS-21-27694

“Single Cell Spatial Analysis Reveals the Topology of Immunomodulatory Purinergic Signaling in Glioblastoma”

Detailed Response to Review

General Comments (GC).

The comments and critiques from the editor and reviewers were invaluable in guiding a major revision of the manuscript. In this General Comments section, and below in a point-by-point response, we describe the substantial additional data and analyses, new figures, and text revision that we have completed to strengthen the study.

Although literature reports have implicated purinergic signalling in the pathophysiology of glioblastoma, they have not provided a detailed description of the cell types and states that contribute to this pathway, nor the spatial organization of these components. The arrangement of tumor and other cell types in physical space is increasingly recognized to mediate tumorigenesis and immune responses and may play an important role in purinergic signalling. Our study integrates complementary data from several distinct experimental modalities including single-cell datasets from multiple independent institutions, and large cohorts of human tumor samples from a broad clinicopathologic cross-section to provide insights into the functional communities of tumor and immune cells present in adult and pediatric high-grade gliomas. We have further strengthened the study with the following additions:

1. Additional single cell RNA-sequencing analysis of tumor and immune populations (Figs. 1, 2)

A valuable suggestion was to perform further analysis of scRNA-seq data to more precisely characterize the cell types and states in glioblastoma that express extracellular purinergic pathway components and better understand the factors that modify their expression. We pursued this objective in multiple ways according to reviewer feedback.

- i.) Reviewer #1 suggested that we leverage the rich and well-annotated scRNA-seq data recently published by Pombo Antunes *et al.* (Nature Neuroscience. 2021 Apr;24(4):595-610.) to more precisely define myeloid and other immune cell populations and correlate functional sub-groups with purinergic pathway marker expression (e.g., *CD39*, *CD73*). This dataset allows for detailed analysis of a wide variety of immune cell sub-types, though unlike our previously reported data it does not include tumor or other cell populations such as oligodendroglia. We now present this analysis in Figure 2d-m and Figure S3a-m. This analysis supported our prior findings that *CD73* expression is present in very few immune cells, and only rare tumor-associated myeloid cells, in contrast with prior literature reports (Nature Medicine. 2020 Jan;26(1):39-46.). These data also supported our hypothesis that *CD39* expression is most strongly associated with tumor-associated myeloid populations, particularly microglia rather than infiltrating peripheral macrophages, with smaller numbers of *CD39*-expressing lymphoid cells. We also show new evidence that *CD39* is also expressed by dendritic cell populations, particularly classical dendritic cells, though these are less numerous than microglia or macrophages. Additionally, this dataset reported that recurrent tumors have a shift in immune population relative to newly diagnosed tumors, with a greater influx of peripheral cells in recurrent tumors. With regard to purinergic signalling, the data shows a greater influx of *CD39*-expressing T regulatory cells and peripheral macrophages. These findings suggest that while *CD39* is principally expressed by tumor-associated myeloid cells, there may be shifts in the composition of *CD39*-expressing cells in tumors that have undergone chemotherapeutic or immunomodulatory treatment.

- ii.) In our first submission we analysed scRNA-seq data from a separate dataset including both tumor and immune cells from 21 adult glioblastoma and 7 pediatric high grade gliomas (Neftel *et al.* Cell. 2019 Aug 8;178(4):835-849.e21.) and showed that *CD73* is most enriched in tumor cells with astrocyte-like (AC-like) differentiation. We extend this analysis to show that *CD73* expression in different age-groups is most commonly linked to two lineage states. In adult IDH-WT glioblastoma, *CD73* is correlated with *EGFR* expression and AC-like differentiation, whereas in pediatric gliomas *CD73* expression is correlated with *PDGFRA* expression and oligodendrocyte-precursor-like (OPC-like) differentiation (Fig 1d-g). Similar expression of these ectoenzymes is found in developing astrocytes and oligodendroglia (Fig. S2) supporting that tumor cells may exhibit lineage-differentiation mirroring developmental states and *CD73* expression may at least partially be a feature of this lineage program. Additionally, we show that less common MES-like tumor cells with elevated *CD73* expression often co-express high levels of hypoxia-response genes (Fig. S1h, i), providing further evidence supporting hypoxia as a modulator of *CD73* expression as requested by multiple reviewers.

2. Additional characterization of tumor microenvironmental properties related to *CD73* levels (Fig. 3)

The Reviewers' suggestions to better characterize the tumor microenvironment (TME) of glioblastoma in relation to the purine pathway prompted us to analyse mRNA-sequencing data from 168 samples from The Cancer Genome Atlas (TCGA) database to examine the status of inflammatory signalling pathways (Nature Communications. 2021 Jan 13;12(1):352.) as well as the population abundance of tumor and tumor-infiltrating immune populations using gene expression signatures derived from scRNA-seq datasets relative to *CD73* and/or *CD39* expression (Fig. 3a-d). Multiple immune populations including myeloid (monocyte, macrophage, microglia), DC (pre-DC, migratory DC, and plasmacytoid DC), and T-cell populations were enriched in *CD73*^{hi} versus *CD73*^{lo} glioblastoma, as were inflammatory signatures for chemokine, T cell inflammation, MHC-1, and IFN γ transcriptional programs. Glioblastoma with combined elevation of *CD73* and *CD39* (i.e., *CD73*^{hi}/*CD39*^{hi}) were enriched for immune populations, while those with low levels of both enzymes (*CD73*^{lo}*CD39*^{lo}) were depleted for nearly all immune cell populations. Notably, 'double-high' tumors had even greater elevation of immune cell types and inflammatory programs, suggesting that purinergic enzyme expression may be modified by the underlying state of the tumor immune microenvironment, with potential combinatorial effects of these enzymes in mediating or adapting to immune activity in glioblastoma. Given the strongly immunosuppressive functions of *CD73* and adenosine in most tissues, and evidence we present of both being present at high levels in glioblastoma, elevated inflammation in *CD73*-enriched tumors was somewhat surprising. This may reflect that *CD73* is upregulated by tumor cells in response to immune activity, or that *CD73* expression provides a selective advantage to tumor cells in the face of immune activity, rather than *CD73* and adenosine outright pre-empting and suppressing immune responses.

3. Analysis of pediatric high-grade gliomas (Figs. 4, 5, 8).

To better understand the pediatric tumor cell population with high *CD73* expression that was identified by scRNA-seq and to determine whether *CD73* and *CD39* co-localization into functional communities is a general concept in glioma, we analysed a new cohort of 102 pediatric brain tumor specimens from 69 patients that were made available to us through the generosity of the Children's Brain Tumor Network (CBTN). We provide clinicopathologic (Fig. 4c,d, Fig. 5f-l, Fig. S8) and spatial (Fig. 8k,l, Fig. S12d,e) characterization of purine signalling in this cohort which demonstrates that:

- i.) A large subset of pediatric high-grade gliomas (HGG) express high levels of *CD73* similar to adult glioblastomas, including H3K27M-mutant diffuse midline gliomas which are invariably

- lethal in children and young adults. Our study is the first analysis of key purinergic pathway regulators such as CD73 and CD39 in multiple important pediatric tumor categories.
- ii.) CD73 levels are significantly correlated with *CD39* and *PDGFRA* by mRNA and protein levels (IHC and mass spectrometry measurements) in pediatric tumors. Pediatric HGG also exhibit a similar spatial association between CD73-expressing tumor cells and CD39-expressing microglia. Demonstration of strong CD73 expression and correlation of tumor CD73 with myeloid CD39 in both pediatric and adult HGG suggests that this may be a generalized organizational feature of gliomas and increases the importance and potential clinical impact of our study by implicating these tumors as additional potential therapeutic targets.

4. Pan-cancer analysis of CD73 and purine pathway marker expression

As requested by the reviewers, we show that adult glioblastomas exhibit statistically significantly higher levels of *CD73* mRNA than many other cancer subtypes in the TCGA pan-cancer dataset, with levels similar to several other cancers which are showing promising responses to anti-CD73 therapy in clinical trials (e.g., pancreatic adenocarcinoma, NSCLC) (Fig. 4a). We also report new analyses showing that other components of the purinergic signalling pathway, including *CD39* and several adenosine receptor subtypes (*ADORA1/A1R*, *ADORA3/A3R*) are also highly expressed in gliomas compared to other cancers (Fig. S5). CD73 and CD39 may represent attractive biomarkers for further clinical trial development, including with our IHC scoring system which was validated by single-cell expression of CD73 by multiplexed immunofluorescence, and robustly correlates with mRNA and mass spectrometry measurements of CD73 expression (Fig. 4, Fig. 7, Fig. S8). Since the initial submission of our paper, additional data from the COAST non-small cell lung cancer (NSCLC) trial was reported (NCT03794544), showing a promising effect of combination durvalumab (anti-PD1) and oleclumab (anti-CD73) in significantly lengthening progression-free survival compared to PD-1 inhibitor (durvalumab) alone (the current standard of care). These findings likely significantly increase interest in the development of anti-CD73 therapies in additional cancer indications, and our findings may support exploration of trial opportunities in both adult and pediatric HGG, as well as suggesting avenues for future investigation in meningioma and low-grade glioma which also express high levels of CD73.

5. Analysis linking hypoxia with CD73 induction

In response to comments from multiple reviewers, we sought to further analyse the relationship between hypoxia and CD73 expression in glioma. The manuscript now reports the following additional analyses:

- i.) Previous reports of scRNA-seq data from the dataset of 21 adult and 7 pediatric gliomas (Fig. 1) showed that expression of hypoxia-response genes (e.g., *HILPDA*) are limited to the MES-like differentiation state (Cell. 2019 Aug 8;178(4):835-849.e21.). While most tumor cells which strongly express *CD73* exhibit an AC-like or OPC-like differentiation-state, corresponding to adult and pediatric HGG respectively, we noted a smaller subset of cells in the MES-like cluster. We now show that many of these cells exhibit strong expression of hypoxia-response genes, including cells without expression of *EGFR* or *PDGFRA*, supporting that hypoxia may be an additional regulator of *CD73* in a subset of cells (Fig. S1h,i).
- ii.) While *CD73* is not commonly expressed in immune populations in our analysis (Fig. 1,2), new scRNA-seq analysis of immune sub-groups shows that hypoxic myeloid populations comprise many of the immune cells with *CD73* expression in glioblastoma (Fig. 2, Fig. S2), suggesting that hypoxia may also lead to increased *CD73* expression in a subset of immune cells.

- iii.) As suggested by the reviewers, we leveraged microregional spatial transcriptomic and in situ hybridization data from the IVY Glioblastoma Atlas Project (IVY GAP) resource to better substantiate a link between hypoxia and induction of CD73. This data showed that *CD73* and hypoxia-response genes are strongly expressed in regions of palisading necrosis and peri-necrotic tumor, with significantly greater *CD73* in these regions compared to non-necrotic cellular tumor regions, providing additional spatial evidence for a link between hypoxia and *CD73* in tumor tissue (Fig. 7f-i).
- iv.) In our initial submission we had shown a spatial correlation between tumor CD73 expression and expression of HIF1A in adult glioblastoma. We now report the analysis of a cohort of pediatric HGG, including H3K27M-mutant midline gliomas, showing a spatial association between tumor CD73 and additional hypoxia-response genes such as GLUT1, iNOS, and ARG1. Furthermore, we find that such associations appear to be correlated with palisading or peri-necrotic tumor regions in some cases (Fig. S12d,e).

Collectively, these data further support the presence of a spatially influenced association between CD73 expression and hypoxia in multiple tumor subtypes.

6. Spatial organization of CD73 and CD39 (Fig. 8).

In response to reviewer feedback, we sought to further expand our analysis of spatial relationships between tumor CD73 and aspects of the tumor microenvironment.

- i.) In the first case, we expand our spatial analysis by using Latent Dirichlet Allocation (LDA), a modelling method that can be used to identify recurrent cellular neighbourhoods in tissues. This analysis supports the presence of a positive spatial correlation between CD73 tumor cells and myeloid cell components expressing PD-L1 and CD39 (please see comments to Reviewer #4 for a detailed description of the LDA method and the results).
- ii.) We expand our analysis of spatial correlations to a new cohort of pediatric HGG. These data show that pediatric HGG also exhibit a strongly significant correlation between tumor CD73 and myeloid CD39 expression, as well as myeloid CD163 and CD11b, supporting that a similar interaction of CD73-expressing tumor cells and inflammatory microglia with CD39 expression exists in multiple tumor subtypes.
- iii.) Throughout the manuscript, we sought to further elaborate and clarify the methods and their significance for understanding spatial relationships among purinergic pathway components in glioblastoma and pediatric HGG. Generation and computational processing of multiplexed immunofluorescence data to define cell populations and examine functional states by single-cell and neighbourhood analyses as well as spatial statistics approaches is a relatively recent advance, and such methods have not been applied at this scale to glioblastoma previously. Identification of a synergistic spatial correlation between CD73 and CD39 on distinct neoplastic and non-neoplastic cell populations which is present in both adult and pediatric gliomas advances our understanding of the organization of the glioma microenvironment, and potentially other cancers if such spatial correlations are a generalized principle of purinergic pathway activity and may help to partially explain the poor efficacy of immunotherapeutic approaches in glioblastoma to date.

7. Experiments to address key technical considerations (Fig. 6, Fig. S10).

The Reviewers highlighted several technical areas that required more attention which we now address with a series of additional experiments.

- i.) The reviewers noted that the antibody clone used in the multiplexed immunofluorescence (CyCIF) experiment was unexpectedly discontinued by the vendor, and we had not initially presented evidence validating this marker's specificity for CD73. To further validate this marker and qualify other anti-CD73 clones for potential use in future experiments, we performed an additional immunofluorescence experiment and correlated multiple anti-CD73 antibodies at a pixel-by-pixel level as previously described (Nature Protocols 14, 2900–2930 (2019)). This showed an excellent correlation in signal between CD73 antibodies, but not unrelated markers (Fig. S10a). We also note that several of these antibody clones were knockout validated by the vendor (EPR6114, D7F9A), or in associated publications (D7F9A) (PMID: 33086655, 33609609, 28158983), further supporting the specificity of antigen binding.
- ii.) Reviewer #2 noted that the mass spectrometry methods used in the paper cannot distinguish between extracellular and intracellular adenosine. While this is indeed the case that current technology and methods cannot resolve the localization of metabolites such as adenosine at a sufficient resolution to resolve extracellular and intracellular localization, the extracellular localization of CD73 enzymatic activity and enrichment of adenosine in tumors with high CD73 levels relative to those with low levels suggests that the observed increase in adenosine levels is extracellular resulting from CD73 enzymatic activity (Fig. 6c). To further understand whether adenosine levels correlate with CD73 expression in other settings, we also examined adenosine and CD73 levels in whole-mount mouse brain, showing a strong spatial correlation between adenosine levels and CD73 expression. While we cannot definitively rule out a contribution to observed adenosine signal from intracellular stores, particularly given the expression of equilibrative nucleoside transporters (ENT1-4) on cells in the tumor microenvironment (Fig. 1b), the tight spatial association with CD73 suggests that at least some, and likely a substantial part, of adenosine signal is derived from extracellular CD73 enzymatic activity and accordingly likely affects adenosine receptor signalling which mediates downstream immunomodulatory effects.

Please note that we have substantially revised the text and the figures so that we could incorporate the following additional experiments:

- i.) scRNA-seq analysis of the glioblastoma immune dataset generated by Pombo Antunes *et al.* (Fig. 2).
- ii.) Detailed analysis of cell populations and inflammatory states in the glioblastoma immune microenvironment (Fig. 3).
- iii.) Clinicopathologic (Fig. 4, 5, S8) and spatial (Fig. 8, S12) analysis of a pediatric tumor cohort, including many tumor types which have not been analysed previously.
- iv.) Additional analyses of the correlation between purinergic pathway markers such as CD73, immune populations, and hypoxia (Fig. 1,2,7,8, Fig. S12).
- v.) Additional spatial analysis of adult glioblastoma (Fig. 8).
- vi.) Additional experiments to address key technical considerations (Fig. 6, Fig. S10).

Specific changes and a point-by-point rebuttal are provided below.

Reviewer #1, expert in sc-RNAseq and spatial methods/GBM/microglia

Remarks to the Author:

This is an interesting study that beautifully integrates various single-cell multi-omic modalities to assess the significance of CD73 and CD39 expression in glioblastoma (GBM) progression. The results that are presented are novel and extensive and combined form a strong argument for the importance of purinergic signalling in GBM. One of the highlights is their multiplexed spatial analysis. The article is well-written and has a logical flow.

1) The authors rely on published scRNA-seq datasets to assess the expression of CD73 and CD39 in GBM tumors. However, I feel that especially for CD39 expression in macrophages/immune cells, the analysis can be more fine-grained and informative. The authors primarily mention and consider microglia-derived tumor macrophages. It is true that this is the largest fraction in newly-diagnosed GBM tumors. However, monocyte-derived macrophages also form a functionally important component, which furthermore strongly increase upon recurrence. Is CD39 differentially expressed between these two ontogenetically distinct populations? Furthermore, is CD39 differential in tumor macrophage populations that exhibit specific activation states? And what about CD39 in other tumor infiltrating immune cells, for example dendritic cell subsets or immunomodulatory Tregs? The authors could rely on the recently published scRNA-seq and CITE-seq datasets of human newly-diagnosed and recurrent GBM (Antunes et al. 2021 10.1038/s41593-020-00789-y) to investigate this.

These are excellent suggestions that provided us with an opportunity for further characterizing CD39 and CD73 expression in multiple immune lineages and functional states in the glioblastoma tumor microenvironment. The manuscript by Pombo Antunes, Movahedi and colleagues (*Nature Neuroscience* 2021) was published shortly before we submitted our manuscript for review, and it was unfortunately not included in our analyses. In this revision we now leverage the substantial data made available to the public to more broadly and precisely assess the immune states associated with purine signalling.

This new analysis reveals the following:

1. Similar to our prior analyses, CD73 expression was detectable in very few immune cells in newly diagnosed tumors (0.5% of immune cells) (Fig. 2e,f).
2. CD39 expression was more widespread (38.9% of immune cells) and present in subsets of all cell types, although the strongest and most frequent CD39 expression was seen in tumor-associated microglia (TAM2). Other myeloid populations also expressed CD39 in a substantial number of cells, including dendritic cells (DC), with smaller numbers of CD39-expressing lymphoid cells of any type (B,T,NK cells) in newly diagnosed tumors (Fig. 2g,h).
3. Functional sub-group analysis of myeloid cells again showed little CD73 expression (0.4% of myeloid cells). The few weakly CD73 expressing cells were found mainly in hypoxic and transitory monocyte populations (Fig. 2j,k), suggesting that hypoxia may modulate CD73 expression in immune as well as tumor cells.
4. CD39 was broadly expressed by all myeloid sub-groups (44.9% of myeloid cells), though there was enrichment of CD39^{hi} cells in the microglial TAM cluster (Fig. 2l,m).
5. CD73 expression was not detected in any dendritic cells (0%), while CD39 was expressed by multiple dendritic sub-groups (41.9% of dendritic cells), though expression was more common in classical dendritic cells, and less common in plasmacytoid dendritic cells (pDC) and migratory DC (migDC) (Fig. S3l,m)

6. In recurrent tumors, *CD73* is still expressed by few immune cells, most prominently hypoxic monocytic populations (Fig. S3b,c,g,h). Recurrent tumors still show the greatest enrichment of *CD39* in microglial populations. As described in Pombo Antunes *et al.*, there is a shift in immune populations with greater numbers of peripherally derived myeloid and lymphoid cells. Such peripherally derived macrophages (TAM1) also exhibit *CD39* expression. Most lymphoid cells do not express *CD39*, and the increased numbers of lymphoid cells in recurrent tumors reduce the overall percentage of *CD39* positive cells. However, there is a greatly expanded population of *CD39*-expressing T regulatory cells which may presumably contribute to increased purine catabolism and immunosuppression in recurrent lesions.

Revisiting the single cell transcriptomic data also allowed us to improve upon the previously included scRNA-seq analysis so that we better delineate the connections of cell lineage-differentiation and clinicopathologic features of tumors (Fig. 1d-g, Fig. S1b-d). Importantly, we further characterize a second cluster of *CD73*-expressing tumor cells in the OPC-like lineage and show that AC-like and OPC-like clusters correspond to adult and pediatric high-grade gliomas. This prompted a further analysis of a large cohort of pediatric high-grade gliomas which we describe subsequently. Additionally, we further analysed the connection between *CD73* and hypoxia-response genes in this dataset (Fig. 1h-j, Fig. S1h,i) (see also point #1 in the General Comments above). This analysis showed that while *CD39*-expressing cells are less common in the MES-like differentiation-state, many correspond to cells with high expression of hypoxia-response genes, providing further evidence that hypoxia may modulate *CD73* expression in a sub-population of cells.

2) On page 12 the authors mention:

"Further sub-classification of CD39- expressing microglia (which display a strong spatial association with CD73-high tumor cells) indicates that they exhibit a pro-inflammatory state highly correlated with hypoxia in glioma tissue."

I am not convinced that the data the authors present justifies this claim. The authors present no direct evidence that CD39 expression on microglia correlates with hypoxia.

On review of the data, we agree with the reviewer that while the myeloid expression signature derived from spatial analysis corresponds to a microglial population in the Sankowski *et al.* dataset which may be associated with expression of hypoxia-response genes, we do not find other strong evidence of a connection between hypoxia and *CD39* expression. In the micro-region spatial transcriptomic data from the IVY GAP dataset, we show that *CD39* levels do not specifically correlate with hypoxia, with stronger expression in regions of microvascular proliferation than the peri-necrotic tumor regions which are enriched for hypoxia-response genes and *CD73* (Fig. 7f-i). We have accordingly modified the description in the results section, and now write: "While *CD39* was broadly expressed across myeloid and microglial clusters in the scRNA-seq data, mapping a signature characterized by combined expression of *CD39*, *CD163*, and *CD11b* in this dataset revealed strong enrichment for a specific tumor-associated microglial subtype (Cluster 14 from Sankowski *et al.*) that is characterized by inflammatory, interferon, and hypoxia-response genes (Fig. S12b)."

3) The authors suggest that hypoxia may be one of the drivers of CD73 expression. However, work by Neftel et al. has indicated that the mesenchymal state (and not the astrocyte-like state which the authors show to correlate with CD73) is linked to hypoxia. The author should comment on this.

We appreciate the reviewer's insightful comments, which prompted us to further analyze the lineage-differentiation data in the scRNA-seq dataset. Most cells in the MES-like differentiation-state express very low levels of *CD73* (Fig. 1d). Neftel *et al.* previously described that the MES-like state includes a population of cells associated with high expression of hypoxia-response genes such as *HILPDA*, *LDHA*, and *DDIT3*. t-SNE mapping of *HILPDA* confirms this to be the case, with tight clustering in the MES-like state (Fig. S1h). Close inspection of the data shows that while few cells in the MES-like state strongly express *CD73*, those that do often exhibit simultaneous strong expression of hypoxia-response genes such as *HILPDA*, and these cells are often negative for the AC-like and OPC-like genes *EGFR* and *PDGFRA* (Fig. S1i). As described by Neftel *et al.*, the MES-like state is heterogenous and includes a subset of cells with elevated expression of hypoxia-responsive genes and another subset of cells without this expression signature. Thus, in addition to the larger clusters in the AC-like and OPC-like states, corresponding to adult and pediatric high-grade gliomas, a small subset of tumor cells (both adult and pediatric) acquire a MES-like differentiation-state associated with expression of hypoxia-responsive genes, implicating hypoxia as an additional modulator of *CD73* expression in glioma.

We also present additional evidence that supports a link between hypoxia and *CD73* expression in glioma:

- i.) Analysis of scRNA-seq data from CD45-positive immune populations (Pombo Antunes *et al.*) shows that while *CD73* expression is rare in the immune cells from glioblastoma, hypoxic monocytic populations comprise the largest subset of the small fraction of *CD73* expressing immune cells.
- ii.) We had previously included a spatial statistics analysis of CyCIF multiplexed imaging data showing a spatial correlation of HIF1A and *CD73* (Fig. 8h). We now additionally present spatial statistics analysis of pediatric high-grade gliomas, showing that tumor *CD73* expression correlates with expression of multiple hypoxia-responsive genes in these tumors (Fig. S12d), and this expression appeared to be similarly correlated with peri-necrotic regions (Fig. S12e). However, this data does not show a significant spatial correlation with HIF1A, unlike adult glioblastomas.
- iii.) GSEA analysis of scRNA-seq (Fig. 1h-j) that shows that a hypoxia gene signature is a significantly enriched in tumor cells with high level *CD73* expression compared to those with low expression.
- iv.) Prompted by a suggestion from Reviewer #2, we also took advantage of data from the Glioblastoma Atlas Project (IVY GAP) resource to further assess the relationship of hypoxia and *CD73* levels. RNA-seq data generated from 7 micro-dissected regions from 10 glioblastomas (including the leading tumor edge, infiltrating tumor, cellular tumor, peri-necrotic tumor, palisading tumor cells, hyperplastic vasculature, and microvascular proliferation). A heat map of this data (Fig. 7f) shows high expression of *CD73* in peri-necrotic and palisading regions which are characteristically hypoxic (confirmed by the enrichment of multiple hypoxia markers in these regions). We further confirmed this finding in situ hybridization (ISH) data which showed expression in peri-necrotic and palisading tumor regions (Fig. 7g-h).

Thus, several additional analyses provide compelling evidence that *CD73* expression is regulated by hypoxia in human tumors as well as lineage-differentiation and associated genomic changes such as *EGFR* amplification. We clarify these points in the manuscript text and figures and provide further context in the discussion.

Reviewer #2, expert in GBM and metabolism

Remarks to the Author:

Immunosuppression of glioblastoma has been one of the major challenges to treat this deadly cancer, and the immunomodulatory role of CD73 and CD39 has been extensively studied in multiple cancers, including glioblastoma. In this study, Coy et al. set out to define the expression of CD73 in glioblastoma using a wide variety of databases and patient specimens. Also, the authors have taken cutting-edge technologies, such as scRNA-seq, multiplexed tissue imaging, and t-CyCIF. While the changes in CD73 expression in cancers have been well known, the authors found the spatial colocalization of CD73-expressing glioblastoma and CD39-expressing microglia by the t-CyCIF imaging of 172 glioblastoma specimens, concluding that these newly found populations are likely corresponding to the immunosuppressive feature of glioblastoma microenvironment. This is a dense, well-articulated work. On the other hand, a mechanistic insight and functional outcome (i.e., immunosuppression) are lacking.

Major points:

1) There is a number of enzymes, such as adenosine deaminase/CD26, ATP transporter, and ENTs, and receptors involved in extracellular purinergic signalling. While the finding is potentially exciting, functional relevance in GBM immunosuppression is unclear.

We appreciate the positive comments from the reviewer. With new data and a substantially rewritten text, we hope that we now convey the importance of using information from multiple modalities to construct a more complete understanding of the landscape of immunoregulatory purinergic signalling in glioblastoma. Analysis of scRNA-seq data indicates that cell type and lineage correlate with expression of key components of the purinergic signalling pathway, including core regulatory enzymes and receptor subtypes (Fig. 1). In response to the reviewer's comment, we have additionally reported a brief analysis of additional components of the purine signalling pathway, including the downstream adenosine deaminase (*ADA*) enzyme, which is predominantly expressed by oligodendroglial and lymphoid populations, and the nucleoside transporters *ENT1-4* which may transport adenosine between the intracellular and extracellular space. Core regulatory enzymes (*CD39* and *CD73*), adenosine receptor subtypes, and adenosine transporters exhibit lineage-specific expression patterns, with regulatory components distributed between both neoplastic and non-neoplastic cell types (Fig. 1b; summarized in schematic in Fig. 9). Glioblastomas exhibit high expression of *CD73*, *CD39*, and adenosine receptors (*ADORA1*, *ADORA3*) compared to other cancer subtypes (Fig. 4a, Fig. S5), and low expression of checkpoint proteins such as *CD274/PD-L1*, suggesting that they may be particularly reliant on purinergic signalling to evade immune activity.

To better understand the states associated with increased expression of *CD39* and *CD73*, we performed a new analysis of transcriptomic data from 168 glioblastoma specimens from the TCGA dataset. This analysis showed that *CD73* and *CD39* are typically correlated with increased quantities of immune cells in the tumor microenvironment. Analysis of transcriptomic signatures associated with functional inflammatory states showed that tumors with high *CD73* exhibit a greater degree of inflammatory signalling in multiple pathways (e.g., interferon-gamma, TNF α , chemokine). Moreover, tumors with coordinate high expression of both *CD73* and *CD39* exhibit particularly high levels of inflammation compared to tumors with low expression. *CD73* and *CD39* expression are also correlated in pediatric high-grade gliomas.

Collectively, these data suggest that key regulatory components of the purinergic signalling pathway are distributed between multiple neoplastic and non-neoplastic cell types, most prominently tumor cells (*CD73*) and myeloid cells (*CD39*). The presence of an inflammatory microenvironment in *CD73*-enriched tumors indicates that *CD73* expression may be upregulated in response to inflammation,

limiting immune activity against tumor cells. Alternatively, CD73 expression may arise due to oncogenic signalling in tumors with permissive lineage-differentiation (AC-like and OPC-like cells) given the association with specific underlying genomic alterations such as *EGFR* amplification and provide a selective advantage to tumor cells which more efficiently generate immunosuppressive adenosine, such as those in close proximity to CD39-expressing myeloid cells.

2) Related to the above, the metabolomics analysis (Figure 4), in which levels of adenosine and the other adenylates are higher in the high-CD73 tumor, maybe the most relevant one. However, I believe that the detected signals are mostly from intracellular metabolites, not from extracellular space metabolites, based on the nature of these methods. Also, to me, the data that show 3-5 folds increases in ATP and ADP is a bit surprising since these are highly abundant metabolites.

We agree with the reviewer that while discrimination of intracellular and extracellular metabolites in human tissue would be of great interest, specific measurement of extracellular metabolite levels is currently not possible using spatially resolved mass spectrometry in frozen tissue specimens and we have modified the text and methods to reflect this point. However, despite this limitation, we believe that the significant correlation that we observe between the CD73 protein expression and adenosine levels in glioblastoma tissue samples (Fig. 6c) suggests that the elevated adenosine levels seen in tumor tissue at least partially reflect production by CD73, which exerts its enzymatic activity in the extracellular space, with consequent effects on adenosine receptor signalling.

To explore the relationship between CD73 and adenosine in the CNS more broadly, we additionally analysed serially sectioned non-neoplastic whole mouse brain by CyCIF and MSI. To perform this multi-modal analysis, we developed an analysis pipeline that allowed us to register immunofluorescence images with mass spectrometry images to determine whether adenosine levels are spatially correlated with CD73 levels. In immediately adjacent serial coronal sections of mouse brain, we measured CD73 levels by immunofluorescence and adenosine levels by mass spectrometry imaging using a 9.4 Tesla Solarix XR FT-ICR mass spectrometry (Fig. 6d-i). Assessed across the entire brain section (removing areas with any artifactual signals due to tissue artifacts; Fig. S9c), we found a strong spatial correlation between CD73 bright cells and high adenosine levels. Co-localization can be readily appreciated in the putamen and globus pallidus, which have been previously established as regions employing purines as physiologic extracellular signalling molecules. Importantly, the correlation between adenosine and CD73 bright cells was statistically significant with or without cell-density correction supporting that the spatial correlation did not result due to cell density in certain parts of the brain. This analysis in mouse tissues suggest that a substantial proportion of adenosine is generated by extracellular CD73 enzymatic activity.

We sought to focus on adenosine and closely related metabolites and regulatory proteins in our study, due to the established importance of CD73 upregulation in cancer, its relevance as a therapeutic target, and the function of adenosine as a key immunomodulatory ligand. However, we recognize the complex physiology of purinergic signalling, which involves many other metabolites and processes. Mass spectrometry imaging allows for the simultaneous measurement and mapping of numerous metabolites simultaneously, and we show some limited data from a broader analysis of other metabolites (Fig. 6a,b) which suggest that other downstream metabolites may exhibit differential levels between tumor and non-tumor tissue. However, analysis of a wide spectrum of purine metabolites and related proteins and understanding their significance in glioma and other cancers as biomarkers or therapeutic targets requires further exploration and is an area of significant interest for future studies using similar methodologies to those developed for this study.

3) While I appreciate the immense work and dense data, Fig 1-3 are somewhat distracting and repetitive

of the previous work. For example, the authors use cell line analysis (Figure 1) regardless of mentioning its uncertainty in the introduction. Figure 2 lacks the correlation analysis of CD73 and CD39. Fig 2E partly shows it, but it is not correlated within the same sample. While CD73 protein expression in multiple brain tumors was assessed in Figure 3, there is no analysis on CD39. Another minor concern in Figure 3 is about the validity of the monoclonal anti-CD73 antibody (Abcam: EPR6115) used for the IHC analysis. Has this antibody ever been validated about its specificity in the setting of IHC using proper controls, such as CD73 knockout/knockdown cells? This antibody has been discontinued by the company. For the spatial analysis, the authors seem not to take advantage of using the IvyGAP database, which would be another key resource that could further verify their findings.

These are very helpful suggestions for improving the manuscript. We agree that the cell line data in Figure 1B and 1C is not definitive in establishing the importance of CD73 and/or purine signalling in glioblastoma and does contain some uncertainty with regard to RNAi knockdown. We did seek to report for the first time that as a category glioblastomas are unique among a large cohort of cell lines in showing a significant intrinsic dependence on CD73 for survival, indicating that this subtype may be particularly well positioned for further study and therapeutic targeting. However, the tumor intrinsic functions of CD73 in glioblastoma are not central to the manuscript and we have therefore removed the siRNA knockdown data from the manuscript.

This comment was also very helpful in that it prompted us to reorganize the manuscript in a way that is more intuitive – we first present scRNA-seq analyses (Figs. 1, 2), followed by transcriptomic analysis of gene signatures in a larger cohort (TCGA) to better understand the connection between CD73 levels and overall cell populations and inflammatory signalling in the glioblastoma tumor microenvironment (Fig. 3), followed by clinicopathologic studies of adult and pediatric high-grade gliomas which will be needed to translate these findings into clinical trials (Figs. 4, 5), studies that demonstrate the potential to measure adenosine directly in tissue sections using MSI (Fig. 6) and then finally the spatial analysis of CD73 tumor cells and CD39 myeloid cells in high grade gliomas in adults and children (Figs. 7,8). We hope that this reorganized conceptual flow is more effective.

- 1.) With regard to correlation between *CD73* and *CD39*, single cell analyses show that these enzymes are predominantly expressed by different cell populations, with strong CD73 expression in tumor cells, and strong CD39 expression in myeloid cells. Accordingly, direct correlation of expression in single cells is of uncertain significance. Analysis of bulk transcriptomic data in the TCGA dataset does not show a significant correlation between CD39 and CD73 in adult glioblastoma (Fig. S7g), though this correlation is demonstrated in pediatric tumors (Fig. 5i,j). However, as with analysis of CD73 expression and genotype, bulk transcriptomic data may exhibit considerable noise due to the heterogeneity of glioma tissue. We considered profiling CD39 protein expression in a wide variety of CNS tumors but given that other CNS tumors are somewhat tangential to the main thrust of the manuscript, we sought to focus on understanding the landscape of CD73 expression in this study due to its key position as a rate-limiting enzyme in the purinergic pathway, its contemporary relevance as a clinical therapeutic target, and evidence of CD73 upregulation in other tumor types, such as solid carcinomas. We do not necessarily predict that overall CD73 and CD39 levels are directly correlated in all tumors, but rather, tumor and myeloid cell populations with expression of these key ectoenzymes are found to exhibit a significant and non-random spatial relationship which has implications for tumor biology and clinical outcome. Nevertheless, focused studies of other CNS tumor types would benefit from combined characterization of CD39 and CD73 expression, including spatial analyses, and we have considerable interest in pursuing this characterization in future research.

We appreciate the reviewer's concerns about the specificity of the anti-CD73 antibody, particularly in light of the unexpected discontinuation of this antibody by the vendor. To further validate the antibodies used in the study, we have pursued several approaches based on methods that we have published for qualifying antibodies for multiplexed imaging studies (Du et al., Nature Protocols 2019; PMID: 31534232). For this validation, we applied four different anti-CD73 antibodies to a tissue microarray of glioblastoma samples (clones D7F9A and EPR6115 which were used in the first submission and additional commercially available anti-CD73 clones 4G6E3 and EPR6114). We then measured the fluorescence signal for each of the antibodies and performed pixel-level correlation analysis which showed that all of the antibodies had excellent cross-correlation but that the signal was not correlated with unrelated markers (Fig. S10a). This is a very reassuring finding, but it is not unexpected because we selected antibodies whose specificity had been confirmed by the vendors in knockout experiments (the 4G6E3 and EPR6114 clones were validated to detect signal in wild type but not knockout A-431 cells; D7F9A specificity was confirmed by knockout experiments in multiple prior publications (PMID: 33086655, 33609609, 28158983)). While EPR6115 was not validated by knockout directly, our pixel-level analysis shows near-identical signal with other CD73 clones, confirming its specificity.

- 2.) Additionally, in this revision we now include data we have acquired from a new pediatric high-grade glioma cohort from the Children's Brain Tumor Network (CBTN). The levels of CD73 that we measured from tissues sections from this cohort by both IHC and by immunofluorescence (using clone D7F9A) correlated strongly with both the mRNA levels and CD73 protein levels (from mass spectrometry measurements) that were acquired from the same tumor samples (Fig. S8b,c), again supporting the fidelity of antibody staining.
- 3.) We also greatly appreciate the helpful suggestion to leverage the spatial data available from the IVY GAP dataset. As we also note above, analysis of RNA-Seq data from 10 glioblastoma specimens, including 7 micro-dissected tumor regions defined by histopathologic annotation or molecular characteristics, showed strong induction of *CD73* expression in peri-necrotic and palisading tumor regions. These regions show significant elevation of *CD73* compared to cellular tumor regions (Fig. 7i), as well as strong expression of hypoxia-response genes relative to other regions (Fig. 7f). Moreover, *in situ* hybridization data confirmed strong expression of *CD73* and hypoxia-response genes such as *LDHA* in peri-necrotic and palisading tumor regions (Fig 7g,h). Thus, both the spatial transcriptomics data and ISH data indicate that there is a link between hypoxia and CD73 levels.

Minor points:

- *Each figure lacks a figure number. It was hard to read.*

We apologize for inadvertently omitting the figure numbers. We have also confirmed that appropriate labels are present on all figure panels.

- *The results of RNAi analysis should be further verified to eliminate the possible off-target effect.*

As noted above in response to the reviewer's critique (point 3), for the sake of space, clarity, and focus we have decided to remove the RNAi cell line analysis that we used to explore tumor intrinsic effects of CD73. That topic is tangential to the main focus of the manuscript on the spatial organization and extrinsic roles of CD73 in immunomodulatory signalling and we have thus refocused the manuscript.

- *Page 8, CD73 is also described as NT5E. It was confusing.*

While we had sought to use the correct scientific nomenclature when referring to the gene name (*NT5E*) in nucleotide-based experiments (e.g., RNA and DNA sequencing) or protein name (CD73) in antibody-based or proteomic experiments (e.g., immunofluorescence, mass spectrometry), we agree that the use of both terms can make the manuscript difficult to understand in places. Accordingly, to improve clarity we have changed the first reference to 'NT5E/CD73' for each section (manuscript, figures, legends), and all subsequent references to CD73 (with italics and/or additional descriptors when referring to the gene or nucleotide-based experiments or analyses). Additionally, we have similarly simplified references to other common genes/proteins in the manuscript (i.e., *ENTPD1/CD39*, *SPI1/PU.1*, *MS4A1/CD20*, *CD3G/CD3*). We expect that these changes will enhance the clarity and readability of the text.

- Page 8, I am not convinced that *EGFR* and *CD73* mRNA is significantly correlated with $r=0.17$ and $p=0.03$.

While the TCGA data do support a statistically significant positive correlation between *EGFR* and *CD73* mRNA (p value = 0.03), we agree with the reviewer that the presented association is relatively weak ($r = 0.17$). However, it is important to note that these data are derived from bulk measurements of mRNA which include non-tumor elements (i.e., immune and other stromal elements) that can often make up >50% of the cell volume of a specimen. Thus, due to the cellular heterogeneity of glioblastoma tumors, bulk RNA sequencing data derived from tumor tissues is accordingly 'noisier' and less specific for assessing the tumor cell intrinsic association of *EGFR* mRNA levels and tumor cell *CD73* expression than are the other single cells analyses that we use in the manuscript (scRNA-seq, multiplexed immunofluorescence, spatial analysis). Despite this limitation and the relatively weak association, the large and well curated dataset available through the TCGA is an additional valuable addition that provides data that is supportive of our overall hypothesis, particularly because the dataset does not demonstrate a statistically supportable association between *CD73* and other prominent genetic drivers of glioblastoma such as *PDGFRA*, *CDK4*, and *NF1*. However, we do acknowledge that this analysis does not definitively establishing a functional link between *CD73* and *EGFR*, and we therefore have accordingly clarified the language in this section of the manuscript and are more circumspect in our interpretation.

To further explore these and other questions we now add to the manuscript an additional analysis of a large cohort of pediatric high-grade gliomas from the Children's Brain Tumor Network (CBTN). In contrast to adult-type IDH-WT glioblastomas which are commonly associated with polysomy 7 and/or *EGFR* gene amplification, analysis of bulk RNA sequencing data from pediatric HGGs (which typically lack these alterations) did not show an association between *CD73* and *EGFR* expression. Moreover, these associations were not observed using single cell analysis in our multiplexed tissue imaging data of pediatric tumor tissue. However, in pediatric gliomas we found that the expression of *CD73* protein and mRNA is more often associated with *PDGFRA* expression, suggesting that genetic drivers of *CD73* expression are likely context and lineage dependent.

- Fig. 5C is difficult to interpret based on the data comparison to tonsil. Probably, the authors should use more proper control.

This is a helpful point. The tonsil controls in this figure are indeed confusing, and we have accordingly removed them from the figure panels. We included multiple tonsil cores in the multiplexed immunofluorescence experiments as 'controls' which assist in technically validating the immune markers, cell segmentation, and lineage calling, and cell counting and expression analysis (and mundane aspects such as providing landmarks for orienting the images). However, while the presentation of this technical control data alongside experimental data from glioblastomas was useful for initial interpretation of the validity of the computational data outputs, the high levels of expression of the

immune markers compress and obscure the most important information in the graphs related to the density of immune populations in tumor samples.

- Page 11: *Expression of HIF1 does not mean that the cells were in hypoxic condition, and there could be a number of reasons.*

This is an important point, and it requires further clarification in the text. In our manuscript we sought to understand the potential mechanisms which may induce CD73 in tumor cells given the absence of genomic alterations at the *NT5E/CD73* gene locus in glioblastoma and other tumors (as shown by analysis of the TCGA datasets). In glioblastoma, hypoxic microenvironments are particularly common due to the pathognomonic aberrant vasculature characteristic of these tumors which frequently exhibit regions of palisading necrosis with adjacent cells exhibiting distinctive hypoxic phenotypes. Accordingly, we sought to evaluate whether hypoxia may be one of the drivers of the high CD73 expression levels seen in GBM relative to other cancer subtypes. We agree with the reviewer that the hypoxia phenotype is complex, and simple expression of HIF1A by immunofluorescence is not sufficient to support this association, and therefore additional data from complementary analyses is needed.

In response to Reviewer #1 (point 3), we listed several pieces of data using different methodologies that support a link between hypoxia and CD73 expression. Among these:

- I.) Analysis of scRNA-seq data showed that tumors with the highest levels of *CD73* expression exhibit significant elevation of the hypoxia gene signature (Fig. 1h-j).
- II.) Cell lineage analysis shows that there is a subset of *CD73* expressing tumor cells with MES-like lineage-differentiation and expression of previously defined (Neftel *et al.* Cell. 2019 Aug 8;178(4):835-849.e21.) hypoxia-response genes such as *HILPDA* (Fig. S1h,i), indicating that hypoxia may modulate *CD73* expression in a subset of cells.
- III.) scRNA-seq analysis of immune populations shows that while *CD73* expression is rare in tumor-associated immune cells, it is most often present in hypoxic myeloid populations.
- IV.) As initially reported, multiplexed immunofluorescence shows a significant association between CD73 expression and HIF1A protein expression at a single cell level, as well as a spatial correlation between CD73 expression and HIF1A levels suggesting that CD73 is elevated in cell neighbourhoods experiencing hypoxia (Fig. 8h, Fig. S10i).
- V.) As we also mention in the response above, microregion spatial transcriptomic data available from IVY GAP showed elevated levels of *CD73* mRNA in peri-necrotic tumor regions and in tumor cells associated with palisading necrosis, which also exhibit strong expression of numerous hypoxia-response genes such as *HILPDA*, *LDHA*, and *DDIT3*. These regions exhibit significantly higher *CD73* compared to cellular tumor regions (Figure 7i).
- VI.) The link between hypoxia and CD73 is further supported by in situ hybridization data from IVY GAP (Figure 7g,h).
- VII.) Analysis of pediatric high-grade gliomas also demonstrated a spatial association between tumor CD73 expression, expression of multiple hypoxia-response genes such as GLUT1, ARG1, and iNOS (though HIF1A was not correlated in this tumor group), and regions of palisading necrosis (Fig. S12d,e)

In total, data from multiple approaches supports a model of a spatially-correlated association between *CD73* expression and hypoxic tumor regions which may be identified by gene signature-based analysis, multiplexed immunofluorescence, or microregion spatial transcriptomics. However, we believe that *CD73* expression may be modulated by multiple mechanisms in adult and pediatric HGG, including oncogene signalling (e.g., from *EGFR* or

PDGFRA activation), lineage-differentiation (AC-like and OPC-like states), hypoxia, and inflammation as detailed in the manuscript and comments. We thank the reviewer for this comment indicating that we needed to provide more clarity and additional confirmatory data about this topic in the main text and figures.

Reviewer #3, expert in mass spectrometry and metabolomics

Remarks to the Author:

This manuscript combines multiplexed tissue imaging, single-cell RNA sequencing, and mass spectrometry imaging to characterize the composition and spatial organization of human glioblastoma components. The authors have performed a thorough and systematic exploration of extracellular purinergic signalling in glioblastoma and its clinical importance, and present some interesting and compelling findings. I have a few minor comments for amendment or clarification.

1) In Figure 1A, there doesn't seem to be much visual distinction between glioblastoma and many of the other subtypes. This figure might benefit from some statistics to demonstrate that glioblastoma did in fact exhibit high expression of CD73 in comparison to other tumor sub-types.

The reviewer raises an important point that is highly relevant for understanding the potential therapeutic value of targeting CD73 in glioblastoma, as well as other cancers which highly express this important immunoregulatory protein. We agree that it was difficult to visually discern significant differences in expression between tumor types. We accordingly now include statistical comparisons of CD73 levels in glioblastoma versus other tumors in the TCGA data (unpaired t-test) and we indicate in Figure 4a whether CD73 levels in specific tumor types are significantly different from glioblastoma (p-values for each comparison are provided Table S3). CD73 levels are significantly higher in glioblastoma than most other tumor types, but similar to the levels in some cancers (e.g., non-small cell lung carcinoma, NSCLC) which have recently shown significant benefits from combinatorial treatment with anti-PD1 (durvalumab) and anti-CD73 antibodies (oleclumab) compared to anti-PD1 antibodies alone (trial: NCT03794544). Of note, anti-PD1/PDL1 monotherapy in glioblastoma has shown only modest activity compared to NSCLC. This might be explained by the very low expression of CD274/PD-L1 in glioblastoma cells and tissues (Fig. S1e). By contrast, strong expression of CD73 in glioblastoma raises the possibility that anti-CD73 combination or monotherapy may provide significant clinical benefits similar to other cancers such as NSCLC with similarly strong CD73 expression.

2) Please add additional figure legends where necessary. For instance, in Figure 1b, what do the blue and red markers denote? In Figure 1C, what do the difference box plots relate to? Such details are not clear in some of the figures.

We apologize for the absence of relevant data labels in some figures. Of note, in Figure 1B (from the previous submission), colored datapoints referred to cell lines with genomic alterations in the CD73/NT5E gene locus. In Figure 1C (from the previous submission), the grey box plots referred to aggregate data from all cell lines in the database, while the pink box plots referred to data from the listed lines (e.g., glioblastoma). However, in response to comments from Reviewer #2, and on further refinement of the manuscript with new data, we have now removed these two panels. These data were focused on a potential intrinsic role of CD73 in promoting tumor cell survival in the absence of other cell populations, rather than the extrinsic function and organization of CD73 in the tumor microenvironment which is the central focus of the manuscript. We have reviewed all other figures to ensure all necessary labels are present.

3) Please clarify the mass spectrometry instrumentation used for MS imaging. The results specify timsTOF MS was used, however the methods only specify details for FT-ICR MS. If these techniques were coupled, please clarify and provide more detail.

In the results section, figure legends, and methods section, we now provide further details regarding the instrumentation and methods used for MS imaging experiments and MSMS measurements. Additional information is also present in Table S11. In brief, the FT-ICR was used for imaging experiments, and timsTOF was used to perform tandem MSMS measurements to validate metabolite identity.

4) In the methods section, the authors state that metabolite identification following MS analysis was only performed putatively. If no analyte identification was performed beyond database matching (i.e. using analytical standards or MS/MS), the authors should be cautious about claiming identifications. It should perhaps be made clearer in the main body of the manuscript that metabolite identifications have not been confirmed. If further steps were taken to identify metabolites, details should be provided.

We have added the following details in the methods section to clarify metabolite identifications: “A standard of adenosine was spotted on the same slide containing the tissues and imaged at the time of mass spectrometry imaging (MSI) acquisition. Further validation of metabolite annotation was performed by tandem MSMS from tissues using a timsTOF (Bruker) and compared to MS/MS measurements performed on standards.” Thank you for indicating the need for more detail.

5) When discussing Figure 6, the authors state that CD73-expressing SOX2+ tumor cells clustered in close-proximity with CD39-expressing PU.1+/CD163+ myeloid cells was observed in “multiple samples”. If pertinent, what proportion of samples was this observed in – it is unclear whether this was observed in just few samples or the majority?

We agree that this is useful information to include. While the interaction strength of CD73 expressing tumor cells and CD39 expressing myeloid cells represents a continuous variable without objective thresholds for high vs. low interaction, we provide the conditions used to define ‘high interaction’ in the methods and clarify the number of cases with this state in the adult glioblastoma CyCIF dataset (n=56). We also clarify the number of cases with a high interaction score, but ‘low’ overall levels of CD73 (assessed by IHC and/or mean core intensity) (n=8). Some of these tumors demonstrate a poor progression-free survival, reinforcing the need to account for the spatial distribution of CD73 and CD39 in addition to the overall levels of CD73 in tumor specimens.

Reviewer #4, expert in multiplex immunofluorescence and tissue imaging

Remarks to the Author:

Coy et al. presented an immunosuppressive role of CD39 and CD73 in Glioblastoma. Specifically, tumor CD73 and microglial CD39 were close to each other and their proximity was associated with poor outcome.

1) CD73/CD39/Adenosine have been shown to be linked to immunosuppressive microenvironments in gliomas/glioblastomas many times:

<https://www.ncbi.nlm.nih.gov/pmc/articles/PMC7526457/>

<https://www.sciencedirect.com/science/article/pii/S1471489220300394>

<https://www.deepdyve.com/lp/oxford-university-press/synergy-between-the-ectoenzymes-cd39-and-cd73-contributes-to-vyckmtAd9f>

and many others

2) Because this is a well known mechanism in immunosuppression, the findings were not surprising. The immunohistochemistry and conventional immunofluorescence should be able to provide the sufficient validation of the CD39/CD73 markers in glioblastomas, along with the spatial analysis.

3) Multiplexing proteins by CyCIF was presented in two figures, but they are already established techniques. The analysis of spatial correlations and maps are not novel and only limited to short four paragraph descriptions. Thus, the power of multiplexing is underutilized here and spatial bioinformatics methods do not yield novel insights that can solve the mechanism of CD73/CD39.

Overall, this paper is a conventional pathology study of two biomarkers and the findings remain within the limits of conventional pathology.

Thus, this paper is not suitable for broad interest of Nature Communications, instead a topical pathology, oncology, or neuroscience journal would be better fit to the scope of the presented work.

As we describe in the General Comments section and response to reviewers, our study seeks to provide a comprehensive characterization of purinergic signalling in glioblastoma, using complementary transcriptomic, mass spectrometry, and tissue-based multiplexed imaging modalities. We analyse multiple large cohorts of human tumors drawn from several independent institutions to derive novel insights about the cellular states associated with expression of key purinergic components and their association with genomic and microenvironmental features. We were puzzled by the reviewer's comments that the manuscript describes a 'conventional pathology study' which includes limited data other than simple CD39 and CD73 marker staining, when in fact the study also incorporates comprehensive single-cell transcriptomic analyses, development of mass spectrometry imaging approaches to directly measure purine metabolites such as adenosine in human tissue (which has not been reported previously), and multiplexed immunofluorescence and spatial approaches which leverage recent methods that have not been applied to this pathway in glioma. Each of these approaches provides significant novel insights, and significantly clarify the prior literature. In response to the reviewer, we note that:

- i.) With regard to 1), in our study we leverage scRNA-seq analysis from multiple independent datasets to show that *CD73* is rare in tumor-associated immune cells (Nat Neurosci. 2021 Apr;24(4):595-610.; Cell. 2019 Aug 8;178(4):835-849.e21.). Though multiple studies have shown enrichment of *CD73* expression in glioblastoma, previous literature has reported conflicting results on the contributions and roles of different cell types to purinergic signalling in glioblastoma, and frequently examine only a limited subset of cell types or non-physiologic cell lines. Resolving the confusion in the literature surrounding the cell states associated with purine pathway activity is in fact a major inspiration for our work and is required to optimally and rationally leverage emerging treatments which target components of this pathway:
 - a. One recent study of FACS-sorted immune cells (Nature Medicine. 2020 Jan;26(1):39-46) (which did not characterize tumor cells) argued that *CD73*-expressing macrophages represent a large fraction of tumor-associated myeloid cells in human tumors and play an important role in purinergic signalling and development of chemoresistance in glioblastoma. However, we present data from multiple large datasets showing that while *CD73* expression is present only on a very small subset of myeloid cells, these cells represent a small fraction of overall immune cells (~0.5%), with much more prominent expression of *CD73* by glioma tumor cells. Accordingly, it seems likely that tumor cells, and not myeloid or other populations, likely provide the major contribution to *CD73* enzymatic activity in most glioblastomas.

- b. Another study cited above by the reviewer (<https://www.ncbi.nlm.nih.gov/pmc/articles/PMC7526457/>) used murine glioma cell lines (GL261) to study purinergic signalling in glioblastoma. However, in this study the cells used did not natively express *CD73*, but the authors relied on stable transgenic expression of *CD73* as noted in the manuscript: “Contrary to human glioma cells, which have been shown to express *CD73*, GL261 did not recapitulate this feature. To rule out that the lack of *CD73* expression in the glioma cells is responsible for the modest treatment effect, we stably overexpressed *CD73* in GL261 cells and confirmed the *CD73* enzymatic function. The lack of *CD73* expression on tumor cells may have been one reason for the modest therapeutic efficacy.”

Though murine (GL261) and human tumors may exhibit many similarities and murine models enable *in vivo* functional perturbations which provide valuable insights, there are substantial differences in immune populations and regulation between immunocompetent mouse models and human tumors (Nat Neurosci. 2021 Apr;24(4):595-610.). Notably, mouse tumors exhibit a larger component of lymphoid and dendritic cells than human tumors, which have a more dominant myeloid population. Such differences may inform interpretations of the relative significance and function of different populations in tumor-immune interactions. Notably, the cited study reported a prominent role for *CD73* expression on CD4 and CD8 T cells in glioma, but such populations are extremely rare in newly diagnosed human glioblastomas as we show in our manuscript through analyses of multiple scRNA-seq datasets including an analysis of all cell populations in 21 adult GBM and 7 pediatric HGG (Neftef *et al.* Cell. 2019 Aug 8;178(4):835-849.e21.) (Fig. 1), a separate analysis of immune and microglial sub-populations from 4 adult glioblastoma (Sankowski *et al.* Nat Neurosci. 2019 Dec;22(12):2098-2110.) and a new analysis of CD45-positive immune and myeloid cells from 7 newly diagnosed and 4 recurrent adult GBM (Pombo Antunes *et al.* Nat Neurosci. 2021 Apr;24(4):595-610.). We confirm this finding directly intact human glioma specimens by multiplexed immunofluorescence (Fig. 7).

Accordingly, while this study provides interesting and useful functional data on *in vivo* disruption of *CD73* and adenosine receptors (e.g., $A_{2A}R$) in murine models, its generalizability to human glioblastomas is not clear given the substantial differences in the cell types and states associated with expression of key purine regulators in human tumors that we demonstrate in our study from multiple large independent datasets and several experimental methods. Notably, it seems likely that T cells are not the major contributor to *CD73* or *CD39* activity in newly diagnosed glioblastomas as previously reported by several authors, but rather, tumor cells contribute *CD73* while myeloid cells, particularly tissue-native microglia, provide *CD39* activity. However, as noted this dynamic may shift in recurrent tumors, in which there is an influx of *CD39*-expressing T regulatory cells, an additional new finding of our analysis.

- c. Another study cited by the reviewer (Xu *et al.* Neuro Oncol. 2013 Sep;15(9):1160-72. doi: 10.1093/neuonc/not067) was one of the first to raise the possibility of a functional interaction between *CD39*-expressing immune cells and *CD73*-expressing tumor cells, which was further elucidated by functional studies in cell culture. This study showed that multiple glioma cell lines and tumor cells in human glioma tissue specimens exhibit strong expression of *CD73* but not *CD39*, similar to our results. However, in this study *CD39* expression was assessed only on peripheral blood and tumor-associated CD4 T cells, with targeted analysis of this population but not myeloid populations, and comprehensive single cell analysis of human tumors was not performed. Our results

confirm that CD39-expressing T cells are rare in newly diagnosed glioblastoma, with much more CD39 contributed by myeloid populations (Fig. 1,2).

Thus, our study provides a significantly more comprehensive view than that was provided in the prior literature, leveraging robust single-cell analysis of CD39 and CD73 in human glioblastomas from multiple independent scRNA-seq datasets to cross-validate findings. We believe our more rigorous characterization of the cell populations associated with CD39 and CD73 expression confirms that tumor cells are the predominant source of CD73, while myeloid cells are the predominant source of CD39, and that some prior literature conclusions may have been incomplete as a result of a limited scope of analysis or the models chosen for study (i.e., analysis of sorted immune or T cells but not tumor cells, or analysis of transgenic murine models). Our study accordingly provides a valuable contribution to this literature on this point by clarifying these discrepancies in a larger multi-institutional dataset. We thank the reviewer for allowing us to further clarify these points and encouraging us to further validate our results in additional datasets.

- ii.) Though as the reviewer indicates *CD73* expression by glioblastoma tumor cells has been described in the literature, the association between *CD73* expression and lineage differentiation has not been previously reported. We report for the first time that *CD73* expression is associated with *EGFR* amplification and astrocyte-like differentiation in adult glioblastomas, and *PDGFRA* expression and OPC-like differentiation in pediatric gliomas. Developing a greater understanding of the intermixed and heterogeneous cell populations in glioblastoma tumors is of critical importance to address the substantial resistance of these tumors to chemotherapy and immunotherapeutic approaches. Analysing expression of purinergic pathway components within the context of lineage differentiation and cellular heterogeneity provides substantial new insights into the states associated with purine pathway activity. This analysis may also provide guidance for designing treatment approaches, by indicating that tumors with certain types of differentiation or underlying genotypes may be more vulnerable to anti-purinergic therapy.
- iii.) In accordance with reviewer comments, we have now included a further characterization of the immune cell lineages and functional sub-types, including dendritic and myeloid sub-types, associated with *CD73* and *CD39* expression in glioblastoma, in both primary and recurrent tumors. While other groups have characterized immune populations associated with purinergic marker expression, we believe that this dataset now represents the most extensive characterization of immune cell states associated with marker expression in a large multi-institutional cohort of tumors.
- iv.) The reviewer suggests that conventional CD39 and CD73 staining alone may be sufficient to characterize this pathway in human tumors. Simple characterization of CD73 or CD39 expression in tumors is indeed possible by immunohistochemistry and we use this method to profile CD73 expression in a wide variety of adult and pediatric CNS tumors to understand the relative expression levels of different tumor subtypes. From this analysis we confirm that CD73 is highly expressed in both adult and pediatric gliomas. We also characterize CD73 expression for the first time in multiple tumor subtypes (ependymoma, medulloblastoma, pediatric HGG, H3K27M-mutant diffuse midline glioma), showing that some exhibit minimal expression of CD73 (medulloblastoma and ependymoma), while others may represent additional potential targets of anti-CD73 therapy and subjects of further study (meningioma, pediatric high- and low-grade gliomas).

However, it is very clear from our results that simple application of such markers alone is insufficient to fully understand the purinergic pathway and its spatial organization in high-grade gliomas. Definition and mapping of specific cell subtypes (e.g., B cells, T cells, macrophages, microglia, tumor cells, endothelial cells) and correlation with purine marker expression requires simultaneous application of numerous tissue markers, necessitating more computationally intensive multiplexed imaging approaches. Furthermore, simultaneous single-cell correlation of tumor cell associated CD73 with other markers such as EGFR, HIF1A, or Ki-67 that we describe in our study (which provide important insight into the functional state of CD73 expressing cells) would not be possible with simple immunofluorescence. Similarly detecting immune states such as the CD11b, CD163, CD39, PU.1+ myeloid state, and their association with tumor cells with CD73 (Fig. 8i) would not be possible with conventional approaches. Our study is the first to define and map these states in glioblastoma tissue; these datasets allowed us to validate the findings of scRNA-seq analysis, as well as quantitatively explore the spatial associations of CD73-expressing tumor cells for the first time.

v.) As the reviewer notes, CyCIF and the spatial analytic methods employed in this study are not novel, and their use and application has been reported previously by our group in multiple publications. A partial list of these publications in journals with broad readerships includes:

Lin *et al.* *Elife*. 2018 Jul 11;7:e31657; Jerby-Arnon *et al.*, *Cell*. 2018 Nov 1;175(4):984-997.e24; Du *et al.*, *Nat Protoc*. 2019 Oct;14(10):2900-2930; Färkkilä A *et al.*, *Nat Commun*. 2020 Mar 19;11(1):1459; Gaglia *et al.*, *Nat Cell Biol*. 2020 Feb;22(2):151-158; Izar *et al.*, *Nat Med*. 2020 Aug;26(8):1271-1279; Baker *et al.*, *Cell Syst*. 2020 Sep 23;11(3):272-285.e9; Mehta *et al.*, *Nat Cancer*. 2021 Jan;2(1):66-82; Liu *et al.*, *Nat Med*. 2021 Jun;27(6):985-992; Rashid *et al.*, *Nat Biomed Eng*. 2021; Gaglia *et al.*, *Nat Cell Biol*. 2022 Mar;24(3):316-326; Launonen *et al.*, *Nat Commun*. 2022 Feb 11;13(1):835; Schapiro *et al.*, *Nat Methods*. 2022 Mar;19(3):311-315.

However, such methods have not been applied to analysis of the purinergic pathway, the organization of which is not well-defined in human glioblastoma tissue. In this study, we show for the first time that CD73-expressing tumor cells are spatially associated with CD39-expressing microglia which exhibit a specific differentiation state defined by CD11b and CD163 expression. This analysis further reveals that interaction between these populations is significantly associated with progression-free survival in IDH-WT glioblastoma. These results leverage the unique strengths of multiplexed immunofluorescence by simultaneously correlating phenotypes (which require numerous markers to define) and functional communities of cells. Defining these tissue cellular communities/neighborhoods is a major goal for our team and others in the NCI Moonshot funded Human Tumor Atlas Network (HTAN, *Cell*. 2020 Apr 16;181(2):236-249). The spatial organization and neighborhoods that we describe in our study may be of significant clinical importance if independently validated by subsequent studies, as there are presently few prognostic and predictive clinicopathologic indicators in IDH-WT glioblastoma.

While previously studies have alluded to the interaction between immune and tumor populations in purinergic regulation in glioblastoma, we demonstrate a much more granular analysis of the spatial interactions between CD73-expressing tumor cells and other populations. This data again confirms that myeloid cells, particularly microglia (rather than peripherally derived macrophages), are likely the most important population in regulating ATP-catabolism and immunosuppressive adenosine production in glioblastoma, and not T cell populations as previously described (Xu *et al.* *Neuro Oncol*. 2013 Sep;15(9):1160-72.).

- vi.) In response to reviewer comments, we have also expanded the spatial analysis to include Latent Dirichlet Association analysis (LDA) (Fig. 8d-f), which our group has been developing for characterization of functional neighbourhoods in human tissue specimens. LDA has been widely used in biodiversity studies because it permits the detection of both gradual and abrupt changes in the composition and arrangements of natural elements (e.g., decomposition of biodiversity data into latent communities such as identifying the species of trees in a forest) while effectively accounting for uncertainty and missing data; LDA has been recently applied by our group and others to study the spatial organization of cells in tissues (by the Angelo group in *Nat Immunol.* 2022 Feb;23(2):318-329, and by our group in <https://www.biorxiv.org/content/10.1101/2021.05.23.445310v2> which is currently in press at *Cancer Discovery*). LDA is a probabilistic modelling method that reduces complex assemblies of intermixed entities into distinct component communities (i.e., recurrent cellular neighbourhoods). This analysis complements our prior cell mapping (Fig. 8a), high-resolution 3D-imaging (Fig. 8b,c), and spatial statistics analyses (Fig. 8, Fig. S12) to show that discrete neighbourhoods with association of SOX2-positive tumor cells and CD163, PU.1, CD11b positive myeloid cells and CD39 and CD73 are detectable by unbiased analysis in glioblastoma.
- vii.) Additionally, to extend our analysis, we performed an additional 24-plex multiplexed immunofluorescence and spatial analysis using a new cohort of pediatric high-grade gliomas (HGG) from the Children's Brain Tumor Network (CBTN). Similar to adult glioblastoma, this data showed that CD73 is highly expressed by tumor cells in pediatric HGG and that there is a spatial association between tumor cell CD73 and microglial CD39 (including H3K27M-mutant diffuse midline glioma). Thus, we generalize and extend our findings to multiple additional important tumor categories. We also confirm the findings of a spatial association between tumor cell CD73 and hypoxia using spatial transcriptomic and *in situ* hybridization data from the IVY GAP study, as well as extend this analysis to pediatric HGG. We hope that these extended analyses will further clarify and demonstrate the importance of spatial analysis in understanding purinergic signalling in glioblastoma and other tumors, as well as further confirm the clinically relevant spatial organization of purine components in these tumor subtypes.
- viii.) Finally, we have clarified and extended the description and discussion of the spatial methods used in the manuscript to better clarify the novelty and importance of this analysis.

We hope to have provided the reviewer with additional clarification for the context of the study, as well as the multiple important novel findings reported in the study. We hope that these findings may guide future research of purinergic signalling in glioblastoma, as well as clinical development of anti-purinergic therapies in glioblastoma and pediatric high-grade glioma. Moreover, we hope to have better elucidated the importance of cross-validating findings with multiple analytic methods that capture the full spectrum of cell types and characterize single-cell transcriptional states as well as spatially preserved native tissue organization. While single-cell transcriptomic approaches may reveal the specific lineage and functional states associated with expression of purine regulatory enzymes and pathway activity, multiplexed immunofluorescence and mass spectrometry imaging enable the characterization of spatial correlations between CD73 and cell populations and metabolite levels in the tumor microenvironment, providing novel insights that are not available from dissociative methods and which have not been defined previously.

Recently reported results from the COAST trial in lung adenocarcinoma showed a robust improvement in clinical response with combination anti-CD73 (oleclumab) and anti-PD1 (durvalumab) compared to durvalumab monotherapy (trial: NCT03794544). These results provide additional validation

for the potential effectiveness of targeting CD73, and we hope that our findings may inspire further clinical development of therapeutic purinergic inhibitors for both adults and children with high grade gliomas.

Reviewers' Comments:

Reviewer #1:

Remarks to the Author:

The authors have very nicely and thoroughly addressed the comments that were raised. The manuscript is ready for acceptance and I congratulate the authors for their interesting study.

Reviewer #2:

Remarks to the Author:

I have reviewed the revised manuscript, which significantly has improved and addressed the previously denoted concerns. I appreciate the authors' meticulous responses and efforts to bring to this point.

Reviewer #3:

Remarks to the Author:

After revision of the manuscript, I feel that the authors have addressed all of the points I raised. I have no further comments or recommendations to make.